# Estimating Optimal Policy Value
# in Linear Contextual Bandits Beyond Gaussianity

**Jonathan N. Lee**                                                    *jnl@stanford.edu*
*Stanford University*

**Weihao Kong**                                              *weihaokong@google.com*
*Google Research*

**Aldo Pacchiano**                                                 *pacchian@bu.edu*
*Boston University*

**Vidya Muthukumar**                                    *vmuthukumar8@gatech.edu*
*Georgia Institute of Technology*

**Emma Brunskill**                                          *ebrun@cs.stanford.edu*
*Stanford University*

**Reviewed on OpenReview:** *https://openreview.net/forum?id=RUNiIDU8P7*

## Abstract

In many bandit problems, the maximal reward achievable by a policy is often unknown in advance. We consider the problem of estimating the optimal policy value in the sublinear data regime before the optimal policy is even learnable. We refer to this as $V^*$ estimation. It was previously shown that fast $V^*$ estimation is possible but only in disjoint linear bandits with Gaussian covariates. Whether this is possible for more realistic context distributions has remained an open and important question for tasks such as model selection. In this paper, we first provide lower bounds showing that this general problem is hard. However, under stronger assumptions, we give an algorithm and analysis proving that $\widetilde{\mathcal{O}}(\sqrt{d})$ sublinear estimation of $V^*$ is indeed information-theoretically possible, where $d$ is the dimension. We subsequently introduce a practical and computationally efficient algorithm that estimates a problem-specific upper bound on $V^*$, valid for general distributions and tight for Gaussian context distributions. We prove our algorithm requires only $\widetilde{\mathcal{O}}(\sqrt{d})$ samples to estimate the upper bound. We use this upper bound in conjunction with the estimator to derive novel and improved guarantees for several applications in bandit model selection and testing for treatment effects. We present promising experimental benefits on a semi-synthetic simulation using historical data on warfarin treatment dosage outcomes.

## 1 Introduction

Classic paradigms in multi-armed bandits (MAB), contextual bandits (CB), and reinforcement learning (RL) consider a plethora of objectives from best-policy identification to regret minimization. The meta-objective is typically to learn an explicit, near-optimal *policy* from samples. The best achievable value by a policy in our chosen policy class, typically denoted as the *optimal value $V^*$* is often unknown ahead of time. This quantity may depend in complex ways on the nature of the context space, the action space, and the class of function approximators used to represent the policy class. In many applications, the impact of these properties and design choices is often unclear a priori.

In such situations, it would be useful if it were possible to quickly estimate $V^*$ and assess the target performance value, in order to decide whether to adjust the problem specification or model before spending

valuable resources learning to optimize the desired objective. For example, prior work that used online policy search to optimize educational activity selection has sometimes found that some of the educational activities contribute little to student success (Antonova et al., 2016). In such settings, if the resulting performance is inadequate, knowing this early could enable a system designer to halt and then explore improvements, such as introducing new actions or treatments (Mandel et al., 2017), refining the state representation to enable additional customization (Keramati & Brunskill, 2019) or exploring alternate model classes. These efforts can change the system in order to more effectively support student learning.

A related objective is the problem of *on-the-fly* online model selection in bandit and RL settings (Foster et al., 2019). From a theoretical perspective, a large number of recent attempts at model selection leverage some type of $V^*$-estimation (or closely related gap estimation) subroutine (Agarwal et al., 2017; Foster et al., 2019; Chatterji et al., 2020; Pacchiano et al., 2020b; Lee et al., 2021). In particular, Lee et al. (2021) show that significantly improved model selection regret guarantees would be possible if one could hypothetically estimate $V^*$ faster than the optimal policy. Despite these important practical and theoretical implications, the amount of data needed to estimate $V^*$ is not well understood in real-world settings. A naive approach would be to attempt to estimate an optimal policy from samples and then plug-in an estimate of $V^*$; however, this may necessitate a full algorithm deployment and a prohibitive number of samples in high-stakes applications.

In this work, we pursue a more ambitious agenda and ask: *is it possible to estimate the optimal value $V^*$ faster than learning an optimal policy?* Prior work suggests that this is surprisingly possible but only in a quite restricted setting: Kong et al. (2020) show that, with disjoint linear contextual bandits with Gaussian contexts and known covariances, it is possible to estimate $V^*$ accurately with only $\widetilde{\mathcal{O}}(K\sqrt{d}/\epsilon^2)$ samples, a substantial improvement over the $\widetilde{\mathcal{O}}(d/\epsilon^2)$ samples required to learn a good policy in high dimensional settings (Chu et al., 2011).[1] Unfortunately, the strong distribution assumptions make the Gaussian-specific algorithm impractical for many scenarios and inapplicable to other theoretical problems like model selection that deal with much richer distributions.

The purpose of this work is twofold. (1) We aim to provide an information-theoretic characterization of $V^*$ estimation for linear contextual bandits under much more general distributional assumptions that are more realistic in practice and comparable to typical scenarios of online learning and bandits. In particular, we aim to understand under what conditions sublinear $V^*$ estimation is and is not possible. (2) We aim to devise practical methods to achieve sublinear estimation and in turn realize significant improvements in problems such as model selection and hypothesis testing.

## 1.1 Contributions

As our first contribution, we make progress towards an information-theoretic characterization of the $V^*$ estimation problem (Section 3.2). We prove lower bounds showing that, without structure, one cannot hope to estimate $V^*$ accurately with sample complexity smaller than $\Omega(d)$ in general. Despite this, our first major positive result (Algorithm 1 and Theorem 1) shows that $V^*$ estimation with sample complexity that scales as $\sqrt{d}$ is still information-theoretically possible beyond the Gaussian-context setting given some mild distributional assumptions when lower-order moments are known. In particular, we give an algorithm that achieves $\widetilde{\mathcal{O}}(2^{C'_K/\epsilon}\sqrt{d}/\epsilon^5)$ sample complexity.[2] While the bound is not without undesirable features owing to the exponential dependence on $\epsilon^{-1}$ and $C'_K$, the significance of the result lies in the proof that estimating $V^*$ is possible in the sublinear regime in $d$. In particular, suppose that $K$ is constant (small) and $\epsilon$ is a constant target accuracy. Our result shows that the number of samples needed to get the desired fixed target accuracy scales with only the square root of the problem size $d$, *meaning that sometimes we may need fewer samples than there are parameters to achieve constant accuracy.* This is in contrast to a naive plug-in approach of trying to estimate $\theta$ directly, a guarantee that requires the number of samples to be at least $d$ to say anything non-trivial.

---

[1] Here, $\widetilde{\mathcal{O}}$ omits polylogarithmic factors and lower order terms. $d$ is the dimension, $\epsilon$ is the target accuracy, and $K$ is the number of actions.

[2] $C'_K$ is a constant that depends only on $K$ – see Corollary 3.5 for details. For all intents and purposes, we will consider this bound only when $K$ and $\epsilon$ are constant (e.g. $K = 2$ and $\epsilon = 0.1$).

Our second key contribution is a computationally and statistically efficient algorithm that estimates informative *upper bounds* on $V^*$ with sample complexity $\widetilde{\mathcal{O}}(\sqrt{d}/\epsilon^2)$ (Algorithm 2 and Theorem 2). This algorithm avoids the exponential dependence and holds with weaker requirements.

We first leverage upper bounds on $V^*$ to tighten the bandit model selection guarantee of Foster et al. (2019) in the large-action setting from $\mathcal{O}(K^{1/3})$ to $\mathcal{O}(\sqrt{\log K})$ (Theorem 3). Second, we show that upper bounds on $V^*$ can be used to develop provably efficient tests for treatment effects to decide whether we should expand our set of treatments from a low-stakes-but-limited set, to a more ambitious treatment set that may be costly to implement (Theorem 4), and provide experimental evidence on a synthetic domain and on a semi-synthetic simulation using historical data on warfarin treatment dosage.

## 1.2 Related Work

One can show (see Proposition A.1) that in the MAB setting, estimating $V^*$ is no easier than solving a best arm identification task Bubeck et al. (2009); Audibert et al. (2010); Gabillon et al. (2012); Karnin et al. (2013); Jun et al. (2016); Even-Dar et al. (2006); Maron & Moore (1994); Mnih et al. (2008); Jamieson et al. (2014); Katz-Samuels & Jamieson (2020). In the linear setting (Hoffman et al., 2014; Soare et al., 2014; Karnin, 2016; Tao et al., 2018; Xu et al., 2018; Fiez et al., 2019; Jedra & Proutiere, 2020), as well as in the non-disjoint linear contextual bandit setting (Chu et al., 2011), there is significantly more shared structure across actions: all the unknown information in the problem is encapsulated in *one* unknown, $d$-dimensional parameter. Surprisingly little work has been spent on estimating $V^*$ even though we do know that certain functionals of the unknown parameter, such as the signal-to-noise ratio (Verzelen et al., 2018; Dicker, 2014; Kong & Valiant, 2018) are estimable at the fast $\widetilde{\mathcal{O}}(\frac{\sqrt{d}}{n})$ rate. The $V^*$ estimation problem was first proposed by Kong et al. (2020) who showed estimation is possible with $\widetilde{\mathcal{O}}(\frac{\sqrt{d}K}{\epsilon^2})$ samples. However, the algorithmic tools are highly specialized to Gaussian context distributions. Thus, we require novel approaches to handle more general context distributions. We are able to show that $V^*$ estimation is possible under significantly broader distribution models with many practical implications.

A particularly critical application of $V^*$ estimation arises in online model selection in CB. Multiple approaches to model selection make use of estimators of $V^*$ to weed out misspecified models (Agarwal et al., 2017; Lee et al., 2021; Pacchiano et al., 2020a; Foster et al., 2019; Chatterji et al., 2020; Pacchiano et al., 2020b; Lee et al., 2022a;b; Muthukumar & Krishnamurthy, 2022; Ghosh et al., 2021). In our current work, we leverage our faster estimators of $V^*$ to improve the model selection results of Foster et al. (2019) in the linear CB setting. Our more sophisticated approaches to $V^*$ estimation imply a logarithmic $\mathcal{O}(\sqrt{\log K})$ scaling on the leading term in the regret, exponentially improving upon the $\mathcal{O}(K^{1/3})$ of the original work.

## 2 Preliminaries

**Notation.** We use $[n] = \{1, \ldots, n\}$ for $n \in \mathbb{N}$. For any vector $v \in \mathbb{R}^d$, $\|v\| = \|v\|_2$. For any matrix $M \in \mathbb{R}^{d \times d}$, $\|M\|$ denotes the operator norm and $\|M\|_F$ the Frobenius norm. When $M \succeq 0$, $\|v\|_M := \sqrt{v^\top M v}$. We denote the $d$-dimensional unit sphere $\mathbf{S}^{d-1} = \{v \in \mathbb{R}^d \ : \ \|v\| = 1\}$. We call $\binom{[n]}{s}$ the set of $s$-combinations of $[n]$ and use the symbol $\mathbb{I}_d$ to denote the $d \times d$ identity matrix. We use $C, C_1, C_2, \ldots$ to refer to absolute constants independent of problem parameters. Throughout, we use the failure probability $\delta \leq 1/e$. The inequality $a \lesssim b$ implies $a \leq Cb$ for some constant $C > 0$. For a random variable $Z$, we denote the variance as $\mathrm{var}(Z)$ and $\|Z\|_{L^2}^2 := \mathbb{E}|Z|^2$. $Z$ is said to be sub-Gaussian if there exists $\sigma > 0$ such that $\mathbb{E}\left[|Z|^p\right]^{1/p} \leq \sigma \sqrt{p}$ for all $p \geq 1$ and we define $\|Z\|_{\psi_2}$ as the smallest such $\sigma$: $\|Z\|_{\psi_2} := \sup_{p \geq 1} p^{-1/2} \mathbb{E}\left[|Z|^p\right]^{1/p}$. We also use $Z \sim \mathrm{subG}(\sigma^2)$ to denote that $\|Z\|_{\psi_2} \lesssim \sigma$. A random vector $\overline{Z}$ is sub-Gaussian if there exists $\sigma$ such that $\|\overline{Z}\|_{\psi_2} := \sup_{v \in \mathbf{S}^{d-1}} \|\langle \overline{Z}, v \rangle\|_{\psi_2} \leq \sigma$.

**Setting.** We consider the stochastic contextual bandit problem with a set of contexts $\mathcal{X}$ and a finite set of actions $\mathcal{A} = [K]$ (with $K = |\mathcal{A}|$). At each timestep, a context-reward pair $(X_t, Y_t)$ is sampled i.i.d from a fixed distribution $\mathcal{D}$, where $X_t \in \mathcal{X}$ and $Y_t \in \mathbb{R}^K$ is a reward vector indexable by actions from $\mathcal{A}$. Upon seeing the context $X_t$, the learner chooses an action $A_t$ and collects reward $Y_t(A_t)$. Let $r^*(x, a) = \mathbb{E}[Y(a) \mid x]$ and let $\pi^*$ be the optimal policy such that $\pi^*(x) \in \arg\max_{a \in \mathcal{A}} r^*(x, a)$.

The quantity of interest throughout this paper is the average value of the optimal policy, defined as

$$V^* := \mathbb{E}\, Y(\pi^*(X)) = \mathbb{E}_X \max_{a \in \mathcal{A}} r^*(X, a) \tag{1}$$

For an arbitrary policy $\pi : \mathcal{X} \to \mathcal{A}$, we define $V^\pi = \mathbb{E}\, Y(\pi(X))$. A typical objective in contextual bandits is to minimize regret $\text{Reg}_T(\pi_{1:T}) = \sum_{t \in [T]} V^* - V^{\pi_t}$. While our focus will be estimation of the quantity $V^*$, we will consider the regret problem in applications of our results (Section 4.1).

We restrict our attention to the *linear* contextual bandit, which is a well-studied sub-class of the general setting described above (Auer, 2002; Chu et al., 2011; Abbasi-Yadkori et al., 2011). We assume that there is a known feature map $\phi : \mathcal{X} \times \mathcal{A} \to \mathbb{R}^d$ and unknown parameter vector $\theta \in \mathbb{R}^d$ such that $r^*(x, a) = \langle \phi(x, a), \theta \rangle$ for all $x \in \mathcal{X}$ and $a \in \mathcal{A}$. As in standard CB settings, we consider the case where $|\mathcal{X}|$ is prohibitively large (i.e. infinite) and $d \ll |\mathcal{X}|$, but $d$ can still be very large (e.g. on the order of $T$). As is standard, we assume that the noise $\eta(a) := Y(a) - r^*(X, a)$ is independent of $X$ and sub-Gaussian with $\|\eta\|_{\psi_2} \leq \sigma = \mathcal{O}(1)$, and the features $\phi(X, a)$ are sub-Gaussian with $\|\phi(X, a)\|_{\psi_2} \leq \tau = \mathcal{O}(1)$. We will assume that $\mathbb{E}[\phi(X, a)] = 0$ for any fixed $a \in \mathcal{A}$. For the results of Section 3, this is easily relaxed; however, for Section 4, there are a number of potential ways to deal with non-zero mean arms that may be lead to varying degrees of approximation error. We maintain mean zero for simplicity and defer further discussion to the relevant sections. Finally, in order to enable non-trivial results (see justification in Section 3.1), we will consider well-conditioned distributions.

**Assumption 1.** *The covariance matrices given by $\Sigma_a := \mathbb{E}_X \left[ \phi(X, a)\phi(X, a)^\top \right]$ and*

$$\Sigma_{a,a'} = \mathbb{E}_X \left[ \left( \phi(X, a) - \phi(X, a') \right) \left( \phi(X, a) - \phi(X, a') \right)^\top \right]$$

*for $a \neq a'$ are known. The minimum eigenvalue of the average covariance matrix of all actions $a \in [K]$ is bounded below by a positive constant $\lambda_{\min}(\Sigma) \geq \rho > 0$ where $\Sigma := \frac{1}{K} \sum_{a \in [K]} \Sigma_a$.*

The assumption that the covariances are known is made for simplicity of the exposition, as many of our results continue to apply if the covariances are unknown but there is access to $\widetilde{\mathcal{O}}(d)$ samples of *unlabeled* contexts $X$, which allows for sufficiently accurate estimation of the necessary covariances. We provide thorough details and derivations of this extension in Appendix F. This is common in many applications where there exist data about the target context population (such as customers or patients), but with no actions or rewards e.g. in bandits (Zanette et al., 2021; Kong et al., 2020) and active and semi-supervised learning (Hanneke, 2014; Singh et al., 2008). Interestingly, for our model selection results, no additional unlabeled data is required at all (see Theorem 3 and Appendix D.1).

## 3 Information-Theoretic Results

Our first order of business is to make progress towards a sample complexity characterization of the formal $V^*$ estimation problem. We are interested in understanding under what conditions it is possible to achieve sample complexity that scales sublinearly as $\sqrt{d}$ in an information-theoretic sense. Kong et al. (2020) showed that $\Omega(K\sqrt{d}/\epsilon^2)$ samples were necessary even in the Gaussian case, but our general setting is a major departure from their work. We will find that in general the linear structure and well-conditioning (Assumption 1) are essentially necessary to avoid linear dependence on $d$. We will then move on to our first major result, a moment-based algorithm that achieves $\widetilde{\mathcal{O}}(2^{C'_K/\epsilon}\sqrt{d}/\epsilon^5)$ sample complexity, resolving affirmatively and constructively the question of whether sublinear estimation in $d$ is information-theoretically possible in these general distributions.

While the exponential dependence on $\epsilon^{-1}$ is undesirable and we ultimately provide a more practical approximation in Section 4, we believe the sample complexity result of this section is important to characterize for two reasons. (1) The results provide first steps of a characterization of the exact problem of estimating $V^*$ to arbitrary accuracy rather than approximations of it. (2) The algorithm and theorem serve as theoretical evidence that sublinear estimation of $V^*$ is possible in the first place. Prior to this, it was not clear whether this was possible, regardless of $\epsilon$ dependence. This can serve as a stepping stone towards improved analyses.

---

**Algorithm 1** Moment-Based Estimator

---

1: **Input**: Number of samples $n$, failure probability $\delta$, degree $t$, coefficients $\{c_\alpha\}_{|\alpha| \leq t}$ of polynomial approximator $p_t$.
2: Define $q = 48 \log(1/\delta)$, $m = \frac{n}{q}$, initialize empty datasets $D^1, \ldots, D^q$
3: Whiten features with $\phi(\cdot) = \Sigma^{-1/2}\phi(\cdot)$.
4: **for** $k = 1, \ldots, q$ **do**
5:    **for** $i = 1, \ldots, m$ **do**
6:       Sample independently $x_i^k \sim \mathcal{D}$ and $a_i^k \sim \text{Unif}[K]$. Receive reward $y_i^k$.
7:       Set $\phi_i^k = \phi(x_i^k, a_i^k)$.
8:       Add tuple $(\phi_i^k, y_i^k)$ to $D^k$.
9:    **end for**
10: **end for**
11: **for** $\alpha$ such that $s := |\alpha| \leq t$ **do**
12:    Compute independent moment estimators $\forall k = 1, \ldots, q$:

$$\hat{S}_{m,\alpha}^k := \frac{1}{\binom{m}{s}} \sum_{\ell \in \binom{[m]}{s}} \mathbb{E}_X \prod_{j \in [s]} \left\langle y_{\ell_j}^k \phi_{\ell_j}^k, \phi(X, a_{(j)}) \right\rangle \qquad (2)$$

13:    Set $\hat{S}_{n,\alpha} \leftarrow \text{median}\{\hat{S}_{m,\alpha}^k\}_{k=1}^q$.
14: **end for**
15: **Return** $\hat{S}_n := \sum_{\alpha \,:\, |\alpha| \leq t} c_\alpha \hat{S}_{n,\alpha}$.

---

### 3.1 Hardness Results

A natural starting point is the classical $K$-armed bandit problem where $r^*(x, a)$ is independent of $x$ (and for this part only we assume that the means are non-zero). It is not immediately clear whether one can estimate $V^*$ with better dependence on either $K$ or $\epsilon$. Proposition 3.1 implies that, for $V^*$ to be estimable, the bandit must also be learnable. See Appendix A for a formal statement. Proposition 3.2 shows $\Omega(d)$ samples necessary for linear contextual bandits when Assumption 1 is violated.

**Proposition 3.1.** *[informal] There exists a class of $K$-armed bandit problems such that any algorithm that returns an $\epsilon$-optimal estimate of $V^*$ with constant probability must use $\Omega(K/\epsilon^2)$ samples.*

**Proposition 3.2.** *There exists a class of linear contextual bandit problems with $\phi : \mathcal{X} \times \mathcal{A} \to \mathbb{R}^d$ and $K \geq d$ such that Assumption 1 is violated and any algorithm that returns an $\epsilon$-optimal estimate of $V^*$ with probability at least $2/3$ must use $\Omega(d/\epsilon^2)$ samples. Under the same assumption, there exists a class of problems with $K = 2$ and an absolute constant $c$, such that any algorithm that returns an $c$-optimal estimate of $V^*$ with probability at least $2/3$ must use $\Omega(d)$ samples.*

These lower bounds suggest that, without more structure, $V^*$ estimation is no easier than learning the optimal policy itself. One might wonder then if a sublinear in $d$ sample complexity is possible at all without a Gaussian assumption. In the following section, we answer this affirmatively.

### 3.2 A Moment-Based Estimator

We now present, to our knowledge, the first algorithm that achieves sublinear sample complexity in $d$ by leveraging the well-conditioning of the covariance matrices (Assumption 1). We remark that the method of Kong et al. (2020) crucially leveraged a Gaussian assumption which meant the problem could be specified solely by a mean and covariance matrix, thus easing the task of estimation. Such steps are unfortunately insufficient to generalize beyond Gaussian cases, where we are likely to be dealing with distributions that are far more rich than what can be specified by a covariance matrix alone. These limitations motivate a fundamentally new approach, which we present here.

We present the full algorithm for $V^*$ estimation with general distributions in Algorithm 1. The main idea is to first consider a $t$th-order $K$-variate polynomial approximation of the $K$-variate max function, and reduce the problem to estimating the expectation of the polynomial. We define such an approximator generically as in the definition to follow.

As an important note, while we will state all bounds in terms of $K$ to explicitly showcase dependencies in this section, we will generally assume $K$ is small or constant when discussing the significance of results.

**Definition 3.3.** *Consider a $t$-degree polynomial $p_t : [-1,1]^K \to \mathbb{R}$ written as $p_t(z_1, \ldots, z_K) = \sum_{|\alpha| \leq t} c_\alpha \prod_{a \in [K]} z_a^{\alpha_a}$ where $z \in [-1,1]^K$, $\alpha$ is a multiset given by $\alpha = \{\alpha_1, \ldots, \alpha_K\}$ for $\alpha_a \in \mathbb{N}$, and we denote $|\alpha| = \sum_{a \in [K]} \alpha_a$. We say that $p_t$ is a $(\zeta, c_{\max})$-polynomial approximator of the $K$-variate $\max$ function on a given CB instance if it satisfies the following conditions:*

1. *$\sup_{x \in \mathcal{X}} |p_t(z_1(x), \ldots, z_K(x)) - \max\{z_1(x), \ldots, z_K(x)\}| \leq \zeta$ where $z_a(x) = \langle \theta, \phi(x,a) \rangle$ and*

2. *$|c_\alpha| \leq c_{\max}$ for all multisets $\alpha$ with $|\alpha| \leq t$.*

Many such polynomial approximators exist and we will discuss several examples shortly with various trade-offs. Algorithm 1 proceeds by estimating the quantity $\mathbb{E}_X [p_t(\{\langle \theta, \phi(X,a) \rangle\}_{a \in \mathcal{A}})]$ which is guaranteed to be $\zeta$-close to $V^*$ if $p_t$ satisfies Definition 3.3. We achieve this by estimating individual $\alpha$-moments between the $\{\langle \theta, \phi(X,a) \rangle\}_{a \in \mathcal{A}}$ random variables using Equation (2) [3]. The intuition is that there are $\binom{m}{|\alpha|}$ ways to construct independent unbiased estimators of $\theta$ in a single term. This step turns out to be crucial for the proof as we show that only sublinear in $d$ samples are sufficient to get accurate estimation of each $S_\alpha$ (Theorem 5 in Appendix B). This follows from a novel variance bound on the individual estimators (Lemma B.1). Before proceeding, we state several technical assumptions specific to the guarantee of this estimator.

**Assumption 2.** *There exists a constant $L > 0$ such that for any $v, u \in \mathbb{R}^d$,*

$$\mathbb{E}\left[\langle \phi(X,a), v \rangle^2 \langle \phi(X,a), u \rangle^2\right] \leq L \cdot \mathbb{E}\left[\langle \phi(X,a), v \rangle^2\right] \mathbb{E}\left[\langle \phi(X,a), u \rangle^2\right]$$

*for all $a \in \mathcal{A}$.*

**Assumption 3.** *For all $a \in \mathcal{A}$ and $x \in \mathcal{X}$, it holds that $\langle \phi(x,a), \theta \rangle \in [-1,1]$.*

Assumption 2, which is also made in linear regression (Kong & Valiant, 2018), is a Bernstein-like condition which says that the fourth moments are controlled by the second moments. However, it is milder as we do not require *all* moments to be controlled. It can also be easily shown to follow from standard hypercontractive conditions (Bakshi & Prasad, 2021). Assumption 3 is a simple boundedness assumption that is almost universal in bandit studies. We furthermore assume that all moments up to degree $t$ of $\langle \phi(X, \cdot), v \rangle$ for any $v \in \mathbf{S}^{d-1}$ are known or can be computed. Such knowledge could come from a large collection of unlabeled or non-interaction batch data (as we demonstrate in Section 4.1.2). Our main technical result of this section, stated below, shows that it is indeed possible to estimate $V^*$ to accuracy up to the polynomial approximation with sample complexity that scales as $\sqrt{d}$ using Algorithm 1.

**Theorem 1.** *Let Assumptions 1, 2, and 3 hold. Let $p_t$ be a $t$-degree $(\zeta, c_{\max})$-polynomial approximator and let $\hat{S}_n$ be the output of Algorithm 1. Suppose that $n \geq 96 \log(1/\delta)t$. There is a constant $C > 0$, depending only on $\tau$, $\sigma$, and $L$, such that with probability at least $1 - t(et/K + e)^K \delta$, $|V^* - \hat{S}_n|$ is bounded by*

$$\zeta + c_{\max} t \left(et/K + e\right)^K \sum_{s=1}^t \left(\frac{Ct^3 \sqrt{d}}{n} \log(1/\delta)\right)^{s/2}$$

The crucial aspect to note in the bound of Theorem 1 is that $d$ only appears in terms that are polynomial in $\frac{\sqrt{d}}{n}$, in contrast to the typical $\frac{d}{n}$ rates required for estimating $\theta$ itself or learning the optimal policy (Chu et al., 2011). As we will see in examples, this readily translates to a sample complexity bound whose dependence

---

[3]Note that in (2), for the multiset $\alpha$ of size $s := |\alpha|$ and $j \in [s]$, we use $a_{(j)}$ to mean the action $a_{(j)} = \max\{a' : \sum_{b < a'} \alpha_b \leq j\}$. That is to say, if we considered the tuple $(\phi(X,1), \ldots, \phi(X,1), \phi(X,2), \ldots, \phi(X,2), \ldots, \phi(X,K))$ where $\phi(X,a)$ is repeated $\alpha_a$ times, $\phi(X, a_{(j)})$ refers to the $j$th element of this tuple.

on $d$ is only $\sqrt{d}$ and dependence on $\epsilon$ and $K$ depends on the instantiation of the polynomial approximator $p_t$. Another interesting observation is that, as far as estimation error is concerned, we have avoided exponential dependence on $t$, which is an easy pitfall because the variance of monomials can easily pick up order $\Theta(2^t)$. Here, this is avoided as long as $n \gtrsim t^3 \sqrt{d} \log(1/\delta)$, which makes each term in the summation less than 1 and thus they are *smaller* for larger $s \in [t]$. The factor $1/n^{s/2}$ acts as a modulating effect for any terms that also pick up exponential in $\frac{s}{2}$ dependence. Our solution to this is one of the primary technical novelties of the proof and is a critical consequence of Lemma B.1. This makes Theorem 1 a fairly modular result: we may plug-in polynomial approximators to observe problem-specific trade-offs between $\zeta$ and $c_{\max}$.

We will now use Theorem 1 to prove that it is indeed possible to estimate $V^*$ in the unlearnable regime with sample complexity sublinear in $d$ in general cases. Specifically, it is possible to estimate $V^*$ even when $d \gg n$, i.e. for high-dimensional problems. We do this by instantiating several example polynomial approximators. We start with the most general version of this result, which does not rely on any additional structure in the bandit problem beyond what's needed in Theorem 1.

**Example 3.4.** *Consider a generic contextual bandit satisfying the assumptions of Theorem 1. There exists a $t$-degree polynomial approximator $p_t^{BBL}$ (named after Bagby et al. (2002)) satisfying Definition 3.3 with $\zeta = \frac{C_K}{t}$ and $c_{\max} = \frac{(2et)^{2K+1}2^{3t}}{K^K}$ where $C_K$ is a constant that depends only on $K$ (see Lemma B.4 for a formal existence statement).*

**Corollary 3.5.** *The estimator $\hat{S}_n$ generated by Algorithm 1 with polynomial $p_t^{BBL}$ satisfies $|V^* - \hat{S}_n| \le \epsilon$ for $\epsilon < 1$ with probability at least $1 - \delta$, $t = 2C_K/\epsilon$, and sample complexity*

$$\mathcal{O}\left(\left(\frac{C_K}{K\epsilon}\right)^K 2^{C_K/\epsilon} \cdot \frac{K\sqrt{d}}{\epsilon^5} \cdot \log\left(\frac{C_K}{\epsilon\delta}\right)\right). \tag{3}$$

Corollary 3.5 has only a $\sqrt{d}$ dependence in the sample complexity of the estimation task, showing sublinear $\sqrt{d}$ sample complexity is possible in the unlearnable regime when $d \gg n$, a broader set of situations. The exponential dependence on $\epsilon^{-1}$, $K$, and $C_K$ is undesired, but still illustrates that interesting behavior is possible. In particular, consider the case where $K$ and $\epsilon^{-1}$ are constant (and thus $C_K$ is also constant) such as $K = 2$ and $\epsilon = 0.1$. Our result suggests that, for large $d$, it is possible to estimate $V^*$ to within 0.1 accuracy in fewer samples than there are parameters. Recall that a plug-in estimator would require *at least $d$* samples to achieve a non-trivial guarantee even if $K$ and $\epsilon^{-1}$ are constant. An alternative view is to consider the asymptotic regime, as $n \to \infty$, in which we require only that $\sqrt{d}/n \to 0$ ensures convergence in probability, while classical bounds would require that $d/n \to 0$.

One might wonder if $K$ and $\epsilon^{-1}$ dependence can be substantially improved while maintaining the same level of generality. One possible solution is to ask if a better polynomial approximator exists since Theorem 1 should be sufficient as long as $c_{\max}$ and $\zeta$ can be properly controlled in Definition 3.3. Unfortunately, for the most general polynomials it is well-known that the $2^t$ order of $c_{\max}$ is essentially tight even for $K = 1$ (Markov, 1892; Sherstov, 2012). While we conjecture that this is in general unimprovable, it remains an important open question to understand. The intuition developed here may serve as a possible stepping-off point for future work to resolve it.

Despite exponential dependence in this most general setting, we can actually achieve much stronger results by employing better polynomial approximators in interesting special cases. Our next example shows that certain contextual bandits can be handled by special case polynomials that have tighter bounds on the coefficients. In this case, we have sublinear $\sqrt{d}$ dependence and purely polynomial dependence on $1/\epsilon$ which comes as a result of the refined polynomial approximator.

**Example 3.6.** *Consider a CB problem satisfying the conditions of Theorem 1 where $\theta \in \mathbb{R}^d$ is a vector of all zeros except at some unknown coordinate $i_*$ where $\theta_{i_*} = \omega \in [-1, 1]$ and $|\omega| = \Omega(1)$. Furthermore, $\phi^i(x, a) \in \{-1, 0, 1\}$, where $\phi^i$ denotes the $i$th coordinate of $\phi$, and for each $(i, x, a)$ tuple, there is another $a'$ such that $\phi^i(x, a) = -\phi^i(x, a')$. Then, there exists a $K$-degree polynomial $p_K^{bin}$ satisfying Definition 3.3 on this instance with $\zeta = 0$ and $c_{\max} = |\omega|^{-K}$.*

**Corollary 3.7.** *On the class of problems in Example 3.6, $\hat{S}_n$ with $p_K^{bin}$ satisfies $|V^* - \hat{S}_n| \le \epsilon$ for $\epsilon < 1$ with probability at least $1 - \delta$ and sample complexity $\mathcal{O}\left(K^8 2^{2K} \cdot \frac{\sqrt{d}}{\epsilon^2} \cdot \log\left(\frac{2K}{\delta}\right)\right)$.*

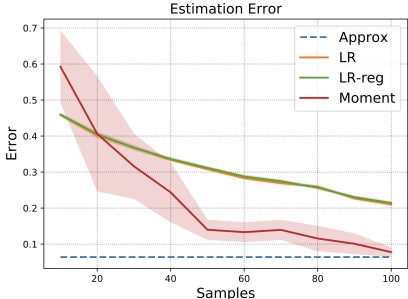

Figure 1: Estimation error of Algorithm 1 (Moment) is shown in red on a simulated high-dimensional CB domain with $d = 300$. The blue dashed line (Approx) represents the bias due to the polynomial approximation. Moment greatly outperforms plug-in baselines (LR in orange and LR-reg in green). Error bars represent standard error over 10 trials.

**Numerical Experiments.** While the results of this section are meant to be purely information-theoretic, we investigate its practical efficacy to provide a deeper understanding. Exact details can be found in Appendix H. We consider a simulated high-dimensional CB setting where $K = 2$ and $d = 300$ in the "unlearnable regime" where $n \leq 100$ is much smaller than $d$. We set the degree $t = 2$ and did not split the data. The error of Algorithm 1 in red (Moment) is shown in Figure 1 compared to linear and ridge regression (regularization $\lambda = 1$) plug-in baselines. Though computationally burdensome, Algorithm 1 is surprisingly effective. These observations suggest that $V^*$ estimation could be a valuable tool for quickly assessing models before committing to a large number of samples to learn the optimal policy.

## 4   An Efficient Procedure: Estimating Upper Bounds on the Optimal Policy Value

The information-theoretic results so far have suggested that there are potentially sizable gains to be realized even in the unlearnable regime, but we seek a practically useful method. In this section, we address this problem by proposing a method (Algorithm 2) that instead approximates $V^*$ with an upper bound and then estimates this upper bound efficiently. The new estimator is both statistically and computationally efficient, achieving a $\widetilde{\mathcal{O}}(\sqrt{d}/\epsilon^2)$ rate. Moreover, we show it has immediate important applications. The procedure is given in Algorithm 2. Our main insight is to view $\{\langle \theta, \phi(X, a) \rangle\}_{a \in [K]}$ as a general stochastic process and then *majorize* it using a Gaussian process $\{Z_a\}_{a \in [K]} \sim \mathcal{N}(0, \Lambda)$ for some covariance matrix $\Lambda \in \mathbb{S}_+^K$ that has approximately the same increments $\| \langle \phi(X, a), \theta \rangle - \langle \phi(X, a'), \theta \rangle \|_{L^2}^2$ as the original process.[4] By majorize, we mean that $\mathbb{E} \max_a Z_a \gtrsim V^*$. Then we attempt to estimate this Gaussian process from data, which is significantly easier than estimating higher moments. To do this, the data is split it into two halves. Each half generates parameter estimators $\hat{\theta}$ and $\hat{\theta}'$ and we construct the increment estimator $\hat{\beta}_{a,a'} = \hat{\theta}^\top \Sigma_{a,a'} \hat{\theta}'$ where we recall that $\Sigma_{a,a'} := \mathbb{E}_X [(\phi(X, a) - \phi(X, a')) (\phi(X, a) - \phi(X, a'))^\top]$ from Assumption 1. From here, we can find a covariance matrix $\tilde{\Lambda}$ that nearly achieves these increments by solving a simple optimization problem (Line 14). This fully specifies our estimate of the majorizing Gaussian process $\tilde{Z} \sim \mathcal{N}(0, \tilde{\Lambda})$, from which we can compute its expected maximum by Monte-Carlo sampling. We show that, with infinite data, this quantity is exactly an upper bound on $V^*$ (under mild conditions) and the aforementioned estimation procedure can estimate this quantity at a sublinear rate. Specific to this section, we first make the following technical assumption on the process.

**Assumption 4.** *There exists an absolute constant $L_0$ such that, for all $v \in \mathbf{S}^{d-1}$ and $a, a' \in [K]$,* $\| \langle \phi(X, a), v \rangle - \langle \phi(X, a'), v \rangle \|_{\psi_2} \leq L_0 \| \langle \phi(X, a), v \rangle - \langle \phi(X, a'), v \rangle \|_{L^2}.$

Assumption 4 can be thought of as a joint sub-Gaussianity assumption on the sequence of random variables $\{\langle \phi(X, a), v \rangle\}_{v \in \mathbf{S}^{d-1}, a \in \mathcal{A}}$ (in the sense of Vershynin (2018)). This assumption is still fairly general, including

---

[4]Despite similar terminology, Algorithm 2 is distinct from Gaussian process *regression*. Rather than modeling a reward function with a prior and kernel, we are majorizing a reward process with a generic Gaussian process.

---

**Algorithm 2** Estimator of Upper Bound on $V^*$

---

1: **Input**: Number of interactions $n$, failure probability $\delta$.
2: Set $m = \frac{n}{2}$.
3: Initialize empty dataset $D$.
4: Whiten features with $\phi(\cdot) = \Sigma^{-1/2}\phi(\cdot)$.
5: **for** $i = 1, \ldots, n$ **do**
6:    Sample independently $x_i \sim \mathcal{D}$ and $a_i \sim \text{Unif}[K]$. Receive reward $y_i$
7:    Add tuple $(x_i, a_i, y_i)$ to $D$
8: **end for**
9: Split dataset $D$ evenly into $\{x_i, a_i, y_i\}_{i \in [m]}$ and $\{x'_i, a'_i, y'_i\}_{i \in [m]}$.
10: Compute estimators $\hat{\theta} = \frac{1}{m}\sum_{i \in [m]} y_i \phi(x_i, a_i)$ and $\hat{\theta}' = \frac{1}{m}\sum_{i \in [m]} y'_i \phi(x'_i, a'_i)$
11: **for** $a, a' \in [K]$ such that $a \neq a'$ **do**
12:    Set $\hat{\beta}_{a,a'} := \hat{\theta}^\top \Sigma_{a,a'} \hat{\theta}'$
13: **end for**
14: $\tilde{\Lambda} = \arg\min_{\lambda \in \mathbb{S}_+^K} \max_{a \neq a'} |\lambda_{a,a} + \lambda_{a',a'} - 2\lambda_{a,a'} - \hat{\beta}_{a,a'}|$
15: **Return** $\hat{U} = \mathbb{E}\max_{a \in [K]} \tilde{Z}$ where $\tilde{Z} \sim \mathcal{N}(0, \tilde{\Lambda})$

---

Gaussian processes, symmetric Bernoulli processes, and spherical distributions in $\mathbb{R}^{dK}$, as well as all rotations and linear combinations. There are indeed counterexamples, but, they involve specialized couplings of $\{\langle \phi(X,a), v \rangle\}$ based on hard truncations that are unlikely to be generated in a linear CB setting (see Appendix C.2 for details).

**Theorem 2.** *Under Assumptions 1 and 4, there are absolute constants $C_1, C_2 > 0$ and a Gaussian process $(Z_a)_{a \in [K]}$ with $U = \mathbb{E}\max_a Z_a$ such that $V^* \leq C_1 \cdot U \leq C_2 \cdot V^* \sqrt{\log K}$. Furthermore, for $\delta \leq 1/e$, Algorithm 2 produces $\hat{U}$ such that with probability at least $1 - \delta$, $|U - \hat{U}| \leq \mathcal{O}\left( \frac{\sqrt{\|\theta\|}\log(K/\delta)}{n^{1/4}} + \frac{d^{1/4}\log^{3/2}(dK/\delta)}{\sqrt{n}} \right)$. If the process $\phi(X, \cdot)$ happens to be Gaussian, we have $V^* = U$ and $\hat{U}$ estimates $V^*$ exactly.*

The proof of Theorem 2 is contained in Appendix C.1. Despite the fact that the majorizing process $Z$ is unknown due to $\theta$ being unknown, we show that our novel procedure of estimating the increments $\beta_{a,a'}$ at a fast rate is sufficient. The norm $\|\theta\|$ is problem-dependent (and may be $0$ – see Section 4.1). Note that $U$ is an adaptive upper bound: it is exactly $V^*$ whenever $\phi(X, \cdot)$ is a Gaussian process, so we are able to *exactly* estimate $V^*$ in the Gaussian and non-disjoint setting. The tightness of the bound changes with the quality of the Gaussian approximation. A notable case when the approximation might be poor is in the symmetric Bernoulli case $\phi(X,a) \sim \text{Unif}\{-1,1\}^d$, and $\theta = (1, 0, \ldots, 0)^\top$, in which case $V^* = \Theta(1)$ but $U = \Theta(\sqrt{\log K})$. Here our estimation is loose by a $\sqrt{\log K}$ factor. Theorem 2 shows that this symmetric Bernoulli example is the worst case, and $U$ itself is at worst bounded by $\mathcal{O}(V^*\sqrt{\log K})$. In Appendix C.3, we discuss possible ways of relaxing the assumption that the features have zero mean.

## 4.1 Applications

To demonstrate the usefulness of upper bounds on $V^*$ and the estimator of Section 4, we consider several important applications. We emphasize that the remaining results will all hold for *unknown covariance matrices* estimated from data alone.

### 4.1.1 Model Selection with Many Actions

Model selection for linear contextual bandits is attracting recent interest (Agarwal et al., 2017; Foster et al., 2019). Lee et al. (2021) posited that fast $V^*$ estimators could improve model selection regret guarantees. We now show how Theorem 2 can be assist this goal. Following a similar setup to that of Foster et al. (2019), we consider two nested linear function classes $\mathcal{F}_1$ and $\mathcal{F}_2$ where $\mathcal{F}_i = \{(s,a) \mapsto \langle \phi_i(s,a), \theta \rangle \ : \ \theta \in \mathbb{R}^{d_i}\}$. Here, $\phi_i$ maps to $\mathbb{R}^{d_i}$ where $d_1 < d_2$, and the first $d_i$ components of $\phi_1$ are identical to $\phi_2$: aka the function classes are *nested*, i.e. $\mathcal{F}_1 \subseteq \mathcal{F}_2$. The objective is to minimize regret, as defined in the Preliminaries, over $T$ online rounds. We assume that $\mathcal{F}_2$ realizes $r^*$. If $\mathcal{F}_1$ also realizes $r^*$, we would ideally like the regret to scale

---

**Algorithm 3** Treatment Effect Test

---

1: **Input**: Number of interactions $n$, failure probability $\delta$, unlabeled contexts $\{\tilde{x}_j\}_{j\in[p]}$
2: Set $n' = \frac{n}{2}$.
3: Compute $\hat{\Sigma}_a = \frac{1}{p} \sum_{i=1}^{p} \phi(\tilde{x}_j, a)\phi(\tilde{x}_j, a)^\top$ for all $a \in \mathcal{A}_2$.
4: Compute $\hat{\Sigma}_{a,a'} = \frac{1}{p} \sum_{i=1}^{p} \left(\phi(\tilde{x}_j, a) - \phi(\tilde{x}_j, a')\right) \left(\phi(\tilde{x}_j, a) - \phi(\tilde{x}_j, a')\right)^\top$ for all $a, a' \in \mathcal{A}_2$ with $a \neq a'$.
5: Compute $\hat{\Sigma} = \frac{1}{|\mathcal{A}_1|p} \sum_{i=1}^{p} \sum_{a\in\mathcal{A}_1} \phi(\tilde{x}_i, a)\phi(\tilde{x}_i, a)^\top$.
6: Initialize empty dataset $D$
7: **for** $i = 1, \ldots, n$ **do**
8:     Sample independently $x_i \sim \mathcal{D}$ and $a_i \sim \text{Unif}\,\mathcal{A}_1$. Receive reward $y_i$
9:     Add tuple $(x_i, a_i, y_i)$ to $D$
10: **end for**
11: Split dataset $D$ evenly into $\{x_i, a_i, y_i\}_{i\in[n']}$ and $\{x'_i, a'_i, y'_i\}_{i\in[n']}$.
12: Compute estimators $\hat{\theta} = \hat{\Sigma}^{-1}\left(\frac{1}{n'}\sum_{i\in[n']} y_i\phi(x_i, a_i)\right)$ and $\hat{\theta}' = \hat{\Sigma}^{-1}\left(\frac{1}{n'}\sum_{i\in[n']} y'_i\phi(x'_i, a'_i)\right)$
13: **for** $a, a' \in [K]$ such that $a \neq a'$ **do**
14:     Set $\hat{\beta}_{a,a'} := \hat{\theta}^\top \hat{\Sigma}_{a,a'} \hat{\theta}'$
15: **end for**
16: $\tilde{\Lambda} = \arg\min_{\lambda\in\mathbb{S}_+^K} \max_{a\neq a'} |\lambda_{a,a} + \lambda_{a',a'} - 2\lambda_{a,a'} - \hat{\beta}_{a,a'}|$
17: **Return** $\hat{U} = \mathbb{E}\max_{a\in[K]} \tilde{Z}$ where $\tilde{Z} \sim \mathcal{N}(0, \tilde{\Lambda})$

---

with $d_* := d_1$ instead of $d_2$, as $d_1 \ll d_2$ potentially. If $\mathcal{F}_1$ does not realize $r^*$, then we accept regret scaling with $d_* = d_2$. Since the class of minimal complexity that realizes $r^*$ is unknown, model selection algorithms aim to automatically learn this online. Our improved estimators for upper bounds on $V^*$ imply improved rates for model selection and are particularly attractive in their leading dependence on $K$.

Our algorithm, given in Algorithm 4 in Appendix D, resembles that of Foster et al. (2019) where we interleave rounds of uniform exploration and rounds of running Exp4-IX (Neu, 2015) with a chosen model class. The key difference is statistical: we rely on a hypothesis test that reframes estimation of the value gap between the two model classes as a $V^*$ estimation problem. The high-level intuition is as follows. Let $V_1^*$ and $V_2^*$ denote the maximal achievable policy values under $\mathcal{F}_1$ and $\mathcal{F}_2$ respectively. Define $\theta_i = \arg\min_{\theta\in\mathbb{R}^{d_i}} \frac{1}{K}\sum_{a\in\mathcal{A}}(\langle\phi_i(x,a),\theta\rangle - r^*(x,a))^2$. We show that the gap between the two values is bounded

$$V_2^* - V_1^* \leq \mathbb{E}\max_{a\in[K]}\left\langle\phi_2(X,a), \theta_2 - (\theta_1, 0)^\top\right\rangle + \mathbb{E}\max_{a\in[K]}\left\langle\phi_2(X,a), (\theta_1, 0)^\top - \theta_2\right\rangle \tag{4}$$

The key observation is that the summands on the right are just $V^*$ values but for a new CB problem where the parameter is the difference of optimal parameters of the original CB. Theorem 2 shows that we can find a majorizing Gaussian process and an upper bound $U$ on each summand[5]. We can then conclude that $V_2^* - V_1^* \leq 2U$. Now consider the case where $\mathcal{F}_1$ realizes $r^*$. If this is true, then the $V^*$ of the new CB problem is equal to zero. Consequently Theorem 2 would imply that $U = 0$ too. Thus, if we can estimate $U$ at a fast rate, we can decide quickly whether $\mathcal{F}_1$ realizes $r^*$. This is precisely what the second part of Theorem 2 allows us to do. In the end, Theorem 3 shows our dependence on $K$ in the leading term is only logarithmic in the leading term, which improves exponentially on the poly$(K)$ factor in Foster et al. (2019).

**Theorem 3.** *With probability at least $1 - \delta$, Algorithm 4 achieves*

$$Reg_T = \mathcal{O}\left(d_*^{1/4}T^{2/3}\log^{3/2}(d_*TK/\delta)\log^{1/2}(K)\right) + \mathcal{O}\left(\sqrt{d_*KT\log(d_*)}\cdot\log(d_*TK/\delta)\right) \tag{5}$$

### 4.1.2 Testing for Treatment Effect

When designing experiments, one must often determine which treatments, or actions, to include in a given problem. In many cases, such as Mandel et al. (2017); Chaiyachati et al. (2018), choosing from a certain

---

[5]In this case, the upper bound $U$ happens to be the same for both summands.

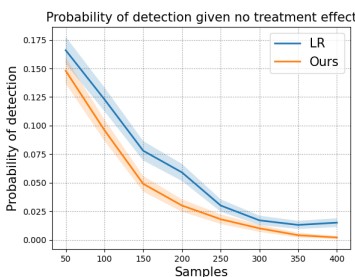 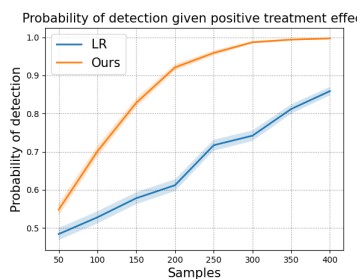 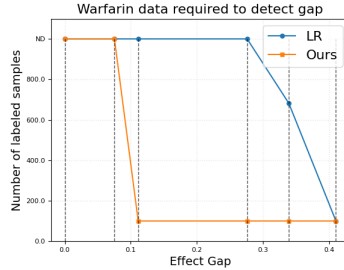

Figure 2: A comparison between our proposed test for treatment effect in Section 4.1.2 and a test based on a linear regression baseline (LR) with $d = 600$ and $p = 2000$. The left figure shows the case where there is no treatment effect, $\Delta = 0$ (closer to 0 is better). The middle figure shows the case where $\Delta = 0.133 > 0$ (closer to 1 is better). Our test is more sensitive to real effects as shown in the middle figure, while remaining comparable in terms of the fraction of false positives. Error bands represent standard error on 1000 replicates. The right figure demonstrates the estimator on warfarin dosage data. The objective is to determine if an optimal policy, which can select from two dosages to administer to patients, would significantly improve outcomes over simply administering a fixed dosage to all patients. The $x$-axis shows the ground truth average effect gap (up to noise) of a policy that incorporates the second dosage over the baseline dosage (e.g. {medium, high} over just {medium}). The $y$-axis shows the number of labeled samples required to detect the given effect gap. ND denotes that the effect was not detected given the available labeled data. A good estimator should quickly detect larger effect sizes (drop to zero along the $x$-axis) but may remain uncertain when the effect size is negligible. The effect gap sizes are ordered from least to greatest in the following order: {med, high}, {med, low}, {low, high}, {low, med}, {high, med}, {high, low}.

treatment set $\mathcal{A}_2$ might lead to far better outcomes and thus larger values of $V^*$ for a given problem than what is achievable by a base set $\mathcal{A}_1$. However, since $\mathcal{A}_2$ is more ambitious in scope, it also may be significantly more costly for the experiment designer to implement and collect data for a variety of reasons. Thus, if the improvement in $V^*$ of using all treatments $\mathcal{A}_1 \cup \mathcal{A}_2$ over just an inexpensive subset $\mathcal{A}_1$ is negligible, it would be more cost-effective to simply use $\mathcal{A}_1$.

We refer to this problem as testing for treatment effect. We assume that both $\mathcal{A}_1$ and $\mathcal{A}_2$ share a known control action $a_0$ such that $r^*(x, a_0) = 0$ for all $x \in \mathcal{X}$. We say that $\mathcal{A}_2$ has zero *additional* treatment effect over the control $a_0$ if $\mathbb{E} \max_{a \in \mathcal{A}_2} \langle \phi(X, a), \theta \rangle = 0$. We do not assume that the covariance matrices are known. Rather, the designer has $p$ *unlabeled* contexts. Typically, $p \gg n$.

We show the method in Algorithm 3 and sketch the main idea. First, define $V_1^* := \mathbb{E} \max_{a \in \mathcal{A}_1} \langle \phi(X, a), \theta \rangle$ and $V_2^* := \mathbb{E} \max_{a \in \mathcal{A}_1 \cup \mathcal{A}_2} \langle \phi(X, a), \theta \rangle$. We prove that the gap can be bounded by the $V^*$ value of a new CB problem over just the action set $\mathcal{A}_2$: $V_2^* - V_1^* \leq \mathbb{E} \max_{a \in \mathcal{A}_2} \langle \phi(X, a), \theta \rangle$. Then, we simply apply Theorem 2 to test if $U = 0$. We then estimate $U$ by the procedure outlined in Algorithm 2, but this time using the unlabeled data to estimate the covariance matrices. Leveraging Theorem 2, we show there exists a test statistic $\hat{U}$ output from Algorithm 3 and test, $\Psi = 0$ if $\hat{U} \lesssim \widetilde{\mathcal{O}}\left(\sqrt{\frac{d}{p}} + \frac{d^{1/4}}{\sqrt{n}}\right)$ and $\Psi = 1$ otherwise, with a low chance of false positives but also a good chance of detecting small effects with limited data.

**Theorem 4.** *Let $K := |\mathcal{A}_1 \cup \mathcal{A}_2|$ be the total number of actions and let $\Delta := V_2^* - V_1^*$ be the gap. Suppose that the size of the unlabeled dataset is $p = \Omega(d \log(K/\delta))$. Then the following conditions are true each with probability at least $1 - \delta$: (1) When there is no additional treatment effect of $\mathcal{A}_2$ over the control, then the test correctly outputs $\Psi = 0$. (2) If $\Delta = \widetilde{\Omega}(\sqrt{d/p} + d^{1/4}/\sqrt{n})$, then the test correctly outputs $\Psi = 1$.*

To achieve the same guarantee with a plug-in estimator as a test statistic, we would only be able to detect a gap $\Delta$ of size $\widetilde{\Omega}(\sqrt{d/n})$– or equivalently, it would require at least $n \geq d$ datapoints to estimate anything non-trivial, and the plug-in estimator accuracy would be going at a much slower rate. Note our bound in Theorem 4 is never worse than $\widetilde{\Omega}(\sqrt{d/n})$ since a constant fraction of the labeled data can always be treated as unlabeled data.

**Synthetic Testing Experiments.** To demonstrate the effectiveness and practicality of our proposed method for testing treatments, we conducted numerical experiments comparing our test with a plug-in linear regression (LR) baseline. We simulated a high-dimensional contextual bandit learning setting with $|\mathcal{A}_1| = 3$ actions and $|\mathcal{A}_2| = 2$ and $d = 600$ features. Note that this remains in the sublinear regime due to the fact that the number of samples $n$ never exceeds $d$.

We first evaluated the methods in the *no-effect* setting ($\Delta = 0$) where coordinates of $\theta$ were set to 0 so as to limit any advantage gained from $\mathcal{A}_2$. We then sampled $\theta$ with a positive value of $\Delta$ and ran the same experiment. Figure 2 shows ours performs strictly better than a plug-in regression baseline at detecting effects while still maintaining a comparable false positive rate when there is no effect. Details to reproduce the experiments can be found in Appendix H.

**Testing for Warfarin Treatment Effect.** In this section, we apply our method to the problem of testing for treatment effect in warfarin dosages. Warfarin is an anticoagulant drug that is commonly prescribed to patients at risk of blood clots. It is important to get the correct dose of warfarin because incorrect dosages can lead to serious side effects. The Pharmacogenetics and Pharmacogenomics Knowledge Base (PharmGKB) provides a publicly available dataset of patient covariates as well as their final dosages, which might be noisy or slightly suboptimal. In this experiment, the doses were grouped into discrete actions of low, medium, and high based on the original data. We viewed the contextual bandit as giving a reward of $+1$ if the learner selects the correct dose for a patient from $\mathcal{A} = \{\text{low}, \text{medium}, \text{high}\}$ and $+0$ otherwise. We consider the following simplified testing problem. Given a fixed baseline action $a \in \mathcal{A}$, does incorporating another target action $a' \in \mathcal{A}$ such that $a \neq a'$ significantly improve outcomes on average over simply applying $a$ to all patients? Note that this question asks more than just whether $a'$ is better than $a$ if applied uniformly to all patients. This is because $a'$ could be applied to only some patients who would significantly benefit from $a'$ over $a$ while others are still able to receive $a$, thus boosting the overall effectiveness.

To evaluate this, we considered each baseline action in $a \in \mathcal{A}$ and paired it with a target action $a' \in \mathcal{A}$ to evaluate if more can be gained form $a'$. There are 3! such permutations. We computed (approximately) the ground truth improvement $\Delta$ under a linear model (which may be misspecified for this real world data). Note that $\Delta$ will always be non-negative (up to noise), but it may be very small. This dataset happens to induce a disjoint linear bandit; however, we must still deal with the issue of sub-Gaussian features. We ran both a naive linear regression and our algorithm to estimate the effect size, varying the amount of labeled data given 4560 unlabeled samples. The right plot in Figure 2 shows the number of labeled samples required before the effect is detected for a given effect size. Recall that the effect size is determined by the selection of the baseline $a$ and target $a'$. If a method does not detect the effect with the labeled data, it is denoted by ND. We include dashed vertical lines indicating the effect sizes that were evaluated. Ideally, we should see points in the top left corner when there is no effect, and then points converging to the bottom right corner (easy to detect large effect sizes). Our method can detect the effects significantly faster by leveraging the more sensitive test and unlabeled data. We caution that these results are proof-of-concept and should not be misconstrued to suggest that a single dosage of warfarin is adequate for all patients.

## 5 Discussion

In this paper, we studied the problem of estimating the optimal policy value in a linear contextual bandit in the unlearnable regime. We considered this beyond the Gaussian case and presented estimators for both $V^*$ and informative upper bounds on $V^*$. In particular, we showed information-theoretically that a sublinear in $d$ sample complexity of $\widetilde{\mathcal{O}}(2^{C'_\kappa/\epsilon}\sqrt{d}/\epsilon^5)$ is possible for estimating $V^*$ directly, given lower-order moments. To circumvent the exponential dependence, we also gave an efficient $\widetilde{\mathcal{O}}(\sqrt{d}/\epsilon^2)$ algorithm for estimating upper bounds and demonstrated its utility by achieving novel guarantees for model selection and testing. There are several open questions for future work. We conjectured that the exponential dependence on $\epsilon^{-1}$ in Corollary 3.5 is unimprovable in general; however, it remains a difficult open problem to verify this. It would also be interesting to explore how the sample complexity degrades if some quantities were estimated without unlabeled data.

**Acknowledgments**

J.N.L. acknowledges support from the NSF GRFP. A.P. would like to thank the support of the Eric and Wendy Schmidt Center at the Broad Institute of MIT and Harvard. This work was supported in part by funding from the Eric and Wendy Schmidt Center at the Broad Institute of MIT and Harvard. V.M. acknowledges support from the NSF (through CAREER award CCF-2239151 and award IIS-2212182), an Adobe Data Science Research Award, an Amazon Research Award, and a Google Research Collabs Award. This work was also supported by NSF #2112926.

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

# A  Proofs of Results in Section 3.1

## A.1  Formal Analysis of Multi-Armed Bandit Lower Bound

In an effort to prove sublinear sample complexity bounds for $V^*$ estimation in bandit problems, a natural starting point is the classical $K$-armed bandit problem where $r^*(x, a)$ is independent of $x$ (and for this part only we assume that the means are non-zero), equivalently represented as a mean vector $\mu \in \mathbb{R}^K$. The feedback is the same: $Y(a) = \mu_a + \eta(a)$. In this case, $V^*$ is defined as $V^* = \max_a \mu_a$.

One might ask whether it is possible to estimate $V^*$ with better dependence on either $K$ or $\epsilon$ than what is typically required to, for example, identify the best arm or identify an $\epsilon$-optimal arm. The following proposition asserts that such a result is not possible.

**Proposition A.1.** *There exists a class of $K$-armed bandit problems satisfying $\|\mu\|_1 = O(1)$ such that any algorithm that returns an $\epsilon$-optimal estimate of $V^*$ with probability at least $2/3$ must use $\Omega(K/\epsilon^2)$ samples.*

The result is fairly intuitive: since there is no shared information between any of the arms, the learner must essentially sample each arm sufficiently to accurately determine the maximal value.

*Proof.* The proof of the lower bound for the $K$-armed bandit problem follows a standard argument via Le Cam's method. Let $\hat{V}_n$ denote the output of a fixed algorithm $\mathcal{A}$ after $n$ interactions with the bandit that achieves $|\hat{V}_n - V^*| \leq \epsilon$ with probability at least $2/3$. We let $\nu$ and $\nu'$ denote two separate bandit instances, determined by their distributions.

For shorthand, $P_\nu$ and $P_{\nu'}$ denote measures under these instances for the fixed, arbitrary algorithm (and similarly expectations $\mathbb{E}_\nu$ and $\mathbb{E}_{\nu'}$). $N_a$ denotes the (random) number of times the fixed algorithm sampled arm $a$.

We let $\nu$ be distributed as $\mathcal{N}(\mu_a, 1)$ for all $a$ where $\mu = (\epsilon, 0, \ldots, 0)$. Then, define $a' \in \arg\min_{a \neq 1} \mathbb{E}_\nu [T_a]$ and let $\nu'$ be distributed as $\mathcal{N}(\mu'_a, 1)$ where $\mu'_a = \mu_a$ for all $a \neq a'$ and $\mu'_{a'} = 4\epsilon$. We define the successful events $E_\nu = \{\hat{V}_n \in [0, 2\epsilon]\}$ and $E_{\nu'} = \{\hat{V}_n \in [3\epsilon, 5\epsilon]\}$.

By Le Cam's lemma and Pinsker's inequality,

$$P_\nu(E_\nu^c) + P_{\nu'}(E_\nu) \gtrsim 1 - \sqrt{D_{KL}(P_\nu, P_{\nu'})} \tag{6}$$

where $D_{KL}(P_\nu, P_{\nu'}) \lesssim \mathbb{E}_\nu [N_{a'}] \epsilon^2 \leq \frac{n\epsilon^2}{K-1}$ Lattimore & Szepesvári (2018). It then follows that the probability of the successful event is bounded as

$$P_\nu(E_\nu) \leq P_{\nu'}(E_\nu) + C\sqrt{\frac{n\epsilon^2}{K-1}} \tag{7}$$

$$\leq P_{\nu'}(E_{\nu'}^c) + C\sqrt{\frac{n\epsilon^2}{K-1}} \tag{8}$$

$$\leq \frac{1}{3} + C\sqrt{\frac{n\epsilon^2}{K-1}} \tag{9}$$

for some constant $C > 0$. Thus, in order for $P_\nu(E_\nu) \geq 2/3$ it must be that $n \geq \frac{(K-1)}{9C^2\epsilon^2}$. It follows that any algorithm that achieves such a condition must incur sample complexity $\Omega(K/\epsilon^2)$. $\square$

## A.2  Proposition 3.2

*Proof.* **Proof of the first lower bound.** Fix algorithm $\mathcal{A}$ for the linear contextual bandit problem. Then consider the class of $\frac{d}{2}$-armed bandit problem with means vectors satisfying $\|\mu\| = \mathcal{O}(1)$. From this class, we construct the following class of linear contextual bandits. Let $\theta = \begin{bmatrix} \mu \\ -\mu \end{bmatrix} \in \mathbb{R}^d$. The set of contexts is

$\mathcal{X} = \{1, 2\}$ and the feature map is defined as

$$\phi(x, a) = \begin{cases} e_a & x = 1 \\ -e_a & x = 2 \end{cases} \tag{10}$$

where $\{e_1, \ldots, e_d\} \subset \mathbb{R}^d$ denotes the set of standard basis vectors. Then, $X = 1$ and $X = 2$ each with probability $\frac{1}{2}$. This ensures that $\mathbb{E}\phi(X, a) = 0$ for any fixed action $a$. Furthermore, $\|X_a\|_{\psi_2} = \Theta(1)$. Note that $V^* = \mathbb{E} \max_a \langle \phi(X, a), \theta \rangle = \max_a \mu_a$.

Now we construct the reduction by specifying an algorithm $\mathcal{B}$ for the $\frac{d}{2}$-armed bandit. At each round, $\mathcal{B}$ samples $X \sim \text{Unif}\{1, 2\}$ and queries $\mathcal{A}$ for an arm $A$. Upon observing feedback $Y(A) = \mu_A + \eta(A)$, $\mathcal{B}$ feeds $Y(A)$ back to $\mathcal{A}$ if $X = 1$ and $-Y(A)$ if $X = 2$. This process is repeated for $n$ rounds and $\mathcal{A}$ outputs an estimate $\hat{V}_n$, which $\mathcal{B}$ also outputs. If $\mathcal{A}$ outputs $\hat{V}_n$ such that $|\hat{V}_n - V^*| \leq \epsilon$ for any given instance in the linear contextual bandit then $\hat{V}_n$ is also an $\epsilon$-optimal estimate of $\max_a \mu_a$. Therefore, to satisfy $|\hat{V}_n - V^*| \leq \epsilon$, it follows that $n = \Omega(d/\epsilon^2)$.

**Proof of the second lower bound.** Here we prove the second statement of the proposition that even for $K = 2$, it takes $\Omega(d)$ samples to estimate $V^*$ up to small constant additive error $c$. The proof simply follows from the hard instance for signal-noise-ratio (SNR) estimation problem in Theorem 3 of Kong & Valiant (2018).

Let $\mathcal{Q}_n(\mathcal{P})$ be the distribution of $(x_1, y_1, \ldots, x_n, y_n)$ such that $(\theta, \sigma, \Sigma) \sim P$, $x_i \sim N(0, \Sigma)$, $y_i = x_i + \eta_i$, $\eta_i \sim N(0, \sigma^2)$. Let the null distribution $\mathcal{P}_0$ satisfies $\theta = 0, \Sigma = \mathbb{I}_d, \sigma^2 = 1$ almost surely.

We define the alternative data distribution $\mathcal{Q}_n(\mathcal{P}_1)$ as follows. First let a rotation matrix $R$ be drawn from a Haar measure over orthogonal matrices in $\mathbb{R}^{(d+1) \times (d+1)}$. Given $R$, define matrix $M = R \begin{bmatrix} \mathbb{I}_d & 0 \\ 0 & 0 \end{bmatrix} R^\top$. Then we draw $n$ i.i.d samples $z_1, z_2, \ldots, z_n$ from Gaussian distribution $N(0, M)$, and let $x_i = z_{i,1:d}$ be the first $d$ coordinate of $z_i$ and, $y_i = z_{i,d}$ be the last coordinate of $z_i$. This definition will implicitly define a distribution over $\Sigma, \theta, \sigma$ which is called $\mathcal{P}_1$.

Now we show that $d_{TV}(\mathcal{Q}_n(\mathcal{P}_0), \mathcal{Q}_n(\mathcal{P}_1)) \leq 0.27$ when $n \leq d/8$.

**Lemma A.2.** *Define product distribution* $\mathcal{D}_0 = N(0, I_d)^{\otimes n}$, *and* $\mathcal{D}_1 = N(0, M)^{\otimes n}$ *where* $M = R \begin{bmatrix} \mathbb{I}_{d-1} & 0 \\ 0 & 0 \end{bmatrix} R^\top$ *and* $R$ *is drawn from a Haar measure over orthogoal matrices in* $\mathbb{R}^{d \times d}$. *Then* $d_{TV}(\mathcal{D}_0, \mathcal{D}_1) \leq 0.27$ *when* $n \leq d/8$.

*Proof.* Since the covariance $M$ is randomly rotated, it is suffice to compute the total variation distance between the Bartlett decomposition of $\mathcal{D}_0$ and $\mathcal{D}_1$. The Bartlett decomposition $A^{(0)} \in \mathbb{R}^{n \times n}$ of $\mathcal{D}_0$ has

$$A_{i,i}^{(0)} \sim \chi_{d-i+1}^2 \ \forall i \in [n]$$
$$A_{i,j}^{(0)} \sim N(0, 1) \ \forall i < j$$
$$A_{i,j}^{(0)} \sim 0 \ \forall i > j$$

The Bartlett decomposition $A^{(0)} \in \mathbb{R}^{n \times n}$ of $\mathcal{D}_1$ has

$$A_{i,i}^{(1)} \sim \chi_{d-i}^2 \ \forall i \in [n]$$
$$A_{i,j}^{(1)} \sim N(0, 1) \ \forall i < j$$
$$A_{i,j}^{(1)} \sim 0 \ \forall i > j$$

Therefore

$$d_{TV}(\mathcal{D}_0, \mathcal{D}_1) = d_{TV}(\chi_{d-1}^2 \times \ldots \times \chi_{d-n}^2, \chi_d^2 \times \ldots \times \chi_{d-n+1}^2).$$

To establish a total variation distance bound, we will first bound the chi-square divergence

$$\chi^2(\chi_d^2 \times \ldots \times \chi_{d-n+1}^2, \chi_{d-1}^2 \times \ldots \times \chi_{d-n}^2)$$
$$=(\chi^2(\chi_d^2, \chi_{d-1}^2) + 1) \cdot \ldots \cdot (\chi^2(\chi_{d-n+1}^2, \chi_{d-n}^2) + 1) - 1$$
$$=\left(\frac{2^{(d-1)/2}\Gamma((d-1)/2)}{2^{d/2}\Gamma(d/2)}\right)^2 \cdots \left(\frac{2^{(d-n)/2}\Gamma((d-n)/2)}{2^{(d-n+1)/2}\Gamma((d-n+1)/2)}\right)^2 \cdot (d-1)\ldots(d-n) - 1$$
$$=\frac{\Gamma^2((d-n)/2)}{2^n\Gamma^2(d/2)} \cdot (d-1)\ldots(d-n) - 1$$

WLOG, assume that $d$ is a multiple of 4 and that $n$ is a multiple of 2. Then

$$\frac{\Gamma^2((d-n)/2)}{2^n\Gamma^2(d/2)} \cdot (d-1)\ldots(d-n) - 1$$
$$=\frac{(d-1)(d-2)\ldots(d-n)}{(d-2)^2(d-4)^2\ldots(d-n)^2} - 1$$
$$=\frac{(d-1)(d-3)\ldots(d-n+1)}{(d-2)(d-4)\ldots(d-n)} - 1$$
$$=(1+\frac{1}{d-2})(1+\frac{1}{d-4})(1+\frac{1}{d-n}) - 1$$
$$\leq (1+\frac{1}{d/2})^{n/2} - 1 \leq e^{1/4} - 1.$$

Now using the fact that

$$d_{TV}(\mathcal{D}_1, \mathcal{D}_0) \leq \frac{\sqrt{\chi^2(\mathcal{D}_1, \mathcal{D}_0)}}{2}.$$

We get

$$d_{TV}(\mathcal{D}_1, \mathcal{D}_0) \leq \sqrt{e^{1/4} - 1}/2 \leq 0.27$$

$\square$

Under $\mathcal{P}_1$, $y$ is a linear function of $x$ almost surely, and the variance $y$ is $1 - R_{d+1,d+1}^2$. Since $R_{d+1}$ is an entry of a unit norm random vector, for sufficiently large $d$, it holds that $1 - R_{d+1,d+1}^2 \geq 0.99$ with probability 0.99.

We construct the alternative bandit instance using $(\theta, \sigma, \Sigma) \sim \mathcal{P}_1$, and for each arm $a \in [2]$, define $\phi(X, a) \sim N(0, \Sigma), Y(a) = \theta^\top \phi(X, a)$. It is easy to see that in this case, $\mathbb{E}V^* = \Omega(1)$. The other bandit instance is a simple "pure noise" example where $\phi(X, a) \sim N(0, \mathbb{I}_d), Y(a) \sim N(0, 1)$, and clearly $\mathbb{E}V^* = 0$ since $\theta = 0$. For any bandit algorithm for estimating $V^*$, after $d/16$ rounds, even if all the rewards (regardless of which arm gets pulled) are shown to the algorithm the total variation distance is between the two example is still bounded by $1/3$ through Lemma A.2. Therefore, we conclude any bandit algorithm must incur $\Omega(1)$ error for estimating $V^*$ with probability at least $2/3$ when $n = c \cdot d$ where $c = 1/16$.

$\square$

## B    Proofs of Results in Section 3.2

### B.1    Proof Mechanism

We give a brief sketch of the proof of Theorem 1. As mentioned, we reduce the problem to just estimating $\mathbb{E}_X\left[p_t(\{\langle\theta, \phi(X, a)\rangle\}_{a\in\mathcal{A}})\right]$, which gives us a $\zeta$-accurate approximation of $V^*$ via Definition 3.3. Note that $\mathbb{E}_X\left[p_t(\{\langle\theta, \phi(X, a)\rangle\}_{a\in\mathcal{A}})\right] = \sum_{|\alpha|\leq t} c_\alpha \prod_a \mathbb{E}_X \langle\phi(X, a), \theta\rangle^{\alpha_a}$ is just a linear combination of all the $\alpha$-moments $S_\alpha := \mathbb{E}_X \prod_a \langle\phi(X, a), \theta\rangle_a^\alpha$. Recall that $m$ is the sample size of the split dataset. In Lemma B.2 we show that $\hat{S}_{m,\alpha}^k$ in Algorithm 1 is an unbiased estimator of $S_\alpha$. We then show that the variance is bounded.

**Lemma B.1.** *There exist constants $C > 0$ so that $\mathrm{var}(\hat{S}_{m,\alpha}^k) \leq \sum_{u=1}^s (Cs^3 m^{-1} \sqrt{d})^u$ where $s = |\alpha|$.*

The proof of this variance bound is non-trivial and, as one might guess, Lemma B.1 does most of the heavy-lifting in the proof Theorem 1. Observe that this step is where the $\sqrt{d}/n$ rate is achieved (since $n \approx m$ up to logarithmic factors). To achieve this, we must be careful not to leak $\sqrt{d}$ factors into the variance bound through, for example, hasty applications of Cauchy-Schwarz or sub-Gaussian bounds. This is achieved through a Hanson-Wright-like inequality (Lemma B.3). Also, to prevent any exponential blow-ups in the degree of the polynomial, we must ensure that any exponential terms in the numerator can be modulated by $m^u$ in the denominator. This comes down to a careful counting argument. The last steps of the proof revolve around using the variance bound to prove concentration.

### B.2 Understanding the influence of polynomial approximation

Theorem 1 and Lemma B.1 provide some interesting observations about the influence of the polynomial approximation. One of the key terms to understand is the summation. Each term looks like $\left( \frac{Ct^3 \sqrt{d}}{n} \log(1/\delta) \right)^{s/2}$ for each $s \in [t]$. This means that as long as $n > Ct^3 \sqrt{d} \log(1/\delta)$, this term is less than one and thus we have $|V^* - \hat{S}_n| \leq \mathcal{O}\left( \zeta + c_{\max} t^2 (et/K + e)^K \sqrt{\frac{Ct^3 \sqrt{d} \log(1/\delta)}{n}} \right)$ Let us take $K = O(1)$ for ease of exposition. Recall that the first term ($\zeta$) is the approximation error of the polynomial while the second term can be interpreted as the effect of variance. The estimation error can be made $\frac{\epsilon}{2}$ small by by taking

$$ n = \mathcal{O}\left( c_{\max}^2 \mathrm{poly}(t) \frac{\sqrt{d} \log(1/\delta)}{\epsilon^2} \right). $$

The approximation error can be made $\zeta \approx \frac{\epsilon}{2}$ small by choosing stronger polynomial approximators. However, this is where the trade-off occurs. To obtain better polynomial approximators, we typically require that the degree $t$ increases accordingly. The magnitude of the coefficients $c_{\max}$ typically also increase (for very general polynomial approximators). This means that there is a chance the sample complexity can increase significantly if $t$ and $c_{\max}$ are functions of the accuracy $\epsilon$. While this does not affect the dependence on $\sqrt{d}$ (and thus the results remain sublinear), it can lead to worse $\epsilon$-dependence. See Corollaries 3.5 and 3.7 for instantiations.

### B.3 Proof of Theorem 1

First note that the data-whitening pre-processing step does not contribute more than a constant to the sample complexity since we reframe the problem as

$$ r^*(x, a) = \langle \phi(x, a), \theta \rangle = \left\langle \Sigma^{-1/2} \phi(x, a), \Sigma^{1/2} \theta \right\rangle $$

Observe $\|\Sigma^{1/2}\theta\| \lesssim \|\theta\|$ since $\phi(X, a) \sim \mathrm{subG}(\tau^2)$ with $\tau = \mathcal{O}(1)$. Also $\Sigma^{-1/2}\phi(X, a) \sim \mathrm{subG}(\tau^2/\rho)$ where $\tau^2/\rho = \mathcal{O}(1)$. Note that this means we must also linearly transform $\Sigma_a$ and $\Sigma_{a,a'}$ for $a, a' \in \mathcal{A}$, but an identical calculation shows that they are similarly unaffected up to constants. Therefore, without loss of generality, we may simply take $\Sigma = \mathbb{I}_d$.

Next, we verify that $\hat{S}_{m,\alpha}^k$ for $k = 1, \ldots, \lceil 48 \log(1/\delta) \rceil$ are unbiased estimators of the moments of interest.

**Lemma B.2.** *Given $\hat{S}_{m,\alpha}^k$ defined in (2), it holds that $\mathbb{E}_{D^k}\left[ \hat{S}_{m,\alpha}^k \right] = \mathbb{E}_X \prod_{a \in [K]} \langle \theta, \phi(X, a) \rangle^{\alpha_a}$.*

*Proof.* We drop the superscript $k$ notation denoting which of the datasets is being used as the argument is identical. Fix $\ell \in \binom{[m]}{s}$ as an $s$-combination of the indices $[n]$. Since the data in $D$ is i.i.d, we have that

$$\mathbb{E}_D \mathbb{E}_X \prod_{j \in [s]} \left\langle y_{\ell_j} x_{\ell_j}, \phi(X, a_{(j)}) \right\rangle = \mathbb{E}_X \prod_{j \in [s]} \left\langle \mathbb{E}_D \left[ y_{\ell_j} x_{\ell_j} \right], \phi(X, a_{(j)}) \right\rangle \tag{11}$$

$$= \mathbb{E}_X \prod_{j \in [s]} \left\langle \theta, \phi(X, a_{(j)}) \right\rangle \tag{12}$$

$$= \mathbb{E}_X \prod_{a \in [K]} \left\langle \theta, \phi(X, a) \right\rangle^{\alpha_a} \tag{13}$$

where the second equality uses the fact that $\mathbb{E}_D x_i x_i^\top = \mathbb{I}_d$. for all $i \in [m]$. $\qquad\square$

Next, we establish a bound on the variance in preparation to apply Chebyshev's inequality.

**Lemma B.1.** *There exist constants $C > 0$ so that $\mathrm{var}(\hat{S}_{m,\alpha}^k) \leq \sum_{u=1}^s (Cs^3 m^{-1} \sqrt{d})^u$ where $s = |\alpha|$.*

*Proof.* As before, we will drop the superscript $k$ notation as the argument is identical for each independent estimator. Let $s = |\alpha|$.

By definition, the variance is given by

$$\mathrm{var}_D \left( \hat{S}_{m,\alpha} \right) = \mathbb{E}_D \left[ \hat{S}_{m,\alpha}^2 \right] - \mathbb{E}_D \left[ \hat{S}_{m,\alpha} \right]^2 \tag{14}$$

where

$$\hat{S}_{m,\alpha}^2 = \frac{1}{\binom{m}{s}^2} \sum_{\ell,\ell'} \mathbb{E}_{X,X'} \prod_{i \in [s]} \left\langle y_{\ell_i} x_{\ell_i}, \phi(X, a_{(j)}) \right\rangle \cdot \prod_{i \in [s]} \left\langle y_{\ell_i'} x_{\ell_i'}, \phi(X', a_{(i)}) \right\rangle \tag{15}$$

$$\mathbb{E}_D \left[ \hat{S}_{n,\alpha} \right]^2 = \mathbb{E}_{X,X'} \prod_a \left\langle \theta, \phi(X, a) \right\rangle^{\alpha_a} \cdot \prod_a \left\langle \theta, \phi(X', a) \right\rangle^{\alpha_a} \tag{16}$$

where again $\ell$ and $\ell'$ are $s$-combinations $[n]$. Similar to Kong & Valiant (2018), we can analyze the variance as individual terms in the sum over $\ell$ and $\ell'$:

$$\mathbb{E}_D \mathbb{E}_{X,X'} \prod_{i \in [s]} \left\langle y_{\ell_i} x_{\ell_i}, \phi(X, a_{(i)}) \right\rangle \cdot \prod_{j \in [s]} \left\langle y_{\ell_i'} x_{\ell_i'}, \phi(X', a_{(i)}) \right\rangle \tag{17}$$

$$- \mathbb{E}_{X,X'} \prod_a \left\langle \theta, \phi(X, a) \right\rangle^{\alpha_a} \cdot \prod_a \left\langle \theta, \phi(X', a) \right\rangle^{\alpha_a} \tag{18}$$

There are two important cases to consider: (1) when $\ell$ and $\ell'$ do not share any indices and (2) when there is partial or complete overlap of indices.

1. **No intersection of $\ell$ and $\ell'$** In this case, we may see that there is no contribution to the variance for this term due to independence:

$$\mathbb{E}_D \mathbb{E}_{X,X'} \prod_{i \in [s]} \left\langle y_{\ell_i} x_{\ell_i}, \phi(X, a_{(i)}) \cdot \prod_{i \in [s]} \left\langle y_{\ell_i'} x_{\ell_i'}, \phi(X', a_{(i)}) \right\rangle \right. \tag{19}$$

$$= \mathbb{E}_{X,X'} \prod_{i \in [s]} \left\langle \theta, \phi(X, a_{(i)}) \right\rangle \cdot \prod_{i \in [s]} \left\langle \theta, \phi(X', a_{(i)}) \right\rangle \tag{20}$$

$$= \mathbb{E}_{X,X'} \prod_a \left\langle \theta, \phi(X, a) \right\rangle^{\alpha_a} \cdot \prod_a \left\langle \theta, \phi(X', a) \right\rangle^{\alpha_a} \tag{21}$$

This term simply cancels with $-\mathbb{E} \left[ \hat{S}_{m,\alpha} \right]^2$.

2. **Partial or complete intersection of $\ell$ and $\ell'$**

In this case, there are some samples that appear twice. Let $\beta = \{(i,j) \;:\; \ell_i = \ell'_j\}$ be the set of indices that refer to the same sample in $D$. Also define $\gamma, \gamma' \subset [s]$ as the subsets of indices of $\ell$ and $\ell'$ respectively that are not shared.

The left-hand side of this term can be then be written as

$$\mathbb{E}_D \mathbb{E}_{X,X'} \prod_{i \in [s]} \left\langle y_{\ell_i} \phi_{\ell_i}, \phi(X, a_{(j)}) \right\rangle \cdot \prod_{j \in [s]} \left\langle y_{\ell'_j} \phi_{\ell'_i}, \phi(X', a_{(i)}) \right\rangle \tag{22}$$

$$= \mathbb{E}_{X,X'} \prod_{(i,i') \in \beta} \mathbb{E}_{D_n} \left[ y_{\ell_i}^2 \left\langle \phi_{\ell_i}, \phi(X, a_{(i)}) \right\rangle \left\langle \phi_{\ell_i}, \phi(X', a_{(i')}) \right\rangle \right] \tag{23}$$

$$\times \prod_{i \in \gamma} \left\langle \theta, \phi(X, a_{(i)}) \right\rangle \prod_{i' \in \gamma'} \left\langle \theta, \phi(X', a_{(i')}) \right\rangle \tag{24}$$

To proceed, we require the following critical lemma which bounds separate moments in the factors that come from shared indices. The proof given in Section G.

**Lemma B.3.** *Let $p \geq 1$ be an integer and define $M = \mathbb{E}_{D_n} \left[ y_{\ell_i}^2 \phi_{\ell_i} \phi_{\ell_i}^\top \right]$. There is a constant $C$ such that*

$$\left( \mathbb{E}_{X,X'} |\phi(X, a_{(i)})^\top M \phi(X', a_{(i)})|^p \right)^{1/p} \leq C \cdot p\tau^2 (\sigma^2 + L\|\theta\|^2) \sqrt{d} \tag{25}$$

Through standard sub-Gaussian arguments, we also have that, for $p \geq 1$, it holds that $\left( \mathbb{E}| \left\langle \theta, \phi(X, a_{(i)}) \right\rangle |^p \right)^{1/p} \leq C\tau \|\theta\| \sqrt{p}$ for some constant $C > 0$. And the same holds for the $X'$ factors.

For convenience, let $\gamma = (\sigma^2 + L\|\theta\|^2)$ and let $u = |\beta| \leq s$ be the size of the overlap. By the generalized Holder inequality, the term in (23) is upper bounded by

$$\left( \prod_{(i,i') \in \beta} \mathbb{E}_{X,X'} |\phi(X, a_{(i)})^\top M \phi(X', a_{(i)})|^{2s} \prod_{i \in \gamma} \mathbb{E}_{X,X'} | \left\langle \theta, \phi(X, a_{(i)}) \right\rangle |^{2s} \right)^{1/2s} \tag{26}$$

$$\times \left( \prod_{i' \in \gamma'} \mathbb{E}_{X,X'} | \left\langle \theta, \phi(X', a_{(i')}) \right\rangle |^{2s} \right)^{1/2s} \tag{27}$$

$$\leq \left( \prod_{(i,i') \in \beta} \mathbb{E}_{X,X'} |\phi(X, a_{(i)})^\top M \phi(X', a_{(i)})|^{2s} \right)^{1/2s} \tag{28}$$

$$\leq (C_0 \cdot (2s)\tau^2 \gamma \sqrt{d})^u \tag{29}$$

$$= C_1^u s^u \tau^{2u} \gamma^u d^{u/2} \tag{30}$$

where we have used Assumption 3 and $C_0, C_1 > 0$ are constants.

In summary, we have shown that there is no contribution to the variance when no indices are shared between $\ell$ and $\ell'$ and the contribution to the variance when $m$ indices are shared is bounded by $\widetilde{\mathcal{O}}(d^{u/2})$. It suffices now to count the terms to see the total contribution for each $u = 1, \ldots, s$.

It can be checked that the number of terms where the size of the intersection $u = |\beta|$ is

$$\binom{m}{s} \binom{s}{u} \binom{m-s}{s-u} \tag{31}$$

since there are $s$ elements $\ell$, $u$ of which may have an intersection, and a remaining $s - u$ elements to be chosen for $\ell'$ that are not shared with $\ell$ (recall that $m \geq 2s$). Counting all of these contributions up, this

implies that the variance can be bounded as

$$\operatorname*{var}_{D}\left(\hat{S}_{m,\alpha}\right) \leq \frac{1}{\binom{m}{s}^2} \sum_{u=1}^{s} \binom{m}{s}\binom{s}{u}\binom{m-s}{s-u} C_1^u s^u \tau^{2u} \gamma^u d^{u/2} \tag{32}$$

$$\leq \sum_{u=1}^{s} \frac{\binom{s}{u}\binom{m-s}{s-u}}{\binom{m}{s}} C_1^u s^u \tau^{2u} \gamma^u d^{u/2} \tag{33}$$

Now, will bound the factor involving binomial coefficients. First note that we have

$$\binom{s}{u} \leq \frac{s^u}{u!} \qquad \text{and} \qquad \binom{m-s}{s-u} \leq \frac{(m-s)^{s-u}}{(s-u)!} \qquad \text{and} \qquad \binom{m}{s} \geq \frac{(m-s+1)^s}{s!} \tag{34}$$

Therefore,

$$\binom{m}{s}\binom{s}{u}\binom{m-s}{s-u} \leq \frac{s^u(m-s)^{s-u}s!}{u!(s-u)!(m-s+1)^s} \tag{35}$$

$$\leq \binom{s}{u}\frac{s^u}{(m-s)^u} \tag{36}$$

$$\leq \binom{s}{u}\frac{(2s)^u}{m^u} \tag{37}$$

$$\leq \left(\frac{2es^2}{m}\right)^u \tag{38}$$

where in the third line, we have used the fact that $m = n/48\log(1/\delta) \geq 2t \geq 2s$. Then, we can conclude that the variance is bounded as

$$\operatorname*{var}_{D}\left(\hat{S}_{m,\alpha}\right) \leq \sum_{u=1}^{s}\left(C_2 s^3 \tau^2 \gamma d^{1/2}\right)^u \tag{39}$$

where $C_2 > 0$ is a constant. Since it was assumed that $\tau$, $\sigma^2$, $L$ and $\|\theta\|$ are $\mathcal{O}(1)$, the final claim follows. $\square$

The error bound result on the median of the estimators follows almost immediately.

**Theorem 5.** *There exists a constant $C = \mathcal{O}(1)$ for all $k$ such that, with probability at least $2/3$,*

$$|\hat{S}_{m,\alpha}^k - \mathbb{E}\hat{S}_{m,\alpha}^k| \leq \epsilon(m,d,s) \tag{40}$$

*where*

$$\epsilon(m,d,s) := \sum_{u=1}^{s}\left(\frac{Cs^3\sqrt{d}}{m}\right)^{u/2} \tag{41}$$

*Furthermore, defining $\hat{S}_{n,\alpha} = \operatorname{median}\{\hat{S}_{m,\alpha}^k\}_{k=1}^q$, with probability $1 - \delta$,*

$$|\hat{S}_{n,\alpha} - \mathbb{E}\hat{S}_{n,\alpha}^k| \leq \epsilon(m,d,s) \tag{42}$$

*Proof.* The first statement follows immediately from Chebyshev's inequality and the second applies the median of means trick for the independent estimators $\{\hat{S}_{m,\alpha}^k\}_{k=1}^q$ given the choice of $q$ Kong et al. (2020) $\square$

### B.3.1 Final Bound

We now combine the estimation and approximation error bounds to derive the final result, which is reproduced here.

**Theorem 1.** *Let Assumptions 1, 2, and 3 hold. Let $p_t$ be a $t$-degree $(\zeta, c_{\max})$-polynomial approximator and let $\hat{S}_n$ be the output of Algorithm 1. Suppose that $n \geq 96 \log(1/\delta)t$. There is a constant $C > 0$, depending only on $\tau$, $\sigma$, and $L$, such that with probability at least $1 - t(et/K + e)^K \delta$, $|V^* - \hat{S}_n|$ is bounded by*

$$\zeta + c_{\max} t \left(et/K + e\right)^K \sum_{s=1}^{t} \left(\frac{Ct^3\sqrt{d}}{n} \log(1/\delta)\right)^{s/2}$$

*Proof.* We first start by bounding the full estimation error $|\mathbb{E}_X \left[p_t(\{\langle \theta, \phi(X, a) \rangle\}_{a \in \mathcal{A}})\right] - \hat{S}_n|$. For convenience, let us denote $S = \mathbb{E}_X \left[p_t(\{\langle \theta, \phi(X, a) \rangle\}_{a \in \mathcal{A}})\right]$. The degree 0 and degree 1 moments are already known exactly; thus we may consider $2 \leq s \leq t$. By the union bound and triangle inequality combined with the result of Theorem 5, with probability at least $1 - t(et/K + e)^K \delta$,

$$|S - \hat{S}_n| \leq \sum_{\substack{s \in [2,t] \\ \alpha \; : \; |\alpha| = s}} c_\alpha |\hat{S}_{n,\alpha} - \mathbb{E}\hat{S}_{n,\alpha}^k| \tag{43}$$

$$\leq \sum_{\substack{s \in [2,t] \\ \alpha \; : \; |\alpha| = s}} c_{\max} \epsilon(n/q, d, s) \tag{44}$$

For each $s$, there are $\binom{s+K-1}{K-1} \leq (es/K + e)^K$ monomials for possible choices of $\alpha$. Therefore, the good event implies that

$$|S - \hat{S}_n| \leq t \left(et/K + e\right)^K c_{\max} \cdot \epsilon(n/q, d, t) \tag{45}$$

Next, we may apply the approximation error. By the triangle inequality

$$|\mathbb{E}\max_a \langle \theta, \phi(X, a) \rangle - \hat{S}_n| \leq \zeta + t \left(et/K + e\right)^K c_{\max} \cdot \epsilon(n/q, d, t) \tag{46}$$

$\square$

## B.4 Generic Polynomial Approximator in Example 3.4

Throughout this subsection only, we will use $p_t$ to refer to $p_t^{\mathrm{bbl}}$.

**Lemma B.4.** *Let $f : [-1, 1]^K \to \mathbb{R}$ be defined as $f(z) = \max_a z_a$. There exists a degree-$t$ polynomial $p_t : [-1, 1]^K \to \mathbb{R}$ of the form in Definition 3.3 such that*

$$\sup_{z \in [0,1]^K} |f(z_1, \ldots, z_K) - p_t(z_1, \ldots, z_K)| \leq \frac{C_K}{t} \tag{47}$$

*for some constant $C_K$ that only depends on $K$. Furthermore, for $t \geq K$, $|c_\alpha| \leq \frac{(2et)^{2K+1}2^{3t}}{K^K} =: c_{\max}$ for all $\alpha$ such that $|\alpha| \leq t$.*

*Proof.* It follows from Lemma 2 of Tian et al. (2017) that, for any 1-Lipschitz $g$ supported on $[0, 1]^K$, a polynomial $q(\hat{z}) = \sum_{\alpha \; : \; |\alpha| \leq t} \hat{c}_\alpha \prod_{u \in \alpha} \hat{z}^u$ exists satisfying (47) with $|c_\alpha| \leq (2t)^K 2^t := \hat{c}_{\max}$ and constant $\frac{C_K}{2}$. The max function $g$ is 1-Lipschitz and thus satisfies this condition. Let $g(\hat{z}) = \max_a \hat{z}_a$ and $\hat{z}_a = \frac{z_a+1}{2}$. Note that $\hat{z} \in [0, 1]^K$ by this definition and $f(z) = 2g(\hat{z}) - 1$. Furthermore $p(z) = 2q(\hat{z}) - 1$ degree $t$ polynomial such that $p(z) = \sum_{|\alpha| \leq t} c_\alpha \prod_{u \in \alpha} z^u$. Therefore, for any $z$, $|f(z) - p(z)| \leq C_K/t$.

The coefficients $c_\alpha$ are different from $\hat{c}_\alpha$ as a result of the change of variables. Note that there are $\sum_{s=0}^{t} \binom{s+K-1}{K-1} \leq (t+1)(et/K + e)^K \leq 2t(2et/K)^K$ terms. Therefore $|c_\alpha| \leq \frac{(2et)^{2K+1}2^{3t}}{K^K}$. The value of $C_K$ is given in equation (13) of Bagby et al. (2002). $\square$

**Corollary 3.5.** *The estimator $\hat{S}_n$ generated by Algorithm 1 with polynomial $p_t^{BBL}$ satisfies $|V^* - \hat{S}_n| \leq \epsilon$ for $\epsilon < 1$ with probability at least $1 - \delta$, $t = 2C_K/\epsilon$, and sample complexity*

$$\mathcal{O}\left(\left(\frac{C_K}{K\epsilon}\right)^K 2^{C_K/\epsilon} \cdot \frac{K\sqrt{d}}{\epsilon^5} \cdot \log\left(\frac{C_K}{\epsilon\delta}\right)\right). \tag{3}$$

*Proof.* To ensure that each term in the sum of Theorem 1 is at most $\frac{\epsilon}{2t}$, it suffices to take

$$n = c_{\max}^2 t^4 (et/K + e)^{2K} \cdot \frac{Ct^3 \sqrt{d} \log(1/\delta)}{\epsilon^2} \tag{48}$$

for some constant $C > 0$. Then, choose $t = \max\{2C_K/\epsilon, K\}$. Therefore,

$$n = \mathcal{O}\left(C_K c_{\max}^2 (4eC_K/\epsilon)^{2K} \cdot \frac{\sqrt{d} \log(1/\delta)}{\epsilon^5}\right) \tag{49}$$

From the definition of $c_{\max}$, this leads to

$$\mathcal{O}\left(\frac{(4eC_K/\epsilon)^{4K+2} 2^{12C_K/\epsilon}}{K^{2K}} \cdot C_K (4eC_K/\epsilon)^{2K} \cdot \frac{\sqrt{d} \log(1/\delta)}{\epsilon^5}\right)$$

To ensure this event occurs with probability at least $1 - \delta'$ we apply a change of variables with

$$\delta' = t(et/K + e)^K \delta = \frac{2C_K (2eC_K/\epsilon K + e)^K}{\epsilon}$$

Therefore the total sample complexity is

$$\mathcal{O}\left(\frac{(4eC_K/\epsilon)^{4K+2} 2^{12C_K/\epsilon}}{K^{2K}} \cdot C_K (4eC_K/\epsilon)^{2K} \cdot \frac{K\sqrt{d}}{\epsilon^5} \cdot \log\left(\frac{2C_K (2eC_K/\epsilon K + e)}{\epsilon \delta'}\right)\right)$$

with probability at least $1 - \delta'$. □

## B.5 Binary Polynomial Approximator in Example 3.6

Throughout this subsection only, we will use $p_K$ to refer to $p_K^{\text{bin}}$.

**Lemma B.5.** *There exists a polynomial $p_K^{bin}$ satisfying the conditions of Example 3.6.*

*Proof.* From the description of the CB problem, we have that

$$V^* = \mathbb{E}_X \max\left\{\langle \phi(X, 1), \theta \rangle \ldots, \langle \phi(X, K), \theta \rangle\right\} \tag{50}$$

where

$$\max\left\{\langle \phi(X, 1), \theta \rangle \ldots, \langle \phi(X, K), \theta \rangle\right\} = \begin{cases} \omega & \exists a \in [K] \text{ s.t. } \langle \phi(X, a), \theta/\omega \rangle = 1 \\ 0 & \text{otherwise} \end{cases}$$

$$= |\omega| \left(1 - \prod_a (1 - \langle \phi(X, a), \theta/|\omega| \rangle)\right)$$

Note that the right side is simply a $K$-degree polynomial function of $\{\langle \phi(x, a), \theta \rangle\}$ which we denote by $p_K$ so there is zero approximation error. Furthermore, it can be easily seen that a bound on the largest coefficient is $|\omega|^{-K}$ due to the product of terms. □

We now prove the corollary.

**Corollary 3.7.** *On the class of problems in Example 3.6, $\hat{S}_n$ with $p_K^{bin}$ satisfies $|V^* - \hat{S}_n| \leq \epsilon$ for $\epsilon < 1$ with probability at least $1 - \delta$ and sample complexity $\mathcal{O}\left(K^8 2^{2K} \cdot \frac{\sqrt{d}}{\epsilon^2} \cdot \log\left(\frac{2K}{\delta}\right)\right)$.*

*Proof.* The proof is an essentially identical application of Theorem 1 as in the previous corollary. In this case, we have $t = K$ as the degree. Then, it suffices to choose

$$n = c_{\max}^2 t^4 (et/K + e)^{2K} \cdot \frac{Ct^3 \sqrt{d} \log(1/\delta)}{\epsilon^2}$$

to ensure the error of each term is at most $\epsilon/t$. Then, with $c_{\max}^2 \leq |\omega|^{-K}$ and $t = K$ and the change of variables to $\delta' = t(et/K + e)^K \delta$, we get that the total sample complexity is

$$\mathcal{O}\left(|\omega|^{-2K} K^8 2^{2K} \cdot \frac{\sqrt{d}}{\epsilon^2} \cdot \log\left(\frac{2K}{\delta'}\right)\right)$$

with probability at least $1 - \delta'$ $\qquad\square$

## B.6 Definition of $C_K$

Bagby et al. (2002) define $C_K$ via the restriction of a holomorphic function to $\mathbb{R}^K$, given by $g : \mathbb{R}^K \to \mathbb{R}$. In particular, their Lemma A defines the quantity

$$I_k = \frac{K^k}{k!} \int |w|^k (|w| + 1) |g(w)| dw$$

for the restriction function $g$. In our case, we have $k = 1$ (due to Lipschitzness). It is sufficient in their Theorem 1 to ensure that $\frac{I_1}{t}$, and thus $\frac{C_K}{t}$, upper bounds the following quantity:

$$CR(tR + 1)(2Rt)^K e^{\sqrt{K}(2R+1)^{\delta t}} 2^{-t}$$

where $C$ is an absolute constant (see their Lemma A), $R$ is the $l_\infty$ length of the box (in this case $R = 1$), and $\delta > 0$ is such that $\sqrt{K}(2R + 1)\delta \leq \log 2$, and we recall that $t$ is the degree of the polynomial approximator,

# C Proofs of Results in Section 4

## C.1 Proof of Theorem 2

Perhaps surprisingly, The result makes use of a combination of Talagrand's comparison inequality (which arises from Talagrand's fundamental "generic chaining" approach in empirical process theory Talagrand (2006)) and some techniques from Kong et al. (2020). Here, we state a version of Talagrand's comparison inequality that appears in Vershynin (2018).

**Lemma C.1.** *Let $(W_a)_{a \in [K]}$ be a mean zero sub-Gaussian process and $(Z_a)_{a \in [K]}$ a mean zero Gaussian process satisfying $\|W_a - W_{a'}\|_{\psi_2} \lesssim \|Z_a - Z_{a'}\|_{L^2}$. Then,*

$$\mathbb{E} \max_{a \in [K]} W_a \lesssim \mathbb{E} \max_{a \in [K]} Z_a \tag{51}$$

By Assumption 4, note that

$$\| \langle \phi(X, a) - \phi(X, a'), \theta \rangle \|_{\psi_2}^2 \leq L_0^2 \| \langle \phi(X, a) - \phi(X, a'), \theta \rangle \|_{L^2}^2 \tag{52}$$

Thus, we can define a Gaussian process $Z \sim \mathcal{N}(0, \Lambda)$ that satisfies the condition in Talagrand's inequality by choosing its mean to be zero and its covariance matrix to match the increment of the original sub-Gaussian process $\phi(X, \cdot)$. Note that such a process trivially exists since we can let $\Lambda$ satisfy:

$$\Lambda_{a,a'} = \mathrm{cov}(Z_a, Z_{a'}) = \mathbb{E}\left[ \langle \phi(X, a), \theta \rangle \langle \phi(X, a'), \theta \rangle \right] \tag{53}$$

Then, the first inequality in the theorem is satisfied with $U = \mathbb{E} \max_{a \in [K]} Z_a$. The proof of the second inequality is deferred to Section C.1.1.

Since $\theta$ is unknown, our goal now is to estimate the increment $\| \langle \phi(X, a) - \phi(X, a'), \theta \rangle \|_{L^2}^2$ from samples. Specifically, we aim to estimate the following quantity:

- For all $a, a' \in [K]$ such that $a \neq a'$, $\beta_{a,a'} := \mathbb{E}\left[\langle \phi(X,a) - \phi(X,a'), \theta \rangle^2\right] = \theta^\top \Sigma_{a,a'} \theta$ where $\Sigma_{a,a'} = \mathbb{E}\left[(\phi(X,a) - \phi(X,a')(\phi(X,a) - \phi(X,a'))^\top\right]$.

We can construct fast estimators for these quantities using similar techniques as those developed in Kong & Valiant (2018). While a similar final result is obtained in that paper by Chebyshev's inequality and counting, here we present a version that is carried out with a couple simple applications of Bernstein's inequality. Algorithm 2 specifies the form of the estimator and the data collection procedure.

**Lemma C.2.** *Fix $a, a' \in [K]$ such that $a \neq a'$ and define $\xi^2 = \tau^2(\tau^2 \|\theta\|^2 + \sigma^2)$. Let $\delta \leq 1/e$. Given the dataset $D_n = \{x_i, a_i, y_i\}$, with probability at least $1 - 3\delta$,*

$$|\hat{\beta}_{a,a'} - \beta_{a,a'}| \leq \sqrt{\frac{\xi^2 \|\Sigma\|^2 \|\theta\|^2}{C_1 m}} \cdot \log(2/\delta) + \sqrt{\frac{\xi^4 \|\Sigma\|^2 d}{C_2 m^2}} \cdot \log^2(2d/\delta) \tag{54}$$

*for absolute constants $C_1, C_2 > 0$.*

Something to note about this result is that it depends on $\|\theta\|$. This is fairly common in sample complexity results and bandits where the size of $\theta$ is akin to the scale of the problem. However, it is problem-dependent. Interestingly, the applications of Section 4.1 make use of the special case when $\theta = 0$.

*Proof.* Consider an arbitrary pair $a, a'$ and covariance matrix $\Sigma_{a,a'}$. For convenience, we drop the subscript notation and just write $\Sigma$. The argument will be the same for all pairs, including when $a = a'$. The dataset $D_n$ is split into two independent datasets $D_m$ and $D'_m$ of size $m = \frac{n}{2}$. Let $\phi_i := \phi(x_i, a_i)$ as shorthand and the same for $\phi'_i$.

First, we verify that $\hat{\beta}_{a,a'}$ is indeed an unbiased estimator of $\beta_{a,a'}$:

$$\mathbb{E}\left[\hat{\theta}^\top \Sigma \hat{\theta}'\right] = \mathbb{E}\left[y_i y'_j \phi_i^\top \Sigma \phi'_i\right] = \theta^\top \Sigma \theta \tag{55}$$

which follows by independence of the datasets $D_m$ and $D'_m$ and the fact that the covariance matrix under the uniform data collection policy is the identity. By adding and subtracting and then applying the triangle inequality, we have

$$|\hat{\theta}^\top \Sigma \hat{\theta}' - \theta^\top \Sigma \theta| = \underbrace{|\theta^\top \Sigma \hat{\theta}' - \theta^\top \Sigma \theta|}_{\text{Term I}} + \underbrace{|\hat{\theta}^\top \Sigma \hat{\theta}' - \theta^\top \Sigma \hat{\theta}'|}_{\text{Term II}} \tag{56}$$

and we focus on bounding each term individually. We start with the first. Note that $\|\theta^\top \Sigma \phi'_i\|_{\psi_2} \leq \|\Sigma \theta\| \tau$ and $\|y'_i\|_{\psi_2} \lesssim \sqrt{\tau^2 \|\theta\|^2 + \sigma^2}$. Therefore, we have that the term $\phi'_{i,k} y'_i$ is sub-exponential with parameter $\|\phi'_{i,k} y'_i\|_{\psi_1} \lesssim \tau \|\Sigma \theta\| \sqrt{\tau^2 \|\theta\|^2 + \sigma^2} = \xi \|\Sigma \theta\|$, where recall that we have defined $\xi^2 = \tau^2(\tau^2 \|\theta\|^2 + \sigma^2)$. Then, by Bernstein's inequality,

$$\Pr\left(\frac{1}{n} \sum_{i \in [n]} \theta^\top \Sigma \phi'_{i,k} y'_i - \theta^\top \Sigma \theta \geq t\right) \leq \exp\left(-C \min\left\{\frac{nt^2}{\|\Sigma \theta\|^2 \xi^2}, \frac{nt}{\|\Sigma \theta\| \xi}\right\}\right) \tag{57}$$

for some absolute constant $C > 0$, and the negative event occurs with the same upper bound on the probability. This implies

$$|\frac{1}{m} \sum_{i \in [m]} \theta^\top \Sigma \phi'_i y'_i - \theta^\top \Sigma \theta| \leq \sqrt{\frac{\xi^2 \|\Sigma \theta\|^2}{Cm}} \cdot \log(2/\delta) \tag{58}$$

For the second term, we condition on the data in $D'$ and then apply the same calculations. The difference is that $\|\phi_i \Sigma \hat{\theta}'\|_{\psi_2} \leq \tau \|\Sigma \hat{\theta}'\|$ and so the bound becomes

$$|\frac{1}{m} \sum_{i \in [m]} y_i \phi_i^\top \Sigma \hat{\theta}' - \theta^\top \Sigma \hat{\theta}'| \leq \sqrt{\frac{\xi^2 \|\Sigma \hat{\theta}'\|^2}{Cm}} \cdot \log(2/\delta) \tag{59}$$

with probability at least $1 - \delta$.

It suffices now to obtain a high probability bound on $\|\hat{\theta}'\|$, showing that it is close in value to $\|\theta\|$. Let $\phi'_{i,k}$ and $\theta_k$ denote the $k$th elements of $\phi'_i$ and $\theta_k$, respectively. Similar to the previous proof, we have that

$$\|\phi'_{i,k} y'_i\|_{\psi_1} \lesssim \xi \tag{60}$$

by multiplication of the sub-Gaussian random variables. By Bernstein's inequality, with probability $1 - \delta$, for all $k \in [d]$,

$$|\frac{1}{m} \sum_{i \in [m]} \phi'_{i,k} y'_i - \theta_k| \leq \sqrt{\frac{\xi^2}{Cm} \cdot \log(2d/\delta)} \tag{61}$$

for some constant $C > 0$. Under the same event,

$$\|\hat{\theta}' - \theta\| \leq \sqrt{\frac{d\xi^2}{Cm} \cdot \log(2d/\delta)} \tag{62}$$

by standard norm inequalities. The triangle inequality then yields

$$\|\hat{\theta}'\| \leq \|\theta\| + \sqrt{\frac{d\xi^2}{Cm} \cdot \log(2d/\delta)} \tag{63}$$

Finally, we are able to put these three events together:

$$|\hat{\theta}^\top \Sigma \hat{\theta}' - \theta^\top \Sigma \theta| \leq \sqrt{\frac{\xi^2 \|\Sigma \theta\|^2}{Cm} \cdot \log(2/\delta)} + \sqrt{\frac{\xi^2 \|\Sigma \hat{\theta}'\|^2}{Cm} \cdot \log(2/\delta)} \tag{64}$$

$$\leq \sqrt{\frac{\xi^2 \|\Sigma \theta\|^2}{Cm} \cdot \log(2/\delta)} + \sqrt{\frac{2\xi^2 \|\Sigma\|^2 \|\theta\|^2}{Cm} \cdot \log(2/\delta)} \tag{65}$$

$$+ \sqrt{\frac{2\xi^4 \|\Sigma\|^2 d}{C_1 m^2} \cdot \log^2(2d/\delta)} \tag{66}$$

$$\leq \sqrt{\frac{8\xi^2 \|\Sigma\|^2 \|\theta\|^2}{C_2 m} \cdot \log(2/\delta)} + \sqrt{\frac{2\xi^4 \|\Sigma\|^2 d}{C_2 m^2} \cdot \log^2(2d/\delta)} \tag{67}$$

with probability at least $1 - 3\delta$ by the union bound over the three events. $\square$

Define $\tilde{\beta}_{a,a'} = \tilde{\Lambda}_{a,a} + \tilde{\Lambda}_{a',a'} - 2\tilde{\Lambda}_{a,a'}$, and $\tilde{Z} \sim N(0, \tilde{\Lambda})$ where $\tilde{\Lambda}$ is the result of the projection onto $\mathbb{S}_+^K$ using $\hat{\beta}$ as defined in Line 13 of Algorithm 2. Since $\Lambda$ is positive semidefinite, the fact that

$$|\beta_{a,a'} - \hat{\beta}_{a,a'}| \leq \mathcal{O}\left( \frac{\|\theta\| \log(K/\delta)}{\sqrt{n}} + \frac{\sqrt{d} \cdot \log^2(dK/\delta)}{n} \right), \tag{68}$$

and the optimality of $\tilde{\Lambda}$ in Algorithm 2, we have

$$|\hat{\beta}_{a,a'} - \tilde{\beta}_{a,a'}| \leq \mathcal{O}\left( \frac{\|\theta\| \log(K/\delta)}{\sqrt{n}} + \frac{\sqrt{d} \cdot \log^2(dK/\delta)}{n} \right). \tag{69}$$

Triangle inequality then immediately implies the following element-wise error bound on the increment

$$|\beta_{a,a'} - \tilde{\beta}_{a,a'}| \leq \mathcal{O}\left( \frac{\|\theta\| \log(K/\delta)}{\sqrt{n}} + \frac{\sqrt{d} \cdot \log^2(dK/\delta)}{n} \right) \tag{70}$$

with probability at least $1 - \delta$.

Now we apply the following error bound due to Chatterjee (2005).

**Lemma C.3** (Theorem 1.2, Chatterjee (2005)). *Let $W$ and $\tilde{W}$ be two Gaussian random vectors with $\mathbb{E}W_a = \mathbb{E}\tilde{W}_a$ for all $a \in [K]$. Define $\gamma_{a,a'} = \|W_a - W_{a'}\|_{L_2}^2$ and $\tilde{\gamma}_{a,a'} = \|\tilde{W}_a - \tilde{W}_{a'}\|_{L_2}^2$ and $\Gamma = \max_{a,a'} |\tilde{\gamma}_{a,a'} - \gamma_{a,a'}|$. Then,*

$$|\mathbb{E} \max_{a \in [K]} W_a - \mathbb{E} \max_{a \in [K]} \tilde{W}_a| \leq \sqrt{\Gamma \log K} \tag{71}$$

Therefore, by the union bound over at most $K^2$ terms $\beta_{a,a'}$, the final bound becomes

$$|U - \mathbb{E} \max_{a \in [K]} \tilde{Z}_a| \leq \mathcal{O}\left( \frac{\sqrt{\|\theta\|} \log(K/\delta)}{n^{1/4}} + \frac{d^{1/4} \log^{3/2}(dK/\delta)}{\sqrt{n}} \right) \tag{72}$$

with probability at least $1 - \delta$.

### C.1.1 Proof of the second inequality

Here we prove the second inequality in the theorem statement that $\sqrt{\log K} \cdot V^* \gtrsim U$

**Lemma C.4.** *Let $(W_a)_{a \in [K]}$ be a mean zero sub-Gaussian process such that $\|W_a - W_{a'}\|_{\psi^2} \lesssim \|W_a - W_{a'}\|_{L^2}$, then*

$$\mathbb{E} \max_{a \in [K]} W_a \gtrsim \max_{a,a' \in [K]} \|W_a - W_{a'}\|_{L^2} \tag{73}$$

*Proof.* Let random variable $W_b, W_{b'}$ achieve the maximum for $\max_{a,a' \in [K]} \|W_a - W_{a'}\|_{L^2}$.
$\mathbb{E} \max_{a \in [K]} W_a \geq \mathbb{E} \max(W_b, W_{b'})$ Define $Z = W_{b'} - W_b$, then

$$
\begin{aligned}
&\mathbb{E} \max(W_b, W_{b'}) \\
=&\mathbb{E}[W_b | Z \leq 0] \Pr[Z \leq 0] + \mathbb{E}[W_b + Z | Z > 0] \Pr[Z > 0] \\
=&\mathbb{E}[W_b | Z \leq 0] \Pr[Z \leq 0] + \mathbb{E}[W_b | Z > 0] \Pr[Z > 0] + \mathbb{E}[Z | Z > 0] \Pr[Z > 0] \\
=&\mathbb{E}[W_b] + \mathbb{E}[Z | Z > 0] \Pr[Z > 0] \\
=&\mathbb{E}[Z | Z > 0] \Pr[Z > 0]
\end{aligned}
$$

Since $\mathbb{E}[Z | Z > 0] \Pr[Z > 0] + \mathbb{E}[Z | Z < 0] \Pr[Z < 0] = 0$, we have

$$\mathbb{E}[Z | Z > 0] \Pr[Z > 0] = \mathbb{E}[|Z|]/2 \tag{74}$$

Thus, we just need to lower bound $\mathbb{E}[|Z|]$. Due to the sub-Gaussian assumption on $Z$, it holds that for a constant $K_0$,

$$\Pr(|Z| > t) \leq \exp(-\frac{t^2}{K_0 \|Z\|_{L^2}^2})$$

Let $C$ be a constant such that

$$
\begin{aligned}
&\int_{C\|Z\|_{L^2}}^{\infty} t \exp(-\frac{t^2}{K_0 \|Z\|_{L^2}^2}) dt \\
=&K_0 \|Z\|_{L^2}^2 \exp(-\frac{C^2}{K_0}) \\
=&\|Z\|_{L^2}^2/20.
\end{aligned}
$$

Then,

$$
\begin{aligned}
\|Z\|_{L^2}^2 &= 2 \int_0^\infty t \Pr(|Z| > t) dt \\
&= 2 \int_0^{C\|Z\|_{L^2}} t \Pr(|Z| > t) dt + 2 \int_{C\|Z\|_{L^2}}^\infty t \Pr(|Z| > t) dt \\
&\phantom{=} le 2 \int_0^{C\|Z\|_{L^2}} t \Pr(|Z| > t) dt + 2 \int_{C\|Z\|_{L^2}}^\infty t \exp(-\frac{t^2}{K_0\|Z\|_{L^2}^2}) dt \\
&\leq 2C\|Z\|_{L^2} \int_0^{C\|Z\|_{L^2}} \Pr(|Z| > t) dt + \|Z\|_{L^2}^2/10 \\
&\leq 2C\|Z\|_{L^2}\mathbb{E}[|Z|] + \|Z\|_{L^2}^2/10.
\end{aligned}
$$

This implies that $\mathbb{E}[|Z|] \geq \frac{9}{20C}\|Z\|_{L^2}$. Combining with Equation 74 yields

$$
\mathbb{E} \max_{a \in [K]} W_a \gtrsim \|W_{b'} - W_b\|_{L^2}
$$

$\square$

**Proposition C.5.** *Let $(Z_a)_{a \in [K]}$ be a mean zero Gaussian process, then*

$$
\mathbb{E} \max_{a \in [K]} Z_a \lesssim \sqrt{\log K} \max_{a,a' \in [K]} \|Z_a - Z_{a'}\|_{L^2} \tag{75}
$$

*Proof.* This is a simple corollary of Sudakov-Fernique's inequality (see Theorem 7.2.11 in Vershynin (2018)). Define mean zero Gaussian process $Y_a, a \in [K]$ such that each $Y_a$ is sampled independently from $N(0, \max_{a,a' \in [K]} \|Z_a - Z_{a'}\|_{L^2}^2)$. By Sudakov-Fernique's inequality, it holds that

$$
\mathbb{E} \max_{a \in [K]} Z_a \leq \mathbb{E} \max_{a \in [K]} Y_a.
$$

We conclude the proof by combining with classical fact that

$$
\max_{a \in [K]} Y_a \lesssim \sqrt{\log K} \max_{a,a' \in [K]} \|Z_a - Z_{a'}\|_{L^2}
$$

$\square$

Applying Lemma C.4 on $V^*$ yields

$$
V^* \gtrsim \max_{a,a' \in [K]} \| \langle \phi(X,a) - \phi(X,a'), \theta \rangle \|_{L^2}.
$$

By the definition of the Gaussian process $Z$, its increment is bounded by

$$
\max_{a,a' \in [K]} \| \langle \phi(X,a) - \phi(X,a'), \theta \rangle \|_{L^2} \tag{76}
$$

, therefore applying Proposition C.5 for $U$ yields

$$
U \lesssim \sqrt{\log K} \max_{a,a' \in [K]} \| \langle \phi(X,a) - \phi(X,a'), \theta \rangle \|_{L^2}.
$$

This concludes the proof.

### C.2 A marginally sub-Gaussian bandit instance that does not satisfy Assumption 4

Given a constant $C$, let us define a bandit instance with $K = 2$ as follows:

$$\phi(X, 1) \sim N(0, 1) \tag{77}$$

$$\phi(X, 2) = \begin{cases} \phi(X, 1) & \text{if } |\phi(X, 1)| \leq C; \\ -\phi(X, 1) & \text{if } |\phi(X, 1)| > C. \end{cases} \tag{78}$$

Since the marginal distribution of $\phi(X, 1)$ and $\phi(X, 2)$ are both $N(0, 1)$, it is easy to see that sub-Gaussian assumption of the Preliminaries (Section 2) is satisfied. To see how Assumption 4 fails to hold, we compute the sub-Gaussian norm and $L_2$ norm of $Z := \phi(X, 1) - \phi(X, 2)$. Notice that $Z$ has Gaussian tail when $|Z| > 2C$, and thus $\|Z\|_{\psi_2} = \Theta(1)$. For the $L_2$ norm, note that $\|Z\|_{L_2} = O(\int_C^\infty t^2 \exp(-t^2)dt) = O(C^2 \exp(-C^2))$ which can be made arbitrarily small by choosing large $C$. Therefore for any constant $L$, there exist a bandit instance such that the marginal sub-Gaussian assumption holds but Assumption 4 does not.

### C.3 Comments on zero-mean features

We have originally assumed each arm has zero-mean features so that $\mathbb{E}[\phi(X, a)] = 0$ for all $a \in \mathcal{A}$ for simplicity. To deal with the non-zero case, it is not sufficient to simply estimate the individual means and subtract them off, as it would yield an incorrect estimate since we aim to estimate the expected max of a random process. If the true process is indeed Gaussian, we can estimate the means and set the estimated Gaussian process $Z$ to have the estimated means. However, doing so with non-Gaussian processes would require a more sophisticated version of Talagrand's comparison inequality, and we are unaware of results of this kind. There are less refined alternatives such as adding back the maximum estimated mean or using a centered Gaussian process $Z$ with large enough increments to subsume the means. Both would guarantee an upper bound on $V^*$, but could affect the tightness of the bound in different ways.

## D Model Selection

The algorithm that achieves the regret bound in Theorem 3 is presented in Algorithm 4. For ease of exposition, we first present this result assuming known (identity) covariance. Unknown covariance is handled Appendix D.1. Here, we provide some further comparisons with existing literature. State-of-the-art model selection guarantees have typically exhibited a tension between either worse dependence on $d_*$ or worse dependence on $T$. For example, Foster et al. (2019); Lee et al. (2021) and our work all pay a leading factor of $T^{2/3}$ while maintaining optimal dependence on $d_*$. In contrast, Pacchiano et al. (2020b) gave an algorithm that gets the correct $\sqrt{T}$ regret but pays $d_*^2$ dependence. It has been shown that this tension is essentially necessary Marinov & Zimmert (2021); Zhu & Nowak (2021), except in special cases (e.g. with constant gaps Lee et al. (2021)) or in cases that assume full realizability. Thus, our improved model selection bound contributes to the former line of work that maintains good dependence on $d_*$ at the cost of slightly worse dependence on $T$.

---

**Algorithm 4** Model Selection with Gaussian Process Upper Bound

---

1: **Input**: Rounds $T$, failure probability $\delta \leq 1/e$, constants $C_0, C_1$
2: **if** $\Sigma_a, \Sigma_{a,a'}, \Sigma$ are known for all $a, a' \in [K]$ such that $a \neq a'$ **then**
3:     Set $t_{\min} = C_0 \log^{3/2}(T \log T/\delta)$
4:     Set $\alpha_t = C_1 \cdot \frac{d_2^{1/4} \log^{3/2}(K d_2 T/\delta)}{t^{1/3}}$
5: **else**
6:     Set $t_{\min} = C_0 \left(1 + \log^{3/2}(T \log KT/\delta) + \frac{\tau^4}{\rho^2}\left(d_2 + \log(KT/\delta)\right) + d_2 \log(KT/\delta)\right)$
7:     Set $\alpha_t = C_1 \cdot \frac{\sqrt{d_2} \log(K d_2 T/\delta)}{t^{1/2}} + C_2 \cdot \frac{d_2^{1/4} \log^{3/2}(K d_2 T/\delta)}{t^{1/3}}$
8: **end if**
9: Initialize exploration dataset $S_0 = \{\}$
10: Initialize algorithm $\text{Alg}_1 \leftarrow \text{Exp4-IX}(\mathcal{F}_1)$.
11: Sampler Bernoulli $Z_t \sim \text{ber}(t^{-1/3})$ for all $t \in [T]$
12: **for** $t = 1, \ldots, T$ **do**
13:     Sample independently $x_t \sim \mathcal{D}$ and
14:     **if** $Z_t = 1$ **then**
15:         Sample $a_t \sim \text{Unif}[K]$, observe $y_t$
16:         Add to dataset: $S_t = (x_t, a_t, y_t) \cup S_{t-1}$
17:         $\mathcal{T}_t = \mathcal{T}_{t-1}$
18:     **else**
19:         Sample $a_t$ from $\text{Alg}_t$, observe $y_t$
20:         Update $\text{Alg}_t$ with $(x_t, a_t, y_t)$
21:         $S_t = S_{t-1}$
22:         $\mathcal{T}_t = \{t\} \cup \mathcal{T}_{t-1}$
23:         $\text{Alg}_{t+1} \leftarrow \text{Alg}_t$
24:     **end if**
25:     Estimate $\hat{U}_t$ from exploration data $S_t$ and covariate data $\mathcal{T}_t$ (if unknown covariance matrices)
26:     **if** $t \geq t_{\min}$ and $\hat{U}_t > 2\alpha_t$ **then**
27:         Set algorithm $\text{Alg}_{t+1} \leftarrow \text{Exp4-IX}(\mathcal{F}_2)$
28:     **end if**
29: **end for**

---

The main idea is that the algorithm starts with model class $\mathcal{F}_1$, the simpler one, and runs an Exp4-like algorithm under $\mathcal{F}_1$. However, it will randomly allocate some timesteps for exploratory actions where the uniform random policy is applied. From the exploration data, if it is detected that the gap is non-zero with high confidence, then the algorithm switches to $\mathcal{F}_2$. The critical component of the algorithm is in detecting the non-zero gap and then bounding the worst-case performance when the gap is non-zero but it has not been detected yet.

We require several intermediate results in order to prove the regret bound. The first is a generic high probability regret bound for a variant of Exp4-IX as given by Algorithm 4 of Foster et al. (2019), which is a modification of the algorithm proposed by Neu (2015). In particular, define

$$\theta_i := \arg\min_{\theta \in \mathbb{R}^{d_i}} \frac{1}{K} \sum_{a \in [K]} \mathbb{E}_X \left(\phi_i(X,a)^\top \theta - Y(a)\right)^2,$$

$$\theta_{\text{diff}} := \theta_2 - \begin{bmatrix} \theta_1 \\ \mathbf{0} \end{bmatrix}, \quad \text{and} \quad V_i^* := \max_{\pi \in \Pi_i} V^\pi$$

where $\Pi_i := \{x \mapsto \arg\max_a \phi_i(x,a)^\top \theta \ : \ \theta \in \mathbb{R}^{d_i}\}$ is the induced policy class. Let $\pi_{\theta_i}$ be the argmax policy induced by $\theta_i$. Note that the policy $\pi_{\theta_1}$ may not be the same as the policy that maximizes value.

**Lemma D.1** (Foster et al. (2019), Lemma 23). *With probability at least $1 - \delta$, for any $t \in [T]$, Exp4-IX for model class $\mathcal{F}_i$ satisfies*

$$\sum_{s=1}^{t} V^{\pi_{\theta_i}} - V^{\pi_s} \leq \mathcal{O}\left(\sqrt{d_i t K \log(d_i)} \cdot \log(TK/\delta)\right) \tag{79}$$

The second result we require is high probability upper and lower bounds on the number of exploration samples we should expect to have at any time $t \in [T]$. We appeal to Lemma 2 of Lee et al. (2021), as the exploration schedules are identical.

**Lemma D.2** (Lee et al. (2021), Lemma 2). *There are constants $C_1, C_2 > 0$ such that, with probability $1 - \delta$, $C_1 t^{2/3} \leq |S_t| \leq C_2 t^{2/3}$ for $t \geq C_0 \log^{3/2}(T \log T/\delta)$.*

The last intermediate result leverages the upper bound estimator from Theorem 2. We will define a Gaussian process, which we prove will act as an upper bound on the gap in value between the model classes. Let $Z \sim \mathcal{N}(0, \Lambda)$ where

$$\Lambda_{a,a'} = \mathbb{E}\left[\langle \phi(X, a), \theta_{\text{diff}} \rangle \langle \phi(X, a'), \theta_{\text{diff}} \rangle\right] \tag{80}$$

for all $a, a' \in [K]$ and $\theta_{\text{diff}} = \theta_2 - \begin{bmatrix} \theta_1 \\ 0 \end{bmatrix}$. The following lemma establishes these upper bounds and shows that we can estimate $\mathbb{E} \max_{a \in [K]} Z_a$ at a fast rate. The critical property of this upper bound is that it is 0 when $\mathcal{F}_1$ satisfies realizability.

A simple transformation of the feature vectors allows us to apply the results from before. For datapoints $(x_i, a_i, y_i)$ collected by the uniform random policy, the following is an unbiased estimator of $\theta_2 - \begin{bmatrix} \theta_1 \\ 0 \end{bmatrix}$:

$$y_i \left(\phi_2(x_i, a_i) - \begin{bmatrix} \phi_1(x_i, a_i) \\ 0 \end{bmatrix}\right) = y_i \left(\phi_2(x_i, a_i) - \begin{bmatrix} \phi_1(x_i, a_i) \\ 0 \end{bmatrix}\right) = y_i \begin{bmatrix} 0 \\ \phi_{d_1:d_2}(x_i, a_i) \end{bmatrix} \tag{81}$$

where $\phi_{d_1:d_2}$ denotes the bottom $d_2 - d_1$ coordinates of the feature map $\phi$. As shorthand, we define $\tilde{\phi}_i(x, a) = \begin{bmatrix} 0 \\ \phi_{d_1:d_2}(x, a) \end{bmatrix}$. Note that $\|\tilde{\phi}_i\|_{\psi_2} \leq \tau$ and this feature vector still satisfies the conditions of Assumption 4 as we can simply zero the top coordinates. Furthermore, define $\tilde{\Sigma}_{a,a'} = \mathbb{E}\left(\tilde{\phi}(X, a) - \tilde{\phi}(X, a')\right)\left(\tilde{\phi}(X, a) - \tilde{\phi}(X, a')\right)^{\top}$ for $a \neq a'$. The estimators for this transformed problem are then

$$\hat{\theta} = \frac{1}{m} \sum_i y_i \tilde{\phi}_i \tag{82}$$

$$\hat{\theta}' = \frac{1}{m} \sum_i y_i' \tilde{\phi}_i' \tag{83}$$

And, as before, the quadratic form estimators are analogously

$$\hat{\beta}_{a,a'} = \hat{\theta}^{\top} \tilde{\Sigma}_{a,a'} \hat{\theta}' \tag{84}$$

**Lemma D.3.** *There is a constant $C$ such that the Gaussian process $Z$*

$$V^* - V^{\pi_{\theta_1}} \leq 2C \cdot \mathbb{E} \max_{a \in [K]} Z_a \tag{85}$$

*and, with probability at least $1 - \delta$, for all $n \in [T]$, the estimator $\hat{U}$ defined in Algorithm 4 with $n$ independent samples satisfies*

$$\left|\mathbb{E} \max_{a \in [K]} Z_a - \hat{U}\right| \leq \mathcal{O}\left(\frac{\sqrt{\|\theta_{diff}\|} \log(TK/\delta)}{n^{1/4}} + \frac{d_2^{1/4} \log^{3/2}(d_2 KT/\delta)}{\sqrt{n}}\right) \tag{86}$$

*Proof.* It is immediate that

$$V^* - \max_{\pi \in \Pi_1} V^\pi \leq V^* - V^{\pi_{\theta_1}} \tag{87}$$

since $\theta_1 \in \mathcal{F}_1$ by definition and $\pi_{\theta_1}$ is an argmax policy. This gap can then be bounded as

$$V^* - V^{\pi_{\theta_1}} = V^{\pi_{\theta_2}} - V^{\pi_{\theta_1}} \tag{88}$$

$$= \mathbb{E} \langle \phi_2(X, \pi_{\theta_2}(X)), \theta_2 \rangle - \mathbb{E} \langle \phi_2(X, \pi_{\theta_1}(X)), \theta_2 \rangle \tag{89}$$

$$= \mathbb{E} \langle \phi_2(X, \pi_{\theta_2}(X)), \theta_2 \rangle - \mathbb{E} \langle \phi_2(X, \pi_{\theta_1}(X)), \theta_2 \rangle \tag{90}$$

$$+ \mathbb{E} \langle \phi_1(X, \pi_{\theta_1}(X)), \theta_1 \rangle - \mathbb{E} \langle \phi_1(X, \pi_{\theta_1}(X)), \theta_1 \rangle \tag{91}$$

$$\leq \mathbb{E} \left\langle \phi_2(X, \pi_{\theta_2}(X)), \theta_2 - \begin{bmatrix} \theta_1 \\ 0 \end{bmatrix} \right\rangle + \mathbb{E} \left\langle \phi_2(X, \pi_{\theta_1}(X)), \begin{bmatrix} \theta_1 \\ 0 \end{bmatrix} - \theta_2 \right\rangle \tag{92}$$

$$\leq \mathbb{E} \max_{a \in [K]} \left\langle \phi_2(X, a), \theta_2 - \begin{bmatrix} \theta_1 \\ 0 \end{bmatrix} \right\rangle + \mathbb{E} \max_{a \in [K]} \left\langle \phi_2(X, a), \begin{bmatrix} \theta_1 \\ 0 \end{bmatrix} - \theta_2 \right\rangle \tag{93}$$

The Gaussian process $Z \sim \mathcal{N}(0, \Lambda)$ satisfies the conditions of Lemma C.1, which implies the Gaussian process upper bound on both of the above terms and, thus, the first claim.

Now we prove the estimation error bound. We apply Algorithm 2 with the constructed fast estimators for quadratic forms $\theta_{\text{diff}}^\top \tilde{\Sigma}_{a,a'} \theta_{\text{diff}}$ for all $a, a' \in [K]$. Let $\tilde{Z} \sim N(0, \tilde{\Lambda})$. We can apply Theorem 2 and get

$$\left| \mathbb{E} \max_{a \in [K]} Z_a - \mathbb{E} \max_{a \in [K]} \tilde{Z}_a \right| \leq \mathcal{O} \left( \frac{\sqrt{\|\theta_{\text{diff}}\|} \log(K/\delta)}{n^{1/4}} + \frac{d_2^{1/4} \log^{3/2}(d_2 K/\delta)}{\sqrt{n}} \right) \tag{94}$$

Setting $\hat{U} = \mathbb{E} \max_{a \in [K]} \tilde{Z}_a$ gives the result.

$\square$

**Lemma D.4.** *Let $\hat{U}$ be the estimate of $\mathbb{E} \max_{a \in [K]} Z_a$ from Lemma D.3 using the same method. Then, with probability $1 - \delta$,*

$$\mathbb{E} \max_a Z_a \leq C \hat{U} \log^{1/2}(K) + \mathcal{O} \left( \frac{(\|\theta_{\text{diff}}\|^{1/2} + d_2^{1/4}) \log(d_2 K/\delta) \log^{1/2}(K)}{\sqrt{n}} \right) \tag{95}$$

*for some constant $C > 0$.*

*Proof.* Here we let $C$ represent an absolute constant, which may change from line to line. For this, we require a multiplicative error bound, which is stated formally in Theorem 6. It is similar to the additive one developed in the proof of Theorem 2. From Theorem 6, and applying the union bound over all pairs of actions in $[K]$, we have with probability at least $1 - K^2\delta$, for all $a' \neq a$,

$$|\beta_{a,a'} - \hat{\beta}_{a,a'}| \leq \frac{\beta_{a,a'}}{2c} + \mathcal{O} \left( \frac{c(\|\Sigma_{a,a'}^{1/2} \theta\| + \sqrt{d}) \log^2(d_2/\delta)}{n} \right) \tag{96}$$

where we simply prepend $\Sigma^{1/2}$ to $\theta$ and the estimators and $c \geq 1$ is to be chosen later.

With this concentration, we now show that if $\hat{U} = \mathbb{E} \max_{a \in [K]} \tilde{Z}_a$ is small, then this must mean that $\max_{a,a'} \beta_{a,a'}$ is also small.

$$\left( \hat{U} \right)^2 \geq \left( \mathbb{E} \max_{a \in [K]} \tilde{Z}_a \right)^2 \tag{97}$$

$$\geq C \max_{a,a' \in [K] \,:\, a \neq a'} \|\tilde{Z}_a - \tilde{Z}_{a'}\|_{L^2}^2 \tag{98}$$

$$= \mathcal{O} \left( \max_{a,a'} \beta_{a,a'} - \frac{\beta_{a,a'}}{2c} - \frac{c(\|\theta_{\text{diff}}\| + \sqrt{d_2}) \log^2(d_2/\delta)}{n} \right) \tag{99}$$

for an absolute constant $C$. The second line uses Lemma C.4. The third line uses the concentration above. Choosing $c$ large enough (dependent only on absolute constants), we get

$$\max_{a \neq a'} \beta_{a,a'} \leq 2\hat{U}^2 + \mathcal{O}\left(\frac{(\|\theta_{\text{diff}}\| + \sqrt{d_2})\log^2(d_2/\delta)}{n}\right) \tag{100}$$

Then, from Proposition C.5, we get the statement:

$$U \leq C\sqrt{\log K} \cdot \sqrt{\max_{a \neq a'} \beta_{a,a'}} \leq C\hat{U}\sqrt{\log K} + \mathcal{O}\left(\frac{(\|\theta_{\text{diff}}\|^{1/2} + d_2^{1/4})\log(d_2/\delta)\log^{1/2}(K)}{\sqrt{n}}\right) \tag{101}$$

Changing the variable $\delta' = \delta/K^2$ gives the result.

$\square$

Armed with these facts, we can prove the regret bound. Let $E = E_1 \cap E_2 \cap E_3 \cap E_4$ denote the good event that satisfies the conditions laid out in the intermediate results where

1. $E_1$ is the event that $\sum_{s \in \mathcal{I}} V^{\pi_{\theta_i}} - V^{\pi_s} \leq \mathcal{O}\left(\sqrt{d_i|\mathcal{I}|K\log(d_i)} \cdot \log(TK/\delta)\right)$ for any interval of times up to $|\mathcal{I}| \leq T$.

2. $E_2$ is the event that $C_1 t^{2/3} \leq |S_t| \leq C_2 t^{2/3}$ for $t \geq t_{\min}$

3. $E_3$ is the event that the following inequality is satisfied for all $t_{\min} \leq t \leq T$:

$$\left|\mathbb{E}\max_{a \in [K]} Z_a - \hat{U}_t\right| \leq \mathcal{O}\left(\frac{\sqrt{\|\theta_{\text{diff}}\|}\log(TK/\delta)}{t^{1/6}} + \frac{d_2^{1/4}\log^{3/2}(d_2KT/\delta)}{t^{1/3}}\right) \tag{102}$$

4. $E_4$ is the event that the following is satisfied for all $t_{\min} \leq t \leq T$:

$$V^* - V^{\pi_1} \leq C\hat{U}_t\log^{1/2}(K) + \mathcal{O}\left(\frac{(\|\theta_{\text{diff}}\|^{1/2} + d^{1/4})\log(d_2KT/\delta)\log^{1/2}(K)}{t^{1/3}}\right) \tag{103}$$

*Proof of Theorem 3 with known covariance matrices.* First note that event $E$ holds with probability at least $1 - 4\delta$ via an application of the union bound (over $T$) and the intermediate results. We now work under the assumption that $E$ holds. The proof is divided into cases when $\mathcal{F}_1$ does and does not satisfy realizability.

First, we bound the instantaneous regret incurred during the exploration rounds. Note that the average value of the uniform policy is zero and $V^* \leq \mathcal{O}\left(\|\theta\|\sqrt{\log K}\right)$ by standard maximal inequalities. This establishes the bound on the instantaneous regret for these rounds.

1. When $\mathcal{F}_1$ satisfies realizability, the algorithm is already running Exp4-IX with model class $\mathcal{F}_1$ from the beginning, so we are left with verifying that a switch to $\mathcal{F}_2$ never occurs in this setting. This can be shown by realizing that $\mathbb{E}\max_{a \in [K]} Z_a = 0$ whenever $\mathcal{F}_1$ satisfies realizability. Therefore $\theta_{\text{diff}} = 0$ and, under the good event, we have that

$$\hat{U}_t \leq C\frac{d_2^{1/4}\log^{3/2}(d_2KT/\delta)}{t^{1/3}} \tag{104}$$

for a some constant $C > 0$. Therefore, for $C_1$ chosen large enough, $\hat{U}_t \leq 2\alpha_t$ for all $t \geq t_{\min}$ and thus a switch never occurs. In this case, the regret incurred is

$$\text{Reg}_T \leq \tilde{\mathcal{O}}\left(T^{2/3} \cdot \log^{1/2}(K) + \sqrt{d_1TK\log(d_1)} \cdot \log(TK/\delta) + t_{\min}\right) \tag{105}$$

where the first term is due to the upper bound on the number of exploration rounds in $E_2$ and the second term is due to the regret bound for Exp4-IX under model $\mathcal{F}_1$.

2. In the second case when $\mathcal{F}_1$ does not satisfy realizability we must bound the regret when the algorithm is still using $\mathcal{F}_1$. The regret may therefore be decomposed as

$$\text{Reg}_T \leq (V^* - V^{\pi_{\theta_1}}) \cdot t_* + \sum_{t \in [t_*]} V^{\pi_{\theta_1}} - V^{\pi_t} + \sum_{t=t_*+1}^{T} V^* - V^{\pi_t} \qquad (106)$$

where $t^*$ is the timestep that the switch is detected. From $t_*$ onward, the algorithm runs Exp4-IX with $\mathcal{F}_2$, so this last term is simply bounded by $\widetilde{\mathcal{O}}(\sqrt{d_2 KT})$ under event $E$. The same is true for the middle term.

Note that before the switch occurs it must be that $\hat{U}_{t_*-1} \leq \alpha_{t_*-1}$. Therefore, from event $E$,

$$V^* - V^{\pi_{\theta_1}} \leq C\hat{U}_t \log^{1/2}(K) + \mathcal{O}\left( \frac{(\|\theta_{\text{diff}}\|^{1/2} + d_2^{1/4}) \log(d_2 KT/\delta) \log^{1/2}(K)}{t^{1/3}} \right) \qquad (107)$$

$$\leq \mathcal{O}\left( \frac{d_2^{1/4} \log^{3/2}(d_2 KT/\delta) \cdot \log^{1/2}(K)}{t^{1/3}} \right) \qquad (108)$$

for $t = t_* - 1$. The final regret bound for this case is then

$$\text{Reg}_T \leq \mathcal{O}\left( d_2^{1/4} T^{2/3} \cdot \log^{3/2}(d_2 KT/\delta) \cdot \log^{1/2}(K) \right) \qquad (109)$$

$$+ \mathcal{O}\left( \sqrt{d_1 TK \log(d_1)} \cdot \log(TK/\delta) + \sqrt{d_2 TK \log(d_2)} \cdot \log(TK/\delta) + t_{\min} \right) \qquad (110)$$

$\square$

## D.1 Contextual Bandit Model Selection with Unknown Covariance Matrix and Completed Proof of Theorem 3

Here we consider a modification of Algorithm 4 and the proof of the previous section in order to handle the case where the covariance matrix is unknown. The addition is small and follows essentially by showing that the estimated covariance matrix is close to the true one while contributing negligibly to the regret. Throughout, we assume that $\Sigma \succeq \rho I$ for some constant $\rho > 0$ and $\rho = \Omega(1)$, which is also assumed by Foster et al. (2019). We will use the fact that $\tau = \mathcal{O}(1)$ and $\|\theta_*\| = \mathcal{O}(1)$.

Let the time $t$ be fixed for now. The covariance matrices are estimated from all previous data during non-exploration rounds. Let $\mathcal{T}_t$ denote the times up to time $t$ for which a non-exploration round occurred as defined in the algorithm (i.e. $Z_s = 0$ for $s \in \mathcal{T}_t$) For $a \neq a'$, we have

$$\hat{\Sigma}_1 = \frac{1}{|\mathcal{T}_t| K} \sum_{s \in \mathcal{T}_t, a} \phi_1(x_s, a) \phi_1(x_s, a)^\top$$

$$\hat{\Sigma}_2 = \frac{1}{|\mathcal{T}_t| K} \sum_{s \in \mathcal{T}_t, a} \phi_2(x_s, a) \phi_2(x_s, a)^\top$$

$$\hat{\Sigma}_{a,a'} = \frac{1}{|\mathcal{T}_t|} \sum_{s \in \mathcal{T}_t} (\phi(x_s, a) - \phi(x_s, a'))(\phi(x_s, a) - \phi(x_s, a'))^\top$$

$$\hat{\Sigma}_{a,a} = \frac{1}{|\mathcal{T}_t|} \sum_{s \in \mathcal{T}_t} \phi(x_s, a) \phi(x_s, a)^\top$$

We also define $\Sigma_i = \mathbb{E}\hat{\Sigma}_i$ and $\Sigma_{a,a'}$ is defined as before. Note that for $t \geq C_1$ for some constant $C_1 > 0$, event $E_2$ ensures that $|\mathcal{T}_t| \geq \frac{t}{2}$. Proposition 12 of Foster et al. (2019) ensures that the following conditions are satisfied with probability at least $1 - \delta$

$$1 - \epsilon \leq \lambda_{\min}^{1/2}\left(\Sigma_i^{-1/2} \hat{\Sigma}_i \Sigma_i^{-1/2}\right) \leq \lambda_{\max}^{1/2}\left(\Sigma_i^{-1/2} \hat{\Sigma}_i \Sigma_i^{-1/2}\right) \leq 1 + \epsilon$$

for all $i \in \{1,2\}$ where $\epsilon \leq \frac{50\tau^2}{\rho}\sqrt{\frac{d_2 + \log(8K^2/\delta)}{|\mathcal{T}_t|}}$ and thus we will require that

$$t_{\min} = C_0\left(C_1 + \log^{3/2}(T\log KT/\delta) + \frac{\tau^4}{\rho^2}\left(d_2 + \log(8K^2T/\delta)\right) + d_2\log(KT/\delta)\right) \tag{111}$$

for a sufficiently large constant $C_0 > 0$. For here on, we will assume that $t \geq t_{\min}$. Note that this new choice does not impact the regret bound significantly since $t_{\min}$ does not influence the regret under model class $\mathcal{F}_1$ and under model class $\mathcal{F}_2$ it contributes a factor linear in $d_2$ but only logarithmic in $T$. Observe that this choice of $t_{\min}$ for sufficiently large $C_0$ ensures that $\epsilon < 1/2$ in the above concentration result.

For convenience, we let $\phi_{s,1}$ denote the $d_1$-dimensional features while $\phi_{s,2}$ denotes the $d_2$-dimensional features at time $s \leq t$. Note that $S_t$ is the set of past times of uniform exploration up to point time $t$. We split the dataset $S_t$ randomly evenly into and let $\mathcal{S}_t$ and $\mathcal{S}'_t$ denote the time indices of each dataset of size $m = \frac{|S_t|}{2}$. We consider the following estimators:

$$\hat{\theta}_2 = \hat{\Sigma}_2^{-1}\left(\frac{1}{m}\sum_{s\in\mathcal{S}_t}\phi_2(x_s, a_s)y_s\right) \tag{112}$$

$$\hat{\theta}_1 = \hat{\Sigma}_1^{-1}\left(\frac{1}{m}\sum_{s\in\mathcal{S}_t}\phi_1(x_s, a_s)y_s\right) \tag{113}$$

and the difference

$$\hat{\theta}_{\text{diff}} = \left(\hat{\Sigma}_2^{-1} - \begin{bmatrix}\hat{\Sigma}_1^{-1} & 0 \\ 0 & 0\end{bmatrix}\right)\left(\frac{1}{m}\sum_{s\in\mathcal{S}_t}\phi_{s,2}y_s\right)$$

$$\theta_{\text{diff}} = \left(\Sigma_2^{-1} - \begin{bmatrix}\Sigma_1^{-1} & 0 \\ 0 & 0\end{bmatrix}\right)\mathbb{E}\left[\phi_{s,2}y_s\right]$$

and we use analogous definitions to define $\hat{\theta}'_{\text{diff}}$ and $\theta'_{\text{diff}}$ with the other half of the data $\mathcal{S}'_t$. Proposition F.1 ensures the following holds.

**Proposition D.5.** *For a fixed $t$ with $t_{\min} \leq t \leq T$, with probability at least $1 - \delta$, for all $a, a'$,*

$$\hat{\theta}_{\text{diff}}^\top \hat{\Sigma}_{a,a'} \hat{\theta}'_{\text{diff}}$$
$$\geq C_1 \theta_{\text{diff}}^\top \Sigma_{a,a'} \theta_{\text{diff}} - \mathcal{O}\left(\frac{\log(Kd_2T/\delta)}{t}\right) - \mathcal{O}\left(\frac{\sqrt{d_2}}{t^{2/3}}\cdot\log^2(Kd_2T/\delta)\right) - \mathcal{O}\left(\frac{d_2\cdot\log(Kd_2T/\delta)}{t}\right)$$

*and*

$$\hat{\theta}_{\text{diff}}^\top \hat{\Sigma}_{a,a'} \hat{\theta}'_{\text{diff}}$$
$$\leq C_2 \theta_{\text{diff}}^\top \Sigma_{a,a'} \theta_{\text{diff}} + \mathcal{O}\left(\frac{\log(Kd_2T/\delta)}{t}\right) + \mathcal{O}\left(\frac{\sqrt{d_2}}{t^{2/3}}\cdot\log^2(Kd_2T/\delta)\right) + \mathcal{O}\left(\frac{d_2\cdot\log(Kd_2T/\delta)}{t}\right)$$

*for absolute constants $C_1, C_2 > 0$.*

*Proof.* The proof follows almost immediately from the general result in Proposition F.1 with dataset $S_t$. Recall that event $E_2$ asserts that

$$t^{2/3} \lesssim |S_t| \lesssim t^{2/3}$$

We then have that that $|\mathcal{T}_t| \geq \frac{t}{2}$ and therefore $\epsilon = \mathcal{O}\left(\sqrt{\frac{d_2 + \log(K/\delta)}{t}}\right)$ for all covariance matrices with probability at least $1 - 2\delta$. We also set $\hat{\Lambda} = \hat{\Sigma}_{a,a'}$, consisting of $|\mathcal{T}_t|$ samples. It remains to find $\epsilon_0$ such that $\|\Lambda - \hat{\Lambda}\| \leq \epsilon_0$ and verify that $\epsilon_0 = \mathcal{O}(1)$ for sufficiently large $t$. A covering argument suffices.

Let $N(\gamma)$ denote the $\gamma$-net of the unit ball in $d_2$. Then,

$$\|\Lambda - \hat{\Lambda}\| \leq (1 - 2\gamma)^{-1} \max_{y \in N(\gamma)} \left\langle (\Lambda - \hat{\Lambda})y, y \right\rangle \tag{114}$$

Note that $|N(\gamma)| \leq C^d$ for some constant $C$, setting $\gamma = 1/4$. For all $y \in N(\gamma)$, by applying Bernstein's inequality, we have

$$\frac{1}{|\mathcal{T}_t|} \sum_{s \in \mathcal{T}_t} y^\top \left( \hat{\Lambda} - \Lambda \right) y = \mathcal{O}\left( \sqrt{\frac{d_2 \log(1/\delta)}{t}} + \frac{d_2 \log(1/\delta)}{t} \right) \tag{115}$$

for all $y \in N(\gamma)$ with probability at least $1 - \delta$. Therefore, we ensure that $\epsilon_0 = \mathcal{O}(1)$ for $t = \Omega(d_2 \log(1/\delta))$, which is accounted for in the new definition of $t_{\min}$. Applying the union bound over these events and a change of variable $\delta' = 10K^2\delta$ gives the result. $\qquad\square$

Using this new concentration result, we can immediately replace Lemma D.4 with the case when the covariance matrices are estimated from data.

Let $E_1'$ and $E_2'$ be the same events as $E_1$ and $E_2$ defined before. Then define the following new events:

1. $E_3'$ is the event that, for all $t$ such that $t_{\min} \leq t \leq T$,

$$\hat{U}_t \lesssim U + \sqrt{\beta_{a,a'} \log K} + \mathcal{O}\left( \frac{\sqrt{d_2} \log(Kd_2T/\delta)}{t^{1/2}} + \frac{d_2^{1/4} \log^{3/2}(Kd_2T/\delta)}{t^{1/3}} \right)$$

2. $E_4'$ is the event that, for all $t$ such that $t_{\min} \leq t \leq T$,

$$U \lesssim \hat{U}\sqrt{\log K} + \mathcal{O}\left( \frac{\sqrt{d_2} \log(Kd_2T/\delta)}{t^{1/2}} + \frac{d_2^{1/4} \log^{3/2}(Kd_2T/\delta)}{t^{1/3}} \right)$$

Finally, we define the intersection $E' = E_1' \cap E_2' \cap E_3' \cap E_4'$.

**Lemma D.6.** *$E'$ holds with probability at least $1 - 4\delta$.*

*Proof.* Events $E_1$ and $E_2$ each fail with probability at most $\delta$ as demonstrated in the previous section. Now we handle $E_3'$. Let $\beta_{a,a'} = \theta_{\text{diff}}^\top \Sigma_{a,a'} \theta_{\text{diff}}$ and let $\hat{\beta}_{a,a',t} = \hat{\theta}_{\text{diff},t}^\top \hat{\Sigma}_{a,a',t} \hat{\theta}_{\text{diff},t}$ denote its estimator at time $t$.

Lemma C.3 shows that

$$\hat{U}_t \leq U + \sqrt{\max_{a,a'} |\beta_{a,a',t} - \hat{\beta}_{a,a'}| \log K}$$

$$\lesssim U + \sqrt{\max_{a,a'} \beta_{a,a'} \log K} + \mathcal{O}\left( \frac{\sqrt{d_2} \log(Kd_2T/\delta) \log K}{t^{1/2}} + \frac{d_2^{1/4} \log(Kd_2T/\delta)\sqrt{\log K}}{t^{1/3}} \right)$$

where the second line fails for any $t_{\min} \leq t \leq T$ with probability at most $\delta$ by Proposition D.5 and a union bound.

For event $E_4'$, we may follow the proof of Lemma D.4 but instead leveraging Proposition D.5 and lower bound $\hat{U}_t$ with

$$\left( \hat{U}_t \right)^2 \gtrsim \max_{a,a'} \beta_{a,a'} - \mathcal{O}\left( \frac{d_2 \log(Kd_2T/\delta)}{t} + \frac{d_2^{1/2} \log^2(Kd_2T/\delta)}{t^{2/3}} \right)$$

where the last line fails for any $t_{\min} \leq t \leq T$ with probability at most $\delta$ following a union bound. Applying Proposition C.5,

$$
\begin{aligned}
U &\lesssim \sqrt{\log K} \cdot \sqrt{\max_{a,a'} \beta_{a,a}} \\
&\lesssim \hat{U}\sqrt{\log K} + \mathcal{O}\left(\sqrt{\frac{d_2 \log(Kd_2T/\delta)\log K}{t}} + \frac{d_2^{1/2}\log^2(Kd_2T/\delta)\log K}{t^{2/3}}\right)
\end{aligned}
$$

which implies $E_4'$. $\qquad\square$

*Proof of Theorem 3 with unknown covariances.* The remainder proof of Theorem 3 is now identical to the known covariance matrix case except that there is an estimation penalty of $\widetilde{\mathcal{O}}\left(\sqrt{d_2/t}\right)$ which is incorporated into the test via the updated definition of $\alpha_t$ and the slightly larger value of $t_{\min}$. This contributes only a factor of $\widetilde{\mathcal{O}}\left(\sqrt{d_2 T}\right)$ to the regret in the case where $\mathcal{F}_2$ is the correct model. $\qquad\square$

# E  Testing for Treatment Effect

## E.1  Setting and Algorithm

Here, we describe in more detail the treatment effect setting of Section 4.1.2. As further motivation for this setting, consider the studied problem of deciding whether to issue ride-sharing services for primary care patients so as to reduce missed appointments. A priori, it is unclear what interventions (e.g. text message reminders, ride-share vouchers, etc.) might actually be effective based on characteristics of a patient. In such cases, we would be interested in developing a sample-efficient test to determine this.

We maintain the assumption throughout that $r^*(x,a) = \langle \phi(x,a), \theta \rangle$. The primary difference between the models is that we may either use all actions $\mathcal{A}_1 \cup \mathcal{A}_2$ (which may be costly from a practical perspective) or just the basic set of actions in $\mathcal{A}_1$. There is a known control action $a_0$ in both $\mathcal{A}_1$ and $\mathcal{A}_2$ such that $\phi(x, a_0) = 0$. We will assume throughout that $|\mathcal{A}_1 \cup \mathcal{A}_2| = K$. We assume that there is at least one action other than the control $a_0$ in $\mathcal{A}_2$. That is, $|\mathcal{A}_2| \geq 2$. Since there are potentially two action sets, there is now ambiguity in the definition of $\Sigma$. Here, we use $\Sigma = \frac{1}{|\mathcal{A}_1|}\sum_{a \in \mathcal{A}_1} \mathbb{E}\phi(X,a)\phi(X,a)^\top$ and assume $\lambda_{\min}(\Sigma) \geq \rho > 0$. However, we could just as easily define it with respect to $\mathcal{A}_1 \cup \mathcal{A}_2$ and then the algorithm would change by taking samples uniformly from $\mathcal{A}_1 \cup \mathcal{A}_2$.

More specifically, we define the test as

$$
\Psi = \begin{cases} 0 & \hat{U} \lesssim C_1 \cdot \sqrt{\frac{d\log^2(dK/\delta)}{p}} + C_2 \cdot \frac{d^{1/4}\log^{3/2}(dK/\delta)}{\sqrt{n}} \\ 1 & \text{otherwise} \end{cases}
$$

for sufficiently large constants $C_1, C_2 > 0$.

## E.2  Analysis

**Lemma E.1.** *Let $\Delta = V_2^* - V_1^*$. There exists a constant $C > 0$ such that*

$$
\Delta \leq C \cdot \mathbb{E}_X \max_{a \in \mathcal{A}_2} Z_a
$$

*where $Z \sim \mathcal{N}(0, \Lambda)$ is a Gaussian process with covariance matrix $\Lambda$ and*

$$
\begin{aligned}
\Lambda_{a,a'} &= \theta^\top \mathbb{E}\left[\phi(X,a)\phi(X,a')^\top\right]\theta \\
\Lambda_{a,a} &= \theta^\top \Sigma_a \theta
\end{aligned}
$$

*for $a, a' \in \mathcal{A}$ and $a \neq a'$.*

*Proof.* Define the following alternative feature mapping $\psi : \mathcal{A}_1 \cup \mathcal{A}_2 \to \mathbb{R}^{2d}$.

$$\psi(x, a) = \begin{cases} \begin{bmatrix} \phi(x, a) \\ 0 \end{bmatrix} & \text{if } a \in \mathcal{A}_1 \\ \begin{bmatrix} 0 \\ \phi(x, a) \end{bmatrix} & \text{if } a \in \mathcal{A}_2 \setminus \mathcal{A}_1 \end{cases}$$

Note that we still have $\psi(x, a_0) = 0$ for the control action. Furthermore, it is readily seen that $r^*(x, a) = \langle \phi(x, a), \theta \rangle = \left\langle \psi(x, a), \begin{bmatrix} \theta \\ \theta \end{bmatrix} \right\rangle$ for all $x \in \mathcal{X}$ and $a \in \mathcal{A}_1 \cup \mathcal{A}_2$. Then, the gap can be bounded above by

$$\Delta = \mathbb{E}_X \left[ \max_{a \in \mathcal{A}_1 \cup \mathcal{A}_2} \left\langle \psi(X, a), \begin{bmatrix} \theta \\ \theta \end{bmatrix} \right\rangle - \max_{a \in \mathcal{A}_1} \left\langle \psi(X, a), \begin{bmatrix} \theta \\ \theta \end{bmatrix} \right\rangle \right]$$

$$= \mathbb{E}_X \left[ \max_{a \in \mathcal{A}_1 \cup \mathcal{A}_2} \left\langle \psi(X, a), \begin{bmatrix} \theta \\ \theta \end{bmatrix} \right\rangle - \max_{a \in \mathcal{A}_1 \cup \mathcal{A}_2} \left\langle \psi(X, a), \begin{bmatrix} \theta \\ 0 \end{bmatrix} \right\rangle \right]$$

$$\leq \mathbb{E}_X \left[ \max_{a \in \mathcal{A}_1 \cup \mathcal{A}_2} \left\langle \psi(X, a), \begin{bmatrix} 0 \\ \theta \end{bmatrix} \right\rangle \right]$$

$$\leq \mathbb{E}_X \max_{a \in \mathcal{A}_2} \langle \phi(X, a), \theta \rangle$$

where the second line follows because we have assumed that $a_0 \in \mathcal{A}_1$ and thus a value of 0 is always attainable in $\mathcal{A}_1$. The last line follows for a similar reason since $a_0 \in \mathcal{A}_2$ also. We may now apply our result in Theorem 2 which guarantees that the Gaussian process $Z$ majorizes $\{\phi(X, \cdot)\}$ with $\mathbb{E}_X \max_{a \in \mathcal{A}_2} \langle \phi(X, a), \theta \rangle \leq C \cdot \mathbb{E} \max_{a \in \mathcal{A}_2} Z_a$ for some constant $C > 0$. $\square$

*Proof of Theorem 4.* **No additional treatment effect**

First, we consider the case where there is no additional treatment effect by $\mathcal{A}_2$. That is, by definition $\mathbb{E}_X \max_{a \in \mathcal{A}_2} \langle \phi(X, a), \theta \rangle = 0$. Lemma C.4 ensures that

$$\max_{a, a' \in \mathcal{A}_2 \, : \, a \neq a'} \| \langle \phi(X, a) - \phi(X, a'), \theta \rangle \|_{L^2} \lesssim \mathbb{E}_X \max_{a \in \mathcal{A}_2} \langle \phi(X, a), \theta \rangle = 0$$

Since $\theta^\top \Sigma_{a, a'} \theta = \| \langle \phi(X, a) - \phi(X, a'), \theta \rangle \|_{L^2}$, we have that $U := \mathbb{E} \max_a Z_a = 0$. It remains to show that $\hat{U}$ concentrates quickly to $U$. For this, we leverage Proposition F.1. First, we must verify that $\| \Sigma_{a, a'} - \hat{\Sigma}_{a, a'} \|$. An identical covering argument in the proof of Proposition D.5 shows that $\| \Sigma_{a, a'} - \hat{\Sigma}_{a, a'} \| = \mathcal{O}(1)$ with probability at least $1 - \delta$ for $p \geq d \log(1/\delta)$.

Proposition 12 of Foster et al. (2019) ensures that the following conditions are satisfied with probability at least $1 - \delta$

$$1 - \epsilon \leq \lambda_{\min}^{1/2} \left( \Sigma^{-1/2} \hat{\Sigma} \Sigma^{-1/2} \right) \leq \lambda_{\max}^{1/2} \left( \Sigma^{-1/2} \hat{\Sigma} \Sigma^{-1/2} \right) \leq 1 + \epsilon$$

where $\epsilon \leq \mathcal{O} \left( \sqrt{\frac{d_2 + \log(8/\delta)}{p}} \right)$. Therefore Proposition F.1 yields

$$\hat{\theta}^\top \hat{\Sigma}_{a, a'} \hat{\theta} \lesssim \theta^\top \Sigma_{a, a'} \theta + \mathcal{O} \left( \frac{d \log(dK/\delta)}{p} + \frac{\sqrt{d} \log^2(dK/\delta)}{n} \right)$$

$$= \mathcal{O} \left( \frac{d \log(dK/\delta)}{p} + \frac{\sqrt{d} \log^2(dK/\delta)}{n} \right)$$

and

$$\hat{\theta}^\top \hat{\Sigma}_{a, a'} \hat{\theta} \gtrsim \theta^\top \Sigma_{a, a'} \theta - \mathcal{O} \left( \frac{d \log(dK/\delta)}{p} - \frac{\sqrt{d} \log^2(dK/\delta)}{n} \right) \tag{116}$$

$$\tag{117}$$

for all $a, a' \in \mathcal{A}_2$ with $a \neq a'$ with probability at least $1 - \delta$. Therefore, under this event, we conclude that

$$\hat{U} \lesssim U + \sqrt{\max_{a,a'} |\beta_{a,a'} - \hat{\beta}_{a,a'}| \cdot \log K} \tag{118}$$

$$= \mathcal{O}\left(\sqrt{\frac{d \log^2(dK/\delta)}{p}} + \frac{d^{1/4} \log^{3/2}(dK/\delta)}{\sqrt{n}}\right) \tag{119}$$

By the definition of the test, this implies that $\Psi = 0$ with probability at $1 - \delta$ when there is no additional treatment effect with $\mathcal{A}_2$

**Additional treatment effect** Next, we consider the case where there is an additional treatment effect. Following the proof of Lemma D.4 and leverage the result of (116), we have that

$$\hat{U}^2 \gtrsim \max_{a,a' \in \mathcal{A}_2 \,:\, a' \neq a} \beta_{a,a} - C_3 \frac{d \log(dK/\delta)}{p} - C_4 \frac{\sqrt{d} \log^2(dK/\delta)}{n}$$

for sufficiently large constants $C_1, C_2 > 0$. Therefore

$$\Delta \leq U$$

$$\lesssim \hat{U} \cdot \sqrt{\log K} + C_3 \sqrt{\frac{d \log^2(dK/\delta)}{p}} + C_4 \frac{d^{1/4} \log^{3/2}(dK/\delta)}{\sqrt{n}}$$

In other words,

$$\frac{\Delta}{\sqrt{\log K}} - C_3 \sqrt{\frac{d \log(dK/\delta)}{p}} - C_4 \frac{d^{1/4} \log(dK/\delta)}{\sqrt{n}} \lesssim \hat{U}$$

Therefore, for $\Delta = \Omega\left(\sqrt{\frac{d \log^3(dK/\delta)}{p}} + \frac{d^{1/4} \log^2(dK/\delta)}{\sqrt{n}}\right)$, we can guarantee that the left side is at least $C_1 \sqrt{\frac{d \log^2(dK/\delta)}{p}} + C_2 \frac{d^{1/4} \log^{3/2}(dK/\delta)}{\sqrt{n}}$, ensuring that $\Psi = 1$.

$\square$

## F  Unknown Covariance Matrix Case

Our goal in this section will be to extend the results developed in the latter half of the paper to the case where the covariance matrix is unknown. The first subsection is dedicated to establishing strong concentration guarantees for estimating quadratic forms in general regression setting. Following the general analysis, we will demonstrate an application to contextual bandits.

### F.1  General estimation

In this section, we consider the abstract regression setting where we observe features and responses $(\phi, y)$ where $\phi \in \mathbb{R}^d$ is a random vector and $y \in \mathbb{R}$ satisfies $y = \phi^\top \theta + \eta$ for some unknown parameter $\theta \in \mathbb{R}^d$ and zero-mean noise $\eta$.

As before, we assume that $\phi \sim \mathrm{subG}(\tau^2)$ and $\eta \sim \mathrm{subG}(\sigma^2)$. We assume access to two independent datasets of $m \in \mathbb{N}$ samples given by $D = \{\phi_i, y_i\}_{i \in [m]}$ $D' = \{\phi_i', y_i'\}_{i \in [m]}$. Furthermore, we consider features $\phi^{(1)} \in \mathbb{R}^{d_1}$ which are the top $d_1$ coordinates of $\phi$ where $d_1 \leq d$. We also consider a third dataset of size $m$ $\{z_i\}_{i \in [m]}$ where $z_i \in \mathbb{R}^d$ and $z_i \sim \mathrm{subG}(C\tau^2)$ for some constant $C > 0$.

Define the following notation:

$$\Sigma = \mathbb{E}\left[\phi\phi^\top\right] \succeq \rho$$

$$\Sigma^{(1)} = \mathbb{E}\left[\phi^{(1)}(\phi^{(1)})^\top\right] \succeq \rho$$

$$\theta^{(1)} = \underset{\theta \in \mathbb{R}^{d_1}}{\arg\min}\, \mathbb{E}\left((\phi^{(1)})^\top\theta - y\right)^2$$

$$\theta_{\text{diff}} = \theta - \begin{bmatrix} \theta^{(1)} \\ \mathbf{0} \end{bmatrix}$$

$$R^\dagger = \Sigma^{-1} - \begin{bmatrix} (\Sigma^{(1)})^{-1} & \mathbf{0} \\ \mathbf{0} & \mathbf{0} \end{bmatrix}$$

$$\hat{R}^\dagger = \hat{\Sigma}^{-1} - \begin{bmatrix} (\hat{\Sigma}^{(1)})^{-1} & \mathbf{0} \\ \mathbf{0} & \mathbf{0} \end{bmatrix}$$

$$\Lambda = \mathbb{E}z_i z_i^\top$$

$$\hat{\Lambda} = \frac{1}{m}\sum_{i \in [m]} z_i z_i^\top$$

where $\hat{\Sigma}$ and $\hat{\Sigma}^{(1)}$ are estimates of $\Sigma$ and $\Sigma^{(1)}$ (independent of $D$ and $D'$ and $\{z_i\}$) satisfying

$$1 - \epsilon \leq \lambda_{\min}^{1/2}\left(\Sigma^{-1/2}\hat{\Sigma}\Sigma^{-1/2}\right) \leq \lambda_{\max}^{1/2}\left(\Sigma^{-1/2}\hat{\Sigma}\Sigma^{-1/2}\right) \leq 1 + \epsilon \tag{120}$$

and the same for $\hat{\Sigma}^{(1)}$ and $\Sigma^{(1)}$ where $0 < \epsilon \leq 1/2$. We will further assume that $\|\Lambda - \hat{\Lambda}\| \leq \epsilon_0$ for some $\epsilon_0 > 0$ for $\epsilon_0 = \mathcal{O}(1)$. Note that this setting also subsumes the case where $d_1 = 0$. Here, we must just make the adjustment that $\Sigma^{(1)} = 0$ and define its inverse to also be zero. The remaining calculations are agnostic to this.

Our objective in this section will be derive a general high-probability error bound on the difference between $\theta_{\text{diff}}^\top\Lambda\theta_{\text{diff}}$ and $\hat{\theta}_{\text{diff}}^\top\hat{\Lambda}\hat{\theta}_{\text{diff}}'$ where we define the estimators

$$\hat{\theta}_{\text{diff}} = \hat{R}^\dagger\left(\frac{1}{m}\sum_{i \in [m]} \phi_i y_i\right) \tag{121}$$

$$\hat{\theta}_{\text{diff}}' = \hat{R}^\dagger\left(\frac{1}{m}\sum_{i \in [m]} \phi_i' y_i'\right) \tag{122}$$

**Proposition F.1.** *With probability at least $1 - \delta$,*

$$\hat{\theta}_{\text{diff}}\hat{\Lambda}\hat{\theta}_{\text{diff}}' \geq C_1\theta_{\text{diff}}^\top\Lambda\theta_{\text{diff}} - \mathcal{O}\left(\frac{\log(2/\delta)}{t} + \epsilon^2 + \frac{\sqrt{d}}{m}\cdot\log^2(2d/\delta)\right) \tag{123}$$

*and*

$$\hat{\theta}_{\text{diff}}\hat{\Lambda}\hat{\theta}_{\text{diff}}' \leq C_2\theta_{\text{diff}}^\top\Lambda\theta_{\text{diff}} + \mathcal{O}\left(\frac{\log(2/\delta)}{t} + \epsilon^2 + \frac{\sqrt{d}}{m}\cdot\log^2(2d/\delta)\right)$$

*for constants $C_1, C_2 > 0$*

*Proof.* We first make the important observation that $R^\dagger\phi_i y_i$ is an unbiased estimator of $\theta_{\text{diff}}$. It therefore suffices to show concentration of $\hat{\theta}$ to its mean and then bound the error between $\hat{R}^\dagger$ and $R^\dagger$. We must also bound the error due to the difference between $\hat{\Lambda}$ and $\Lambda$.

For convenience, also define $\hat{\mu} := \frac{1}{m} \sum_{i \in [m]} \phi_i y_i$ and $\hat{\mu}' := \frac{1}{m} \sum_{i \in [m]} \phi_i' y_i'$ and $\mu := \mathbb{E}[\phi y] = \Sigma \theta$. We will achieve this goal by a series of applications of the triangle inequality:

$$\hat{\mu}^\top \hat{R}^\dagger \hat{\Lambda} \hat{R}^\dagger \hat{\mu}' - \theta_{\text{diff}}^\top \Lambda \theta_{\text{diff}} = \hat{\mu}^\top \hat{R}^\dagger \hat{\Lambda} \hat{R}^\dagger \hat{\mu}' - \mu^\top \hat{R}^\dagger \Lambda \hat{R}^\dagger \mu \tag{124}$$

$$+ \mu^\top \hat{R}^\dagger \Lambda \hat{R}^\dagger \mu - \theta_{\text{diff}}^\top \hat{\Lambda} \theta_{\text{diff}} \tag{125}$$

$$+ \theta_{\text{diff}}^\top \hat{\Lambda} \theta_{\text{diff}} - \theta_{\text{diff}}^\top \Lambda \theta_{\text{diff}} \tag{126}$$

We will bound each of the three terms individually.

**Second Term**

$$\mu^\top \hat{R}^\dagger \hat{\Lambda} \hat{R}^\dagger \mu - \theta_{\text{diff}}^\top \hat{\Lambda} \theta_{\text{diff}} = \mu^\top \hat{R}^\dagger \hat{\Lambda} \hat{R}^\dagger \mu - \mu R^\dagger \hat{\Lambda} R^\dagger \mu \tag{127}$$

$$= \left\langle \hat{R}^\dagger \hat{\Sigma}_{a,a'} \hat{R}^\dagger \mu, \mu \right\rangle - \left\langle R^\dagger \hat{\Sigma}_{a,a'} R^\dagger \mu, \mu \right\rangle \tag{128}$$

$$= \left\langle \left( \hat{R}^\dagger - R^\dagger \right) \hat{\Lambda} \hat{R}^\dagger \mu, \mu \right\rangle - \left\langle R^\dagger \hat{\Lambda} \left( R^\dagger - \hat{R}^\dagger \right) \mu, \mu \right\rangle \tag{129}$$

$$\leq \|\hat{R}^\dagger - R^\dagger\| \cdot \|\hat{\Lambda} \hat{R}^\dagger \mu\| \cdot \|\mu\| + \|\hat{R}^\dagger - R^\dagger\| \cdot \|\hat{\Lambda} R^\dagger \mu\| \cdot \|\mu\| \tag{130}$$

$$\lesssim \epsilon \cdot \|\hat{\Lambda} \hat{R}^\dagger \mu\| + \epsilon \cdot \|\hat{\Lambda} R^\dagger \mu\| \tag{131}$$

where the last inequality applies Proposition 13 of Foster et al. (2019). Bounding the negative follows equivalent steps. Note that

$$\|\hat{\Lambda} \hat{R}^\dagger \mu - \hat{\Lambda} R^\dagger \mu\| \lesssim \|\hat{\Lambda}\| \cdot \|\hat{R}^\dagger - R^\dagger\| \cdot \|\mu\| \tag{132}$$

$$\lesssim \|\hat{\Lambda}\| \cdot \epsilon \tag{133}$$

Therefore, the prior display can now be bounded as

$$\epsilon \cdot \|\hat{\Lambda} \hat{R}^\dagger \mu\| + \epsilon \cdot \|\hat{\Lambda} R^\dagger \mu\| \leq \epsilon \cdot \left( \|\hat{\Lambda} R^\dagger \mu\| + \|\hat{\Lambda}\| \cdot \epsilon \right) + \epsilon \cdot \|\hat{\Lambda} R^\dagger \mu\| \tag{134}$$

$$\lesssim \epsilon^2 + \frac{\theta_{\text{diff}}^\top \hat{\Lambda} \theta_{\text{diff}}}{c} \tag{135}$$

for some constant $c > 0$ to be chosen later. Here, we have used the AM-GM inequality and the fact that $\|\hat{\Lambda} - \Lambda\| \leq \epsilon_0 = \mathcal{O}(1)$.

Throughout, we have used the fact

$$\|\hat{R}^\dagger - R^\dagger\| = \|\hat{\Sigma}^{-1} - \Sigma^{-1}\| + \|(\hat{\Sigma}^{(1)})^{-1} - (\Sigma^{(1)})^{-1}\| \lesssim \epsilon \tag{136}$$

which follows from Proposition 13 of Foster et al. (2019).

**First Term** A bound on the first term follows from a concentration argument.

$$|\hat{\mu}^\top \hat{R}^\dagger \hat{\Lambda} \hat{R}^\dagger \hat{\mu}' - \mu^\top \hat{R}^\dagger \hat{\Lambda} \hat{R}^\dagger \mu| \leq |\hat{\mu}^\top \hat{R}^\dagger \hat{\Lambda} \hat{R}^\dagger \mu - \mu^\top \hat{R}^\dagger \hat{\Lambda} \hat{R}^\dagger \mu| + |\hat{\mu}^\top \hat{R}^\dagger \hat{\Lambda} \hat{R}^\dagger \hat{\mu}' - \hat{\mu}^\top \hat{R}^\dagger \hat{\Lambda} \hat{R}^\dagger \mu| \tag{137}$$

Now again, we deal with both terms individually. Note that we have $\mathbb{E}[\phi_i y_i] = \mu$ and for any vector $v$, $\|v^\top \phi_i y_i\|_{\psi_1} \lesssim \tau \|v\| (\tau \|\theta_*\| + \sigma) = \xi \|v\|$.

Let $v = \hat{R}^\dagger \hat{\Sigma}_{a,a'} \hat{R}^\dagger \mu$. By Bernstein's inequality and independence of the data and covariance matrices,

$$|\hat{\mu}^\top \hat{R}^\dagger \hat{\Lambda} \hat{R}^\dagger \mu - \mu \hat{R}^\dagger \hat{\Lambda} \hat{R}^\dagger \mu| \lesssim \mathcal{O}\left( \sqrt{\frac{\xi^2 \|v\|^2 \cdot \log(2/\delta)}{m}} + \frac{\xi \|v\| \cdot \log(2/\delta)}{m} \right) \tag{138}$$

with probability at least $1 - \delta$. Then, note that

$$\|v\| = \|\hat{R}^\dagger \hat{\Lambda} \hat{R}^\dagger \mu\| \tag{139}$$

$$\lesssim \|\hat{R}^\dagger\| \|\hat{\Lambda}^{1/2}\| \|\hat{\Lambda}^{1/2} \hat{R}^\dagger \mu\| \tag{140}$$

Furthermore

$$\|\hat{R}^\dagger\| \lesssim \|\Sigma^{-1}\| + \|(\hat{\Sigma}^{(1)})^{-1}\| = \mathcal{O}(\rho^{-1}) \tag{141}$$

$$\|\hat{\Lambda}\| \lesssim \|\Lambda\| + \epsilon_0 = \mathcal{O}(1) \tag{142}$$

Then, we can say that

$$|\hat{\mu}^\top \hat{R}^\dagger \hat{\Lambda} \hat{R}^\dagger \mu - \mu^\top \hat{R}^\dagger \hat{\Lambda} \hat{R}^\dagger \mu| \lesssim C_1 \sqrt{\frac{\|\hat{\Lambda}^{1/2} \hat{R}^\dagger \mu\|^2 \cdot \log(2/\delta)}{m}} + C_2 \frac{\|\hat{\Lambda}^{1/2} \hat{R}^\dagger \mu\| \cdot \log(2/\delta)}{m} \tag{143}$$

$$\leq \frac{\|\hat{\Lambda}^{1/2} \hat{R}^\dagger \mu\|^2}{4c} + \mathcal{O}\left(\frac{\left(\|\hat{\Lambda}^{1/2} \hat{R}^\dagger \mu\| + 1\right) \cdot \log(2/\delta)}{m}\right) \tag{144}$$

for constants $C_1, C_2, c > 0$, where the last line applies the AM-GM inequality. For the other term, we have a similar upper bound by defining $\hat{v} = \hat{R}^\dagger \hat{\Lambda} \hat{R}^\dagger \hat{\mu}$:

$$|\hat{\mu}^\top \hat{R}^\dagger \hat{\Lambda} \hat{R}^\dagger \hat{\mu}' - \hat{\mu}^\top \hat{R}^\dagger \hat{\Lambda} \hat{R}^\dagger \mu| \lesssim \mathcal{O}\left(\sqrt{\frac{\|\hat{v}\|^2 \cdot \log(2/\delta)}{m}} + \frac{\|\hat{v}\| \cdot \log(2/\delta)}{m}\right) \tag{145}$$

And the norm is bounded as

$$\|\hat{v}\| = \|\hat{R}^\dagger \hat{\Lambda} R^\dagger \hat{\mu}\| \tag{146}$$

$$\lesssim \left(\frac{1}{\rho} + 1\right) \cdot \|\hat{\Lambda} R^\dagger \hat{\mu}\| \tag{147}$$

Furthermore,

$$\|\hat{\Lambda} \hat{R}^\dagger \hat{\mu} - \hat{\Lambda} \hat{R}^\dagger \mu\| \leq \|\hat{\Lambda} \hat{R}^\dagger\| \cdot \|\frac{1}{m} \sum_i \phi_i y_i - \mu\| \tag{148}$$

$$\leq \mathcal{O}\left(\|\hat{\Lambda} \hat{R}^\dagger\| \cdot \sqrt{\frac{\xi^2 d}{m}} \cdot \log(2d/\delta)\right) \tag{149}$$

where the last inequality follows from Lemma C.2 with probability at least $1 - \delta$. Therefore,

$$\frac{1}{1/\rho + 1} \|\hat{v}\| \leq \|\hat{\Lambda} \hat{R}^\dagger \hat{\mu}\| \lesssim \|\hat{\Lambda}^{1/2} \hat{R}^\dagger \mu\| + \mathcal{O}\left(\|\hat{\Lambda} \hat{R}^\dagger\| \cdot \sqrt{\frac{\xi^2 d}{m}} \cdot \log(2d/\delta)\right) \tag{150}$$

This yields the bound

$$|\hat{\mu}^\top \hat{R}^\dagger \hat{\Lambda} \hat{R}^\dagger \hat{\mu}' - \hat{\mu}^\top \hat{R}^\dagger \hat{\Lambda} \hat{R}^\dagger \mu| \lesssim \mathcal{O}\left(\sqrt{\frac{\|\hat{v}\|^2 \cdot \log(2/\delta)}{m}} + \frac{\|\hat{v}\| \cdot \log(2/\delta)}{m}\right)$$

$$\leq \mathcal{O}\left(\sqrt{\frac{\|\hat{v}\|^2 \log^2(2/\delta)}{m}}\right)$$

$$\leq \mathcal{O}\left(\sqrt{\frac{\|\hat{\Lambda}^{1/2} \hat{R}^\dagger \mu\|^2 \log^2(2/\delta)}{m}} + \sqrt{\frac{d \log^4(2d/\delta)}{m^2}}\right)$$

$$\leq \frac{\|\hat{\Lambda}^{1/2} \hat{R}^\dagger \mu\|^2}{4c} + \mathcal{O}\left(\frac{\sqrt{d}}{m} \cdot \log^2(2d/\delta)\right)$$

where the last line applies the AM-GM inequality.

Therefore, in total, the first term is bounded as

$$|\hat{\mu}^\top \hat{R}^\dagger \hat{\Lambda} \hat{R}^\dagger \hat{\mu}' - \mu^\top \hat{R}^\dagger \hat{\Lambda} \hat{R}^\dagger \mu| \le \frac{\mu^\top \hat{R}^\dagger \hat{\Lambda} \hat{R}^\dagger \mu}{2c} + \frac{\left( \|\hat{\Lambda}^{1/2} \hat{R}^\dagger \mu\| + 1 \right) \cdot \log(2/\delta)}{m} \tag{151}$$

$$+ \mathcal{O}\left( \frac{\sqrt{d}}{m} \cdot \log^2(2d/\delta) \right) \tag{152}$$

$$\le \frac{\mu^\top \hat{R}^\dagger \hat{\Lambda} \hat{R}^\dagger \mu}{2c} + \mathcal{O}\left( \frac{\sqrt{d}}{m} \cdot \log^2(2d/\delta) \right) \tag{153}$$

where we have used the fact that $\|\hat{\Lambda}^{1/2} \hat{R}^\dagger \mu\| = \mathcal{O}(1)$ under the good events.

**Term III** For the third term, we aim to show that $\theta_{\text{diff}}^\top \hat{\Lambda} \theta_{\text{diff}}$ is close to $\theta_{\text{diff}}^\top \Lambda \theta_{\text{diff}}$ and leverage Assumption 3 and Bernstein's inequality for bounded random variables to do so. Define $W_i = \theta_{\text{diff}}^\top (\phi_2(x_s, a) - \phi_2(x_s, a')) z_i z_i^\top \theta_{\text{diff}}$. Note that we have $|W_i| \le C$ for some constant $C > 0$ by Assumption 3. Therefore $\text{var}(W_i) \le \mathbb{E} W_i^2 \lesssim \mathbb{E}[W_i]$ since $W_i$ is non-negative. By Bernstein's inequality, we have

$$|\theta_{\text{diff}}^\top \hat{\Lambda} \theta_{\text{diff}} - \theta_{\text{diff}}^\top \Lambda \theta_{\text{diff}}| \le |\frac{1}{m} \sum_i W_i - \mathbb{E} W_i|$$

$$\le C_1 \sqrt{\frac{\text{var}(W_i) \log(2/\delta)}{m}} + \mathcal{O}\left( \frac{\log(2/\delta)}{t} \right)$$

$$\le \frac{\mathbb{E}[W_i]}{2c} + \mathcal{O}\left( \frac{\log(2/\delta)}{t} \right)$$

$$\le \frac{\theta_{\text{diff}}^\top \Lambda \theta_{\text{diff}}}{2c} + \mathcal{O}\left( \frac{\log(2/\delta)}{t} \right)$$

for some constant $c > 0$ to be chosen later with probability at least $1 - \delta$. The last inequality follows from applying the AM-GM inequality.

**Collecting all terms** Now that we have shown bounds on each of the terms, we are ready to prove the proposition:

From the bound on the first term, we have

$$\hat{\mu}^\top \hat{R}^\dagger \hat{\Lambda} \hat{R}^\dagger \hat{\mu}' \lesssim \left( 1 + \frac{1}{2c} \right) \mu^\top R^\dagger \hat{\Lambda} \hat{R}^\dagger \mu + \mathcal{O}\left( \frac{\sqrt{d}}{m} \cdot \log^2(2d/\delta) \right) \tag{154}$$

From the bound on the second term,

$$\mu^\top R^\dagger \hat{\Lambda} \hat{R}^\dagger \mu \lesssim \epsilon^2 + \left( 1 + \frac{1}{c} \right) \theta_{\text{diff}} \hat{\Lambda} \theta_{\text{diff}} \tag{155}$$

From the bound on the third term,

$$\theta_{\text{diff}} \hat{\Lambda} \theta_{\text{diff}} \lesssim \left( 1 + \frac{1}{2c} \right) \theta_{\text{diff}}^\top \Lambda \theta_{\text{diff}} + \mathcal{O}\left( \frac{\log(2/\delta)}{t} \right) \tag{156}$$

Therefore, for sufficiently large choice of $c > 0$ (not dependent on problem parameters), we have

$$\hat{\mu}^\top \hat{R}^\dagger \hat{\Lambda} \hat{R}^\dagger \hat{\mu}' \lesssim \theta_{\text{diff}}^\top \Lambda \theta_{\text{diff}} + \mathcal{O}\left( \frac{\sqrt{d}}{m} \cdot \log^2(2d/\delta) \right) + \epsilon^2 + \mathcal{O}\left( \frac{\log(2/\delta)}{t} \right) \tag{157}$$

For the other direction, we have

$$\hat{\mu}^\top \hat{R}^\dagger \hat{\Lambda} \hat{R}^\dagger \hat{\mu}' \gtrsim \left(1 - \frac{1}{2c}\right) \mu^\top R^\dagger \hat{\Lambda} \hat{R}^\dagger \mu - \mathcal{O}\left(\frac{\sqrt{d}}{m} \cdot \log^2(2d/\delta)\right) \tag{158}$$

$$\gtrsim \left(1 - \frac{1}{2c}\right)^2 \theta_{\text{diff}} \hat{\Lambda} \theta_{\text{diff}} - \epsilon^2 - \mathcal{O}\left(\frac{\sqrt{d}}{m} \cdot \log^2(2d/\delta)\right) \tag{159}$$

$$\gtrsim \left(1 - \frac{1}{2c}\right)^3 \theta_{\text{diff}} \Lambda \theta_{\text{diff}} - \mathcal{O}\left(\frac{\log(2/\delta)}{t}\right) - \epsilon^2 - \mathcal{O}\left(\frac{\sqrt{d}}{m} \cdot \log^2(2d/\delta)\right) \tag{160}$$

Taken together, the necessary events occur with probability at least $1 - 10\delta$ by the union bound.

$\square$

# G  Supporting Lemmas

The following is a proof of the moment bound in Lemma B.3.

*Proof of Lemma B.3.* For convenience, define $X_{(i)} = \phi(X, a_{(i)})$ and the same for $X'_{(i)}$. Define $A = \begin{bmatrix} \mathbf{0}_d & M \\ \mathbf{0}_d & \mathbf{0}_d \end{bmatrix}$ and $Z = \begin{bmatrix} X_{(i)} \\ X'_{(i)} \end{bmatrix}$. Note that $Z^\top A Z = X_{(i)}^\top M X'_{(i)}$ and $A^\top A = \begin{bmatrix} M^\top M & \mathbf{0}_d \\ \mathbf{0}_d & \mathbf{0}_d \end{bmatrix}$. By Lemma G.3, $Z \sim \text{subG}(C_0 \tau^2)$. Furthermore, $\mathbb{E} Z = 0$ and $\mathbb{E} Z Z^\top = \mathbb{I}_d$. The remaining proof utilizes a variation of the Hanson-Wright inequality due to Zajkowski (2020), stated in Lemma G.1[6]. By this inequality, there exists a constant $C > 0$ such that

$$\Pr\left(|Z^\top A Z - \mathbb{E}\left[Z^\top A Z\right]| \geq \xi\right) \leq \exp\left(-C \min\left\{\frac{\xi^2}{\tau^4 \|A\|_F^2}, \frac{\xi}{\tau^2 \|A\|_F}\right\}\right) \tag{161}$$

By direct calculation, we have that $\mathbb{E}\left[Z^\top A Z\right] = \text{tr}\,\mathbb{E}\left[Z Z^\top A\right] = 0$ and by Lemma G.2, $\|A\|_F \leq \sqrt{d}(\sigma^2 + L\|\theta\|^2)$. To bound the moment, we use the tail-sum-expectation for non-negative random variables. For convenience, define $\sigma_1 = \tau^2 \|A\|_F$.

$$\mathbb{E}|Z^\top A Z|^p = \int_0^\infty \Pr\left(|Z^\top A Z|^p \geq u\right) du \tag{162}$$

$$= \int_0^\infty p v^{p-1} \Pr\left(|Z^\top A Z| \geq v\right) dv \tag{163}$$

$$\leq \int_0^\infty p v^{p-1} \max\left\{e^{\frac{Cv^2}{\sigma_1^2}}, e^{\frac{Cv}{\sigma_1}}\right\} dv \tag{164}$$

$$\leq \int_0^\infty p v^{p-1} e^{\frac{Cv^2}{\sigma_1^2}} dv + \int_0^\infty p v^{p-1} e^{\frac{Cv}{\sigma_1}} dv \tag{165}$$

The first inequality used Lemma G.1. Consider the second term first. Let $r = Cv/\sigma_1$. Then, by a change of variables,

$$\int_0^\infty p v^{p-1} e^{\frac{Cv}{\sigma_1}} dv = p(\sigma_1/C)^p \int_0^\infty r^{p-1} e^{-r} dr \leq 3p(\sigma_1/C)^p \cdot p^p \tag{166}$$

---

[6]Critically, Lemma G.1 applies to quadratic forms of sub-Gaussian, dependent random variables, rather than requiring the coordinates of $Z$ to be independent as in the traditional Hanson-Wright inequality Rudelson et al. (2013); Hanson & Wright (1971). As a consequence, the second term in the minimum of the above tail bound depends on $\|A\|_F$ as opposed to the operator norm $\|A\|$. Further discussion may be found in Zajkowski (2020).

where we have used the Gamma function inequality $\int_0^\infty r^{p-1}e^{-r}dr \le 3p^p$ Vershynin (2018). Consider the first term. Let $r = Cv^2/\sigma_1^2$. Like the previous part, we may apply a change of variables.

$$\int_0^\infty pv^{p-1}e^{-Cv^2/\sigma_1^2}dv = \frac{1}{2}\int_0^\infty p\left(\frac{\sigma_1^2 r}{C}\right)^{\frac{p-1}{2}} e^{-r}\cdot\sqrt{\frac{\sigma_1^2}{rC}}\cdot dr \tag{167}$$

$$= \frac{p}{2}\left(\frac{\sigma_1^2}{C}\right)^{\frac{p}{2}}\int_0^\infty r^{\frac{p}{2}-1}e^{-r}dr \tag{168}$$

$$\le \frac{3p}{2}\left(\frac{\sigma_1^2}{C}\right)^{\frac{p}{2}}\cdot(p/2)^{(p/2)}. \tag{169}$$

Taking these two together,

$$\left(\mathbb{E}|Z^\top AZ|^p\right)^{1/p} \le \left(3p(\sigma_1/C)^p\cdot p^p + \frac{3p}{2}\left(\frac{\sigma_1^2}{C}\right)^{\frac{p}{2}}\cdot(p/2)^{(p/2)}\right)^{1/p} \tag{170}$$

$$\le C'\cdot\sigma_1(p+\sqrt{p}), \tag{171}$$

for some other constant $C' > 0$ since $p^{1/p}$ is bounded by a constant. Since we only consider $p \ge 1$, the claim follows. $\square$

**Lemma G.1** (Restatement of Corollary 2.8 of Zajkowski (2020)). *Let $X \sim \mathrm{subG}(\tau^2)$ be a centered random vector in $\mathbb{R}^d$ and $A \in \mathbb{R}^{d\times d}$. Then, there exists a constant $C > 0$ such that*

$$\Pr\left(|X^\top AX - \mathbb{E}\left[X^\top AX\right]|\right) \le \exp\left(-C\min\left\{\frac{\xi^2}{\tau^4\|A\|_F^2}, \frac{\xi}{\tau^2\|A\|_F}\right\}\right) \tag{172}$$

*where $\|\cdot\|_F$ is the Frobenius norm.*

**Lemma G.2.** *Let $(\phi, y)$ be generated under the uniform-random policy. Define $M = \mathbb{E}\left[y^2\phi\phi^\top\right]$ and $A = \begin{bmatrix}\mathbf{0}_d & M \\ \mathbf{0}_d & \mathbf{0}_d\end{bmatrix}$. Under Assumption 2, $\|A\| \le L\|\theta\|^2 + \sigma^2$ and $\|A\|_F \le \sqrt{d}(L\|\theta\|^2 + \sigma^2)$.*

*Proof.* By definition $\|A\|^2 = \sup_{v\,:\,\|v\|=1} v^\top A^\top Av = \sup_{v\,:\,\|v\|=1} v_1^\top M^\top Mv_1 = \|M\|^2$ where $v_1$ denotes the first $d$ coordinates of $v$. The first equality follows since $A^\top A = \begin{bmatrix}M^\top M & \mathbf{0}_d \\ \mathbf{0}_d & \mathbf{0}_d\end{bmatrix}$. Since $M$ is positive semi-definite,

$$\|M\| = \sup_{v\in\mathbb{R}^d\,:\,\|v\|=1} v^\top Mv \tag{173}$$

$$= \sup_{v\in\mathbb{R}^d\,:\,\|v\|=1} \mathbb{E}\left[y^2(\phi^\top v)^2\right] \tag{174}$$

$$= \sup_{v\in\mathbb{R}^d\,:\,\|v\|=1} \left\{\mathbb{E}\left[(\phi^\top v)^2(\phi^\top\theta)^2\right] + \mathbb{E}\left[(\phi^\top v)^2\eta^2\right]\right\} \tag{175}$$

$$\tag{176}$$

The second term is simply $\mathbb{E}\eta^2 = \sigma^2$ since $\mathbb{E}\phi\phi^\top = \mathbb{I}_d$ and $\phi$ and $\eta$ are independent. The first term may be bounded as $\mathbb{E}\left[(\phi^\top v)^2(\phi^\top\theta)^2\right] \le L\cdot\mathbb{E}\left[(\phi^\top v)^2\right]\mathbb{E}\left[(\phi^\top\theta)^2\right] = L\|\theta\|^2$ by Assumption 2. This concludes the first claim. For the second, we note that $\|A\|_F^2 = \mathrm{tr}\,A^\top A = \mathrm{tr}\,M^\top M \le d\|M\|^2$ and the second claim follows by applying the first. $\square$

**Lemma G.3.** *Let $X, Y\,\mathrm{subG}(\tau^2)$ be two independent sub-Gaussian vectors in $\mathbb{R}^d$. Then, $Z = \begin{bmatrix}X \\ Y\end{bmatrix} \sim \mathrm{subG}(C_0\tau^2)$ for some constant $C_0 > 0$.*

*Proof.* Let $v = \begin{bmatrix} v_1 \\ v_2 \end{bmatrix} \in \mathbb{R}^{2d}$ where $v_1, v_2 \in \mathbb{R}^d$ and $\|v\|_2 = 1$. Then, $v^\top Z = v_1^\top X + v_2^\top Y$ is the sum of independent sub-Gaussian variables where $v_1^\top X \sim \mathrm{subG}(\|v_1\|_2^2 \tau^2)$ and $v_2^\top Y \sim \mathrm{subG}(\|v_2\|_2^2 \tau^2)$ where both $\|v_1\|_2 \leq 1$ and $\|v_2\|_2 \leq 1$. Therefore $v^\top Z \sim \mathrm{subG}(C_0 \tau^2)$ for a constant $C_0 > 0$. Since $v$ was arbitrary, the statement follows. $\qquad\square$

## G.1 Multiplicative Error Bound for Estimating Norms

In this section, we prove a multiplicative error bound for estimating $\|\theta\|^2$, which can potentially be faster. The key is an application of the AM-GM inequality, similar to the work of Foster et al. (2019). As before, we will consider a dataset of $n$ samples split evenly into $D = \{\phi_i, y_i\}$ and $D' = \{\phi_i', y_i'\}$ each of size $m = \frac{n}{2}$. Define

$$\hat{\theta} = \frac{1}{m} \sum_{i \in [m]} \phi_i y_i \tag{177}$$

$$\hat{\theta}' = \frac{1}{m} \sum_{i \in [m]} \phi_i' y_i' \tag{178}$$

Then, we estimate $\theta^\top \theta$ with $\hat{\theta}^\top \hat{\theta}'$.

**Theorem 6.** *Let $\delta \leq 1/e$ and let $c > 1$ be a constant. With $\hat{\theta}$ and $\hat{\theta}'$ defined above with $n$ total samples, the following error bound holds with probability at least $1 - \delta$:*

$$|\hat{\theta}^\top \hat{\theta}' - \theta^\top \theta| \leq \frac{\theta^\top \theta}{2c} + \mathcal{O}\left( \frac{c(\|\theta\| + \sqrt{d}) \max\{\xi^2, \xi\} \log^2(d/\delta)}{n} \right) \tag{179}$$

*Proof.* Similar to the proof of Theorem 2, we apply the triangle inequality use Bernstein's inequality to bound two terms individually with high probability.

The decomposition becomes

$$|\hat{\theta}^\top \hat{\theta}' - \theta^\top \theta| \leq |\hat{\theta}^\top \theta - \theta^\top \theta| + |\hat{\theta}^\top \hat{\theta}' - \hat{\theta}^\top \theta| \tag{180}$$

We start with the first term. By Bernstein's inequality there is a constant $C > 0$ such that

$$\Pr\left( |\hat{\theta}^\top \theta - \theta^\top \theta| \geq \epsilon \right) \leq \exp\left( -C \min\left\{ \frac{m\epsilon^2}{\|\theta\|^2 \xi^2}, \frac{m\epsilon}{\|\theta\|\xi} \right\} \right) \tag{181}$$

since $y_i \theta^\top x_i$ is sub-exponential with $\|y_i \theta^\top x_i\|_{\psi_1} \leq \xi\|\theta\|$, as before. Rearranging, we have that with probability at least $1 - \delta$,

$$|\hat{\theta}^\top \theta - \theta^\top \theta| \leq \sqrt{\frac{\|\theta\|^2 \xi^2 \log(1/\delta)}{Cm}} + \frac{\|\theta\|\xi \log(1/\delta)}{Cm} \tag{182}$$

$$\leq \frac{\|\theta\|^2}{4c} + \frac{c\xi^2 \log(1/\delta)}{Cm} + \frac{c\|\theta\|\xi \log(1/\delta)}{Cm} \tag{183}$$

where the second line follows from the AM-GM inequality. Similarly, conditioned on the dataset $D$, the second term in the triangle inequality may be bounded as

$$|\hat{\theta}^\top \hat{\theta}' - \hat{\theta}^\top \theta| \leq \sqrt{\frac{\|\hat{\theta}\|^2 \xi^2 \log(1/\delta)}{Cm}} + \frac{\|\hat{\theta}\|\xi \log(1/\delta)}{Cm} \tag{184}$$

$$\leq \|\hat{\theta}\| \cdot \left( \sqrt{\frac{\xi^2 \log(1/\delta)}{Cm}} + \frac{\xi \log(1/\delta)}{Cm} \right) \tag{185}$$

with probability at least $1 - \delta$. Finally the proof Theorem 2 shows that, with probability $1 - \delta$,

$$\|\hat{\theta}\| \leq \|\theta\| + \sqrt{\frac{d\xi^2}{Cm}} \cdot \log(2d/\delta) \tag{186}$$

Under both of these events, we have

$$|\hat{\theta}^\top \theta' - \hat{\theta}^\top \theta| \leq \sqrt{\frac{\|\theta\|^2 \xi^2 \log(1/\delta)}{Cm}} + \frac{\|\theta\|\xi \log(1/\delta)}{Cm} \tag{187}$$

$$+ \frac{\sqrt{d}\xi^2 \log^{3/2}(2d/\delta)}{Cm} + \frac{\sqrt{d}\xi^2 \log^2(2d/\delta)}{(Cm)^{3/2}} \tag{188}$$

$$\leq \frac{\|\theta\|^2}{4c} + \frac{c\xi^2 \log(1/\delta)}{Cm} + \frac{\|\theta\|\xi \log(1/\delta)}{Cm} \tag{189}$$

$$+ \frac{\sqrt{d}\xi^2 \log^{3/2}(2d/\delta)}{Cm} + \frac{\sqrt{d}\xi^2 \log^2(2d/\delta)}{(Cm)^{3/2}} \tag{190}$$

where the second line again uses the AM-GM inequality. Putting all three events together and applying the union bound, we have with probability $1 - 3\delta$,

$$|\hat{\theta}^\top \hat{\theta}' - \theta^\top \theta| \leq \frac{\|\theta\|^2}{2c} + \mathcal{O}\left(\frac{c\|\theta\| \max\{\xi^2, \xi\} \log(1/\delta)}{m} + \frac{c\sqrt{d}\xi^2 \log^2(2d/\delta)}{m}\right) \tag{191}$$

Simplifying the error term gives the result. $\qquad\square$

## H   Experiment details

### H.1   Section 3.2 Experiments

We simulated a high-dimensional CB learning setting with $K = 2$ actions and $d = 300$ dimensions. The problem is high-dimensional in the sense that the number of samples $n \in \{10, 20, \ldots, 100\}$ is significantly smaller than $d$. The contexts $X$ are generated such that the $i$th coordinate of the features is distributed as an independent Rademacher random variable $\phi_{(i)}(X, a) \sim \text{Unif}\{-1, 1\}$ for $a \in [K]$ and we select $\theta \in \mathbb{R}^d$ uniformly at random from the unit ball. To reduce the computational burden, we set the degree $t = 2$ and did not split the data $q = 1$.

Algorithm 1 is shown in red (Moment). Additionally, we implemented several plug-in baselines based on linear regression: one that solves minimum-norm least squares problem (LR) and another that is ridge regression with regularization $\lambda = 1$ (LR-reg). Figure 1 shows the absolute value difference between the estimated $V^*$ values of the three methods and the true value of $V^*$. Error bars represent standard error over 10 trials.

We evaluated the estimated values of all three methods by empirically evaluating them. LR, LR-reg, and Approx were evaluated with 4000 samples. Moment was evaluated with 1000 samples since it is more computationally burdensome. We note that these difference in evaluation number should only affect the variance.

We approximated the max function with a polynomial of degree $t = 2$ by minimizing an $\ell_1$ loss under randomly 2000 uniformly randomly generated points in $[-2, 2]$. Note that this is a convex optimization problem and can be solved efficiently. This procedure can be done without any samples from the environment. The noise $\eta$ for the problem was generated uniformly randomly from the set $[-1/2, 1/2]$.

**Comparison with Kong et al. (2020)**   As discussed, the algorithm of Kong et al. (2020) assumes Gaussianity, meaning that we expect it to have significant bias in settings where the process $\{\langle \phi(X, a), \theta \rangle\}$ is not Gaussian. In this part, we demonstrate one such illustrative instance empirically. The setting is the same as before except that we set $\theta$ to be 1-sparse with $\theta_{i_*} = 10$ for some unknown index $i_*$ and $\theta_i = 0$ for all

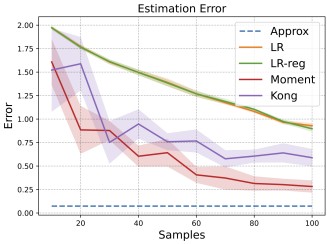

Figure 3: A comparison with the algorithm of Kong et al. (2020) in a similar setting to Figure 1 except that $\theta$ is 1-sparse. Kong et al. (2020) exhibits significant bias due to assuming that the reward process is Gaussian.

$i \neq i_*$. We may also generate better fitting polynomials in the sparse case by concentrating the optimization problem around relevant points encountered by the reward process such as $-10$ and $10$. Figure 3 shows the same evaluation as Figure 1 but in this setting instead.

Note that if we take $\theta$ to be random from the unit ball, we will typically end up with components that are all roughly of the same size meaning that the inner product $\langle \phi(X, a), \theta \rangle$ will look approximately Gaussian via something close to the central limit theorem. In such cases, the algorithm of Kong et al. (2020) may be competitive. However, this is a special case. In many other structured settings like the sparse setting above, the Gaussianity assumption becomes problematic.

## H.2 Section 4.1.2 Experiments

**Simulations.** We constructed another simulated high-dimensional CB setting where $|\mathcal{A}_1| = 3$ and $|\mathcal{A}_2| = 2$ where both $\mathcal{A}_1$ and $\mathcal{A}_2$ contain a control action $a_0$ where $\phi(x, a_0) = 0$ for all $x$. We set $d = 600$ and considered $n \in \{50, 75, 100, 150, 200, 250, 300\}$ which are all smaller than $d$. For feature vectors, we used a similar approach as the previous experiments. For $a \in \mathcal{A}_1$ with $a \neq a_0$, we chose contexts with $\phi_{(i)}(x, a) \sim$ Unif$\{-1, 1\}$ for the $i$th coordinate. For the non-control action in $\mathcal{A}_2$ we chose $\phi_{(i)}(x, a) \sim$ Unif$\{-1, 1\}$ for the first $\frac{d}{2}$ coordinates and then choose $\phi_{(i)}(x, a)$ for the last half. We set $p = 2000$ unlabeled samples.

We generated $\theta$ by first sampling from the unit ball in $\mathbb{R}^d$. Then, in the no effect setting, we set the first $\frac{d}{2}$ coordinates to zero. This ensures that $\mathcal{A}_2$ does not contribute to the reward under the optimal policy and thus $\Delta = 0$ between the two action sets. To evaluate the case with treatment effect, we just let $\theta$ keep its original value. Thus $\Delta$ will be positive (empirically found to be $\approx 0.133$, in this case). To calculate all expected maxes, we empirically evaluated them over 2000 samples. For each $n$, we regenerated the dataset 100 times. The lines in Figure 2 represent the averages over those 100 samples and the bands represent standard error.

The tests for both methods were as follows. For our method,

$$\Psi = \begin{cases} 0 & \hat{U} \leq 0.2 \left( \sqrt{\frac{d}{p}} + \frac{d^{1/4}}{\sqrt{n}} \right) \\ 1 & \text{otherwise} \end{cases}$$

and for the linear regression (LR) plug-in method, it was

$$\Psi = \begin{cases} 0 & \hat{W} \leq 0.33 \sqrt{\frac{d}{n}} \\ 1 & \text{otherwise} \end{cases}$$

$\hat{W}$ is defined as follows. Using an 80/20 split of the $n$ samples into datasets $D$ and $D'$ of sizes $|D| = n_{in}$ and $|D'| = n_{out}$, we compute

$$\hat{\theta} = \frac{1}{n_{in}} \sum_{i \in [n_{in}]} \phi_i y_i$$

with the majority of the data and then evaluate the difference

$$\hat{\Delta} = \frac{1}{n_{out}} \sum_{i \in [n_{out}]} \max_{a \in \mathcal{A}_2} \left\langle \phi(x_i', a_i'), \hat{\theta} \right\rangle$$

The lines in Figure 2 represent the means of the test outcomes over 100 repeated, independent samplings of the dataset of $n$ labeled points and $p$ unlabeled points.

### H.2.1 Warfarin experiments

**Warfarin data.** We first evaluated the ground-truth effect sizes, which we take to the difference between the average value attained by a single baseline action and the average value attained by a linear model that greedily chooses actions, when trained on the dataset with the same features. Note that, due to noise and misspecification on real world data, this might not be the true effect size, but demonstrates what is achievable with a linear model on the full dataset. As discussed in the main text, the reward is modeled as +1 if the correct dose is applied and 0 otherwise.

The data is structured as rows representing each patient with covariate features as well as a dosage of warfarin that is assumed to have succeeded for the patient (up to noise) given a rate measurement of mg/week. We divide the doses into three categories:

- *Low* for less than 21 mg/week

- *Medium* for 21 to 59 mg/week

- *High* for more than 59 mg/week

For patient covariates, we omitted 'Medications' and 'Comorbidities' since they are difficult to featurize. We also dropped rows with missing data in the 'Age', 'Height' and 'Weight' categories. For discrete features with missing data, we simply introduced another category. The baseline performance of each single action on the entire dataset is given below:

- *Low*: 0.32

- *Medium*: 0.56

- *High*: 0.12

To account for non-zero means of each arm in this dataset and the particular structure of the features of the patients, we consider a slight reformulation as a disjoint bandit. We denote the features of patient $x$ by $\phi(x) \in \mathbb{R}^d$ where $d = 193$. Then, we assume the reward has the following model:

$$r^*(x, a) = \mu_a + \langle \phi(x), \theta_a \rangle \tag{192}$$

for unknown vectors $\{\theta_a\}_{a \in \mathcal{A}}$ and unknown values $\mu_a$. Recall that we use $a$ to denote the baseline action and $a'$ to denote the target action. Similarly, we use $\mathcal{A} = \{a, a'\}$ to denote the target action set.

For the linear regression baseline, we perform standard unregularized linear regression on the training dataset to learn both $\hat{\mu}_a$ and $\hat{\theta}_a$ for the candidate actions. We then deploy the learned model on the validation set to estimate the expected difference:

$$\mathbb{E}\left[\max_{a' \in \mathcal{A}'} \{\mu_{a'} + \langle \phi(x), \theta_{a'} \rangle\}\right] - \mu_a \tag{193}$$

As this relies on accurate estimation of $\theta_a$, the threshold for detection scales approximately as $\sqrt{d/n}$ where $d$ is the dimension and $n$ is the number of training samples.

For our method, we split the training set into two equal parts, randomly. Both are used to learn the parameters $\hat{\mu}_a$ and $\hat{\mu}_{a'}$ and $\hat{\theta}_a$ and $\hat{\theta}_{a'}$. Using the validation data, we estimate the centered feature covariance matrix $\hat{\Sigma}$. We the form the covariance matrix:

$$\hat{\Lambda} = \begin{bmatrix} \hat{\theta}_a^\top \hat{\Sigma} \hat{\theta}_a & \hat{\theta}_a^\top \hat{\Sigma} \hat{\theta}_{a'} \\ \hat{\theta}_a^\top \hat{\Sigma} \hat{\theta}_{a'} & \hat{\theta}_{a'}^\top \hat{\Sigma} \hat{\theta}_{a'} \end{bmatrix} \tag{194}$$

which, we project onto the set of positive semi-definite matrices. In practice this step was not necessary since the resulting covariance matrices already had this property, but they are not guarnateed to. We then consider the normal distribution given by $\mathcal{N}(\hat{\mu}, \hat{\Lambda})$ and compute the difference between its expected maximum and the baseline: $\mathbb{E}_Z \max_{a' \in \mathcal{A}'} Z_{a'} - \hat{\mu}_a$, where $Z \sim \mathcal{N}(\hat{\mu}, \hat{\Lambda})$. Here, the threshold follows the rate $\sqrt{d/p} + d^{1/4}/\sqrt{n}$ where $d$ is the dimension, $n$ is the number of labeled training samples and $p$ is the number of unlabeled evaluation samples.

### H.3 Hardware

The experiments of Section 3.2 were run on a standard Amazon Web Services EC2 c5.xlarge instance. The experiments of Section 4.1.2 were conducted on a standard personal laptop with 16GB of memory and an Intel Core i7 processor.

## I Broader Impact

While this work is primarily theoretical, there are several conceivable applications of this theory that could have societal implications. Firstly, this work is meant to assist in the development of effective algorithms for contextual bandits. As a result, any applications of contextual bandit research are potentially influenced by this work such as health care, ads, education, recommender systems, and dynamic pricing. Specific to this paper, we mention applications in health care and testing for treatment effect, specifically for the efficient algorithm in Section 4. The algorithms presented here may be useful in health care settings to determine if it is worthwhile to pose a problem as a contextual bandit before conducting any procedures that might affect patients, even if only limited data is available. Our testing-for-treatment-effect application also has the potential to lower the sample complexity for clinical trials that evaluate the effectiveness of interventions. We advise practitioners to take note of the assumptions made here that may or may not hold in practice, such as realizability and sub-Gaussianity.

