# OpenReview forum: "Estimating Optimal Policy Value in Linear Contextual Bandits Beyond Gaussianity"
_TMLR — Accepted by TMLR_

### Review · Reviewer_m8CN · 2023-10-09

**Summary Of Contributions:**

The paper considers the problem of estimating the expected value of an optimal linear contextual bandit policy ($V^*$), while playing the associated linear contextual bandit problem. Previous work has derived effective schemes for such a problem but under a restrictive Gaussian assumption on the context distribution. This paper considers the more challenging setting where contexts are allowed to follow a general distribution.

The authors provide two main theoretical results: 1. that under full generality of the context distributions, the problem of estimating $V^*$ to within a tolerance $\epsilon$ is information theoretically hard, and 2. that under stronger assumptions (of boundedness and on the relationship between second and fourth moments) $V^*$ can be estimated using $\tilde{O}(\sqrt{d}/\epsilon^5)$ observations, where $d$ is the dimension of the context. The authors also derive an estimator with this property and verify its effectiveness in an empirical study.

The authors also derive a scheme to estimate an upper bound on $V^*$ which is provably more efficient (it has improved order wrt $\epsilon$) at estimating the upper bound than the regular algorithm is at estimating $V^*$ itself.

**Audience:**

Yes

**Broader Impact Concerns:**

I think these are adequately addressed by the remarks in Section I.

**Claims And Evidence:**

Yes

**Requested Changes:**

Title/terminology: I found the use of the phrase 'General Linear contextual bandit' potentially confusing, as it is so close to generalized linear which of course means something quite different, I wonder whether there is an alternative phrasing that would be less loaded?

Clarity around the importance of the sub-Gaussian feature map: I don't necessarily request a change, but would encourage the authors to provide some discussion in the rebuttal phase as to whether there does exist a more challenging unaddressed framework where the features are also general, and consider if such clarification would be beneficial in the paper.

In the Warfarin experiment, it seems that the only context considered are the features $\phi(x)$ themselves, so by assumption, the features/contexts are sub-Gaussian? Therefore, could the method of Kong et al. (2020) not be compared to also? This ties in to the previous point and it may be possible to address them together. However, I would suspect I'm not the only reader that may get confused about this point, so it may be worth some effort to clarify.

**Strengths And Weaknesses:**

The non-trivial theoretical contribution is a clear strength of the paper - the authors combine a range of state of the art techniques to derive and analyse their estimators, and I cannot find fault with the theoretical work that I have had the time to check (I must admit it has not been possible to verify all of it to the standard I might like, as the supplementary material is extensive. However, the most critical aspects seem accurate to me).

The experimental study is also sensibly chosen and well documented. It provides a nice accompaniment to the theoretical work. I think that the work as a whole is likely to be of interest to many in the learning theory and bandits communities.

It seems to me to be a weakness that all of the results focus on a setting where the underlying reward model assumes covariates can be mapped to sub-Gaussian features which interact with the regression parameters - what if the generally distributed contexts were combined directly with the regression parameters in the expected reward calculation? Would this be yet more challenging? Perhaps this is a very standard construction, that is not worth comment which I am just unaware of, but it is perhaps worth comment.

A further potential weakness is the dependence on the desired level of accuracy $\epsilon$. For your main moment based estimator of $V^*$, there remains the question of whether this $\epsilon^{-5}$ dependence is optimal.

---

> ### Author Response · Authors · 2023-11-14
> **Response**
>
> Thank you for your positive feedback on the paper!
>
> **A further potential weakness is the dependence on the desired level of accuracy… There remains the question of whether this $\epsilon^{-5}$ dependence is optimal.**
>
> In case there was any miscommunication, we would like to make sure the reviewer is aware that the version of this result with the fewest assumptions in Section 3 (Cor 3.5) is exponential in $1/\epsilon$ (see also Section 1.1).
>
> Nevertheless, we agree that determining whether this is improvable is an interesting but challenging question. For estimators based on polynomial approximation, exponential dependence is common, and it’s likely that it cannot be substantially improved. However, it is unclear presently whether an entirely different approach could eliminate the exponential dependence on $\epsilon$. In any case, we hope to emphasize that this result is there primarily to highlight that the sublinear in $d$ sample complexity is possible at all, regardless of $\epsilon$ dependence.
>
> **what if the generally distributed contexts were combined directly with the regression parameters in the expected reward calculation? Would this be yet more challenging?**
>
> Thanks for this question. Without an assumption like sub-Gaussianity, this would indeed be far more challenging as it would lead to issues with estimation of the covariance matrices even under standard “supervised learning” feedback. We do note that a more standard assumption in the bandit literature is that the norms of the features are bounded. While this implies sub-Gaussianity, one must be careful to ensure well-conditioning of the covariance matrices as well in our particular setting. However, we agree this is a really interesting consideration and will update the text to reflect this.
>
> **Title/terminology: I found the use of the phrase 'General Linear contextual bandit' potentially confusing… I wonder whether there is an alternative phrasing that would be less loaded?**
>
> Thanks for your feedback on the title! Would “Estimating Optimal Policy Value in Linear Contextual Bandits Beyond Gaussianity” be more appropriate in your view?
>
> **In the Warfarin experiment, it seems that the only context considered are the features**
>
> Thanks for pointing this out. Yes, this is indeed the case. The features are not action-dependent for this problem (in other words, the data is structured as a disjoint contextual bandit problem). However, there is still the issue that they are not Gaussian. In this case, the underlying estimators of both ours and Kong are essentially the same; however, in contrast to prior work, in this paper we provide a new method to leverage the estimator for hypothesis testing for treatment effects and the analysis required to prove that the test is valid requires handling sub-Gaussianity.

---

> > ### Comment · Reviewer_m8CN · 2023-11-17
> > **Follow-up on Response**
> >
> > Hi authors,
> >
> > Thanks for your detailed and clear reply to my comments. I don't have much to further to say other than to congratulate you a nice paper, that it seems the other reviewers also feel positively about. I think your suggested change to the title would be better, yes. It's difficult to be specific without writing a very long title, and I think this strikes a good balance.

---

### Review · Reviewer_D4ff · 2023-10-16

**Summary Of Contributions:**

This paper considers the problem of estimating the maximum expected reward $V^*$ for an optimal policy achieves in contextual linear bandits.
In particular, it focuses on the question of whether estimating $V^*$ can more efficient than learning the optimal policy itself,
in terms of the dependency on the dimensionality $d$ of the unknown parameter.
The main theoretical contributions are threefold:

- It is shown to be impossible to estimate $V^*$ accurately with sample complexity smaller than $\Omega(d)$ in general.
- An algorithm for $V^*$ estimation (Algorithm 1) is proposed, which achieves sample complexity of $O( \sqrt{d} )$,
under the assumption that the fourth moments of context distributions are bounded with the second moments (Assumption 2)
as well as the condition that the number of actions and the target accuracy are constant.
- An algorithm for estimating an upper bound for $V^*$ (Algorithm 2) is proposed, which works well under a kind of the joint sub-Gaussian assumption (Assumption 4).

In addition to these, applications to model selection and numerical experimental evaluations are given.

**Audience:**

Yes

**Claims And Evidence:**

Yes

**Requested Changes:**

- In Section 2 of problem setting, it is stated that the assumption that $\mathbb{E}[\phi(X, a)] = 0$ is easily relaxed.
I would appreciate an explanation or reference as to how exactly this can be relaxed.
- In Theorem 1, $C$ is described as an "absolute constant", but the proof suggests that it actually depends on parameters such as $\sigma$ and $\tau$ given in Section 2 and $L$ given in Assumption 2.
Is this understanding correct?
If so,
I don't think it is standard terminology to say that C is an absolute constant.
I recommend to state that $C$ depends on parameters $\sigma$, $\tau$ and $L$.

Minor typos:
- The first sentence of Section 1.1: the $V^*$ problem $\leftarrow$ the $V^*$ estimation problem
- The last sentence in Page 2: (Theorem 4 $\leftarrow$ (Theorem 4) (close bracket)
- Equation (33): $\zeta$ $\leftarrow$ a typo for $\gamma$?

**Strengths And Weaknesses:**

Strengths:

- The paper is well-structured, and its claims are supported by convincing mathematical arguments.
- Motivation and contribution are well explained.
- The topics of this paper, e.g., the separation of complexities of $V^*$ estimation and of best-policy identification, would be of interest to learning theory communities.

Weaknesses:

- For me, I did not find any major weaknesses.
I would appreciate it if the authors could take a look at Requested Changes and reflect them accordingly.

---

> ### Author Response · Authors · 2023-11-14
> **Response**
>
> Thanks for your positive review and suggestions! We will make the requested typo changes. Please see responses below.
>
> **In Section 2 of problem setting, it is stated that the assumption that $E[\phi(X,a)] = 0$ easily relaxed**
>
> Thanks for raising this. Indeed, it is easy for the first part of the results on exact V* estimation (Section 3), but less clear for the second set of results on estimating an upper bound on V* (Section 4). We will revise the language to reflect this.
>
> For Algorithm 1, the modification is immediate as we can simply consider polynomials that also include a first moment approximation and repeat the same calculations with a first order polynomial term.
>
> For Algorithm 2, there are several possible approaches which have trade-offs. First one could consider bounding $V^*$ by $E \left[\max_a Z_a \right] + \max_a \mu_a$ where $\mu_a$ is the non-centered mean of arm $a$ and $Z$ is the centered Gaussian process with matching increments of the centered process $\phi(X, a)^\top \theta$. The theorem follows immediately with this approach. This may be a sufficient upper bound for some settings, but not all.
>
> One might also consider a more refined approach by constructing a $Z$ gaussian process that shares $L_2$ increments matching the original process (with non-zero means). Algorithmically, this can be incorporated through a similar estimation procedure: we plug in estimates of the increments. To prove an analogous result, one can no longer apply Talagrand’s comparison inequality in its original form. However, one can apply Remark 8.5.4 of Ver18 then the majorizing measure theorem (Theorem 8.6.1 of Ver18) to get a version of the upper bound.
>
> It’s possible that a better version exists that incorporates the best of both approaches; however this would require a new version of Talagrand’s comparison inequality that handles a non-zero mean case where the original process means equal the Gaussian process means. While such a result exists for the analogous Sudakov-Fernique inequality, we are unaware of a statement for the comparison inequality. We suspect it is true but that is yet to be confirmed.
>
> Thanks again for pointing this out. We will revise the statement and add additional discussion, as per above.
>
> **If so, I don't think it is standard terminology to say that C is an absolute constant. I recommend to state that C depends on parameters $\sigma$, $\tau$ and $L$.**
>
> Thank you for raising this concern. Yes, C does depend on these parameters. Throughout, we assume these are constants of order $O(1)$ with respect to $d, \epsilon, K$. While this is common practice in bandits and statistical estimation literature for stating these bounds, we agree it is important to be clear with the terminology. We will update the draft to incorporate your suggestion.

---

### Review · Reviewer_QdYW · 2023-11-07

**Summary Of Contributions:**

The primary focus of this paper centers on estimating the magnitude of the optimal value function, denoted as $V^*$, within the context of linear contextual bandits. Specifically, the authors establish a theoretical lower bound, indicating that a sample complexity of at least $\Omega(d)$ is inevitable in general cases. In contrast, the authors introduce an algorithm that, given lower-order moments, achieves a sample complexity of $O(\sqrt{d}/\text{poly}(\epsilon))$. By leveraging these estimation techniques in various applications, the authors enhance the performance of several previous methods.

**Audience:**

Yes

**Broader Impact Concerns:**

There is no concern about the broader impact.

**Claims And Evidence:**

Yes

**Requested Changes:**

See Weaknesses.

**Strengths And Weaknesses:**

Strengths:

1. The author provides both lower and upper bounds for estimating the magnitude of the optimal value function, denoted as $V^*$, and demonstrates that estimating $V^*$ might be easier than finding the optimal policy.

2. Experimental results substantiate the performance of the proposed algorithm.

Weaknesses:

1. The paper doesn't clearly establish the significance of estimating the magnitude of the optimal value function $V^$ with an error up to $\epsilon$. While it's true that many algorithms may need to select parameters based on the magnitude of $V^$, estimating $V$ within the same order, such that $V \leq V^* \leq C \cdot V$, introduces only a constant factor in performance. It may not be reasonable to learn $V^$ with such a high level of accuracy when $\epsilon \ll V^$.

2. The sample complexity for Algorithm 1 exhibits an exponential dependency on both $K$ and $\epsilon, which is impractical. It is not clear why the author discusses Algorithm 1 when Algorithm 2 offers significantly improved performance.

---

> ### Author Response · Authors · 2023-11-14
> **Response**
>
> Thank you for your review and feedback!
>
> **Significance of estimating $V^\star$**
>
> Thank you for your comments on this. Indeed in many applications (such as our specific case of model selection), it may be sufficient to simply estimate the right ‘regime’ which might not require a lot of accuracy. However in many others, we may be interested in quickly estimating $V^*$ with some stronger guarantees than being within a multiplicative constant. For instance we may suspect that a different decision policy $\pi$ might have higher value, but there may also be significant logistic hurdles to implementing a new decision policy and it is only worthwhile from a stakeholder perspective if the new policy is “sufficiently” better.  In these cases, a rough estimate of the policy value may be sufficient (small to moderately sized $\epsilon$), but an estimate that is only within a constant multiplicative factor may be too coarse depending on the actual value of $V^*$. This contrasts with model selection where we care about $V^*$ relative to zero and a multiplicative factor is tolerable.
>
> **It’s not clear why the author discusses Algorithm 1 given its exponential dependence**
>
> Thank you for the opportunity to clarify this. We completely agree that the $\epsilon$-dependence is undesirable; however, we felt it was useful to include results on Algorithm 1 for two reasons. (1) We want to handle traditional cases where we might ask for some target accuracy without constant (unimprovable) multiplicative error that comes with Algorithm 2. (2) The algorithm and theorem serve as theoretical evidence that sublinear estimation of $V^*$  is possible in the first place. Prior to this, it wasn’t clear whether this was possible, regardless of $\epsilon$ and $K$ dependence, in the broader setting beyond the specific case of disjoint linear bandits with Gaussian covariates considered previously [Kong et al. 2020] . However, we do agree with your assessment that Algorithm 2 can likely find practical relevance much more easily. We will expand our discussion of these issues in the text.

---

### Decision · Action_Editor_h2oC · 2024-01-04

**Recommendation:** Accept as is

**Comment:**

At the end of the discussion, two out of three reviewers were happy to support acceptance of the paper. The third reviewer raised some concerns about the strength of the performance guarantees of Algorithm 1. This reviewer did not engage in further discussion about the paper, and in particular did not comment on the authors' response. Based on my own reading of the paper, I can understand where the original question of said reviewer may have come from, but I find the authors' response to be satisfactory. I recommend that the authors clarify the purpose of including Algorithm 1 in the final version of the paper so that future readers will be less likely to share the concern of reviewer QdYW.

Overall, in light of the otherwise positive reviews and my own evaluation of the paper, I believe that this is a strong submission that is definitely worthy of being published by TMLR.

**Audience:**

There is clearly a sizeable audience within the TMLR readership that will find this paper interesting.

**Claims And Evidence:**

All theoretical claims are supported by rigorous proofs, and all claims about empirical performance are supported by well-designed and well-documented experiments.